

# Acidity and the multiphase chemistry of atmospheric aqueous particles and clouds

Andreas Tilgner[1], Thomas Schaefer[1], Becky Alexander[2], Mary Barth[3], Jeffrey L. Collett, Jr.[4], Kathleen M. Fahey[5], Athanasios Nenes[6,7], Havala O. T. Pye[5], Hartmut Herrmann[1]*, and V. Faye McNeill[8]*

[1]Leibniz Institute for Tropospheric Research (TROPOS), Atmospheric Chemistry Department (ACD), Leipzig, 04318, Germany
[2]Department of Atmospheric Science, University of Washington, Seattle, WA, 98195, USA
[3]National Center for Atmospheric Research, Boulder, CO, 80307, USA
[4]Department of Atmospheric Science, Colorado State University, Fort Collins, CO, 80523, USA
[5]Office of Research and Development, US Environmental Protection Agency, Research Triangle Park, NC, 27711, USA
[6]School of Architecture, Civil and Environmental Engineering, Ecole Polytechnique Fédérale de Lausanne, Lausanne, CH-1015, Switzerland
[7]Institute for Chemical Engineering Sciences, Foundation for Research and Technology Hellas, Patras, GR-26504, Greece
[8]Departments of Chemical Engineering and Earth and Environmental Sciences, Columbia University, New York, NY, 10027, USA

*Correspondence to*: V. Faye McNeill (vfm2103@columbia.edu), H. Herrmann (herrmann@tropos.de)

**Abstract.**

The acidity of aqueous atmospheric solutions is a key parameter driving both the partitioning of semi-volatile acidic and basic trace gases and their aqueous-phase chemistry. In addition, the acidity of atmospheric aqueous phases, e.g. deliquesced aerosol particles, cloud and fog droplets, is also dictated by aqueous-phase chemistry. These feedbacks between acidity and chemistry have crucial implications for the tropospheric lifetime of air pollutants, atmospheric composition, deposition to terrestrial and oceanic ecosystems, visibility, climate, and human health. Atmospheric research has made substantial progress in understanding feedbacks between acidity and multiphase chemistry during recent decades. This paper reviews the current state of knowledge on these feedbacks with a focus on aerosol and cloud systems, involving both inorganic and organic aqueous-phase chemistry. Here, we describe the impacts of acidity on the phase partitioning of acidic and basic gases and buffering phenomena. Next, we review feedbacks of different acidity regimes on key chemical reaction mechanisms and kinetics, as well as uncertainties and chemical subsystems with incomplete information.

Finally, we discuss atmospheric implications and highlight needs for future investigations, particularly with respect to reducing emissions of key acid precursors in a changing world, and needs for advancements of field and laboratory measurements and model tools.



# 1    Introduction

The acidity of the atmospheric aqueous phase (i.e., deliquesced aerosol particles, cloud and fog droplets) impacts human health,

climate, and terrestrial/oceanic ecosystems (e.g., Pye et al. (2020) and references therein). Changes in acidity in these aqueous media can arise due to uptake of acidic or basic gases, coalescence, or chemical reactions in the aqueous phase. In turn, acidity of aerosols influences the phase partitioning of semi-volatile species, particulate matter (e.g., Nenes et al. (2020b)), their deposition rates (e.g., Nenes et al. (2020a)) and the rates and types of their chemical transformations. As a result of this two-way coupling between acidity and chemistry, acidity in atmospheric aqueous-aerosol matrices is controlled not only by

thermodynamic equilibrium, but also by mass transfer and chemical reaction kinetics, and emissions. Multiphase oxidation and reduction processes in atmospheric waters are strongly linked to the acidity-dependent uptake of acidic or basic compounds, which in turn affects the phase partitioning and the composition of aerosol particles. Moreover, the acidity level directly impacts chemical transformations, but the acidity itself is also influenced as a consequence of such processes. Figure 1 illustrates important tropospheric chemical processes in aqueous atmospheric matrices that are influenced by acidity and

affecting acidity.

The most important source of acidity in aqueous aerosols in the troposphere is the uptake and in-situ formation of strong acids, including sulfuric acid, a classic and important compound connected to anthropogenic pollution. Acid formation in aqueous atmospheric phases is itself influenced by acidity, but, more importantly, it also substantially increases the acidity of those media. Important acidity-influenced chemical processes, such as the conversion of sulfur(IV) to sulfur(VI) (Calvert et al.,

1985; Faloona, 2009; Harris et al., 2013; Turnock et al., 2019), as well as acid-driven and acid-catalyzed reactions of organic compounds (McNeill et al., 2012; Herrmann et al., 2015), contribute significantly to both secondary inorganic (SIA) and organic (SOA) aerosol formation. These constituents are often responsible for a large fraction of fine particulate matter (Jimenez et al., 2009). Due to their importance, they are strongly associated with aerosol effects on climate (Charlson et al., 1992; Boucher et al., 2013; Seinfeld et al., 2016; McNeill, 2017), air quality (Fuzzi et al., 2015), visibility (Hyslop, 2009),

ecosystems (Keene and Galloway, 1984; Adriano and Johnson, 1989; Baker et al., 2020) and human health (Pöschl, 2005a; Pope and Dockery, 2012; Lelieveld et al., 2015). Therefore, changes in acidity can significantly affect the global impacts of aerosols (Turnock et al., 2019).

Acidity-dependent chemical reactions also modify the tropospheric multiphase oxidant budget. For instance, the activation of halogen radicals is promoted by acidity (see Fig. 1 and Sect. 4.2) and can substantially affect the tropospheric oxidative

capacity (Vogt et al., 1996; von Glasow et al., 2002a; Pechtl and von Glasow, 2007; Sherwen et al., 2016; Sherwen et al., 2017; Hoffmann et al., 2019b). Acidity can indirectly affect aerosol and cloud composition by promoting the solubilization of transition metals and other bioavailable nutrients such as phosphorous (Meskhidze et al., 2005; Nenes et al., 2011; Shi et al., 2011; Stockdale et al., 2016). Soluble transition metal ions (TMIs) can initiate enhanced $HO_X$ chemistry in aqueous aerosol particles and clouds or catalyze S(IV) oxidation. Moreover, these solubilized metals, phosphorous and semi-volatile inorganic

reactive nitrogen molecules ($NH_3$, $HNO_3$) can deposit to the ocean surface, contribute to the bioavailable nutrient budget, and





thus impact biological activity and the carbon cycle. TMI solubilization also influences the impacts of atmospheric aerosols on human health (Fang et al., 2017). On the other hand, chemical interactions of marine and crustal primary aerosol constituents (e.g., carbonates, phosphates, halogens), dissolved weak organic acids, and other weak acids (e.g., $HNO_3$, $HCl$, $HONO$) and bases (e.g., ammonia and amines) can lead to a buffering of the acidity of aqueous solutions (see Fig. 1; Weber et al. (2016); Song et al. (2019a); and Sect. 2.2 for details).

**Figure 1.** Schematic of chemical processes influenced by and effects on acidity in tropospheric aerosols.

In comparison to other aqueous environments, such as sea water and continental surface waters, which are characterized by rather small acidity variations, atmospheric aqueous environments show much higher diversity (see Pye et al. (2020) for details). This is in part because of the huge concentration range of dissolved species in atmospheric waters, but also due to the decoupled exchange of acidic and basic species between the gas and condensed phases. Due to the technical challenges of





sampling and/or characterizing the pH of aerosols, fogs, and cloud water, there is also comparatively limited data on the acidity

of these phases in time and space. A companion article, Pye et al. (2020), provides a more complete overview of literature data
on the acidity of atmospheric waters, which we briefly summarize here: Typical pH values for cloud and fog droplets lie
between 2-7, while pH values for continental and marine aerosol particles have a larger range, -1-5 and 0-8, respectively
(Herrmann et al. (2015); Pye et al. (2020) and references therein). Because of the importance of aerosol and cloud acidity for
atmospheric processes and the environment, acidity has been a key subject of research for three decades. The majority of those

studies were focused on clouds, motivated by acid rain as well as SIA formation. A detailed review on observations,
thermodynamic processes and implications of atmospheric acidity is given in the companion paper (Pye et al., 2020).
Here, we review in detail the impact of acidity on the chemical transformations of atmospheric aerosols, clouds, and fog water,
with a focus on aqueous-phase chemical reaction kinetics and mechanisms. We also highlight how chemical reactions control
acidity in atmospheric aqueous media. We first discuss the uptake of acidic and basic gases and buffering phenomena, then

describe feedbacks between particle and droplet acidity and inorganic chemical reactions (SO₂ oxidation and halogen
chemistry) and aqueous-phase organic chemistry. Finally, a summary addresses atmospheric implications and needs for future
investigations, for example, in the context of reduced fossil fuel combustion emissions of key acid precursors in a changing
world.

## 2    Fundamental physical and chemical processes of importance for acidity

### 2.1    Aqueous-phase partitioning of acidic and basic gases

The partitioning of acidic or basic gases to atmospheric aerosols or cloud/fog droplets can have a major influence on
condensed-phase acidity. Likewise, the acidity of the aqueous phase itself influences the partitioning of dissociating species
from the gas phase. Condensed-phase acidity also governs back transfer or evaporation of dissociating compounds into the gas
phase (see Sect. 2.2).


*The phase partitioning of acids and bases*

The partitioning of a compound, between the gas phase, aqueous-phase, and its ionic forms, is usually achieved in < 1 hour
for fine-mode aqueous aerosols and small cloud droplets (Dassios and Pandis, 1998; Ervens et al., 2003; Ip et al., 2009; Koop
et al., 2011). Therefore, equilibrium conditions are often assumed in order to estimate the aqueous-phase concentrations.

Exceptions include large droplets with higher pH-values, droplets or particles with surface coatings, viscous aerosol particles,
or highly reactive dissolving compounds where mass transfer limitations in the gas or aqueous phase can prevent the attainment
of equilibrium partitioning on relevant timescales. The assumption of a thermodynamic equilibrium in such a case may result
in model biases (Ervens et al., 2003).





Assuming an ideal aqueous solution at equilibrium, i.e., neglecting, for example, mass transport limitations, chemical
production and degradation processes and non-ideal solution effects (i.e., considering the activity of ions in solution equal to
their aqueous concentration), the aqueous-phase concentration of a soluble compound ([A]$_{aq}$) is proportional to the partial
pressure of the compound in the gas-phase ($p_{A(air)}$) and its Henry's law constant $H_A$. The Henry's law constant (in mol L$^{-1}$ atm$^{-1}$)
is defined as:

$$H_A = \frac{[A]_{aq}}{p_{A(air)}} \tag{1}$$

Once an acid is taken up into an aqueous solution, it can dissociate into a hydrogen ion (H$^+$) and anions ($A^{z-}$), the degree of
which depends on its tendency for dissociation, characterized by an equilibrium dissociation constant $K_a$, and the acidity of
the aqueous environment. Consequently, an effective Henry's Law constant, $H_A^*$, e.g. for a diacid, is defined by Eq. 2a. For a
monoprotic acid, the third term in the parenthesis is omitted ($K_{a2} = 0$). For typical atmospheric monoprotic bases, such as NH$_3$
or dimethylamine, the corresponding effective Henry's Law constant, $H_A^*$, is defined by Eq. 2b. In Eq. 2b, $K_a$ is the equilibrium
dissociation constant $K_a$ of the base cation.

$$H_{A(acid)}^* = H_A \left(1 + \frac{K_{a1}}{[H^+]} + \frac{K_{a1}K_{a2}}{[H^+]^2}\right) \tag{2a}$$

$$H_{A(base)}^* = H_A \left(1 + \frac{[H^+]}{K_a}\right) \tag{2b}$$

Together with the liquid water content (LWC), the acidity of an aqueous solution can substantially affect the partitioning of
dissociating compounds to the aqueous aerosol or cloud phase. Increasing acidity leads to a decrease of the effective
partitioning of acids and to an increase in the effective partitioning of bases, and vice versa. For example, the partitioning of
nitrate to the particle phase varies dramatically across the typical range of aerosol pH, with nearly 100% of nitrate existing as
HNO$_3$ in the gas phase at pH 1, and near-complete particle-phase partitioning at pH 4. As a result, even small biases in predicted
particle pH in air quality models can result in over- or under-predictions of fine particle mass (Vasilakos et al., 2018). Since
atmospheric waters are typically acidic, bases are predominantly present in their protonated form and their partitioning is not
greatly altered by typical variations in pH. Hence, this section mainly focuses on the impact of acidity on the partitioning of
weak acids into aqueous aerosols and cloud/fog droplets.

From Eq. 1 and the ideal gas law, the concentration of the dissociating compound in the gas ($C_{A_{air}}$) and aqueous ($C_{A_{aq}}$) phase
with respect to the volume of air can be determined. Moreover, the aqueous-phase fraction of A ($X_{A_{aq}}$), i.e. the ratio of the
aqueous-phase concentration of compound A and the overall multiphase concentration (sum of A in gas and aqueous phase
(including undissociated and dissociated forms of A)) can accordingly be calculated by Eq. 3 (see Seinfeld and Pandis (2006)
for details).

$$X_{A_{aq}} = \frac{C_{A_{aq}}}{(C_{A_{aq}} + C_{A_{air}})} = \frac{H_A^* \cdot R^* \cdot T \cdot LWC \cdot 10^{-6}}{1 + H_A^* \cdot R^* \cdot T \cdot LWC \cdot 10^{-6}} \tag{3}$$

Here, $C_{A_{air}}$ is the concentration of A in air [mol L$^{-1}_{air}$], $C_{A_{aq}}$ is the aqueous-phase concentration of A in the volume of air
[mol L$^{-1}_{air}$], R$^*$ is the universal gas constant [0.082058 atm L$_{air}$ mol$^{-1}$ K$^{-1}$] ; T [K] is the temperature, $H_A^*$ is the effective Henry's





Law constant [mol $L^{-1}_{water}$ atm$^{-1}$] and LWC is the liquid water content [g m$^{-3}_{air}$]. Considering activities instead of concentrations, Eq. 3 modifies to Eq. 3a and 3b for monoprotic acids and bases (see Nenes et al. (2020b) and Guo et al. (2017) for details):

$$X_{aq,acid} = \frac{H_A \cdot K_{a1} \cdot R^* \cdot T \cdot LWC \cdot 10^{-6}}{\gamma_{H+} \cdot \gamma_{A-} \cdot [H^+] + H_A \cdot K_{a1} \cdot R^* \cdot T \cdot LWC \cdot 10^{-6}} \tag{3a}$$

$$X_{aq,base} = \frac{H_A \cdot K_{a1} \cdot R^* \cdot T \cdot LWC \cdot 10^{-6}}{1 + \frac{\gamma_{H+}}{\gamma_{B+}} \frac{H_A \cdot [H^+]}{K_{a1}} \cdot R^* \cdot T \cdot LWC \cdot 10^{-6}} \tag{3b}$$

where $\gamma_{H+}, \gamma_{A-}, \gamma_{B+}$ are the single-ion activity coefficients for H$^+$, the acid anion (A$^-$) and the base cation (B$^+$), respectively,

which can be calculated for a known ion composition using thermodynamic models (e.g., ISORROPIA-II (Fountoukis and Nenes, 2007), E-AIM (Clegg and Seinfeld, 2006), AIOMFAC (Zuend et al., 2008)).

Figure 2 displays the aqueous fraction, $X_{A_{aq}}$, of 8 weak atmospheric acids (sulfurous acid, nitrous acid, formic acid, acetic acid, glycolic acid, lactic acid, benzoic acid, phthalic acid, 2-nitrophenol, 2,4-dinitrophenol) and two important atmospheric bases (ammonia and dimethylamine) as a function of the LWC and acidity, calculated by Eq. 3. For the plots, an acidity range

([H$^+$] = 10$^{-1}$-10$^{-7}$ mol L$^{-1}$) and a liquid water content range (10$^{-6}$-1 g m$^{-3}$) have been considered that represent typical values for tropospheric aqueous aerosols, cloud/fog droplets and haze (see Herrmann et al. (2015)). A temperature of 298 K was assumed. It should be noted that temperature plays an important role on the effective solubility of trace gases. In general, as temperature decreases, the trace gas effective solubility increases. Thus, clouds at the top of the mixing layer height ($\sim$ 285 K typically) have higher aqueous fractions than aerosol water near the surface on a hot summer day. Similarly, winter hazes should also

have higher aqueous fractions than summertime haze events. Therefore, the aqueous fractions shown in Fig. 2 should be used carefully. Note, the $H_A$ and $pK_a$ values applied for the idealized calculation of LWC and acidity dependent aqueous fraction $X_{A_{aq}}$ are listed in Table S1 in the Supporting Information.

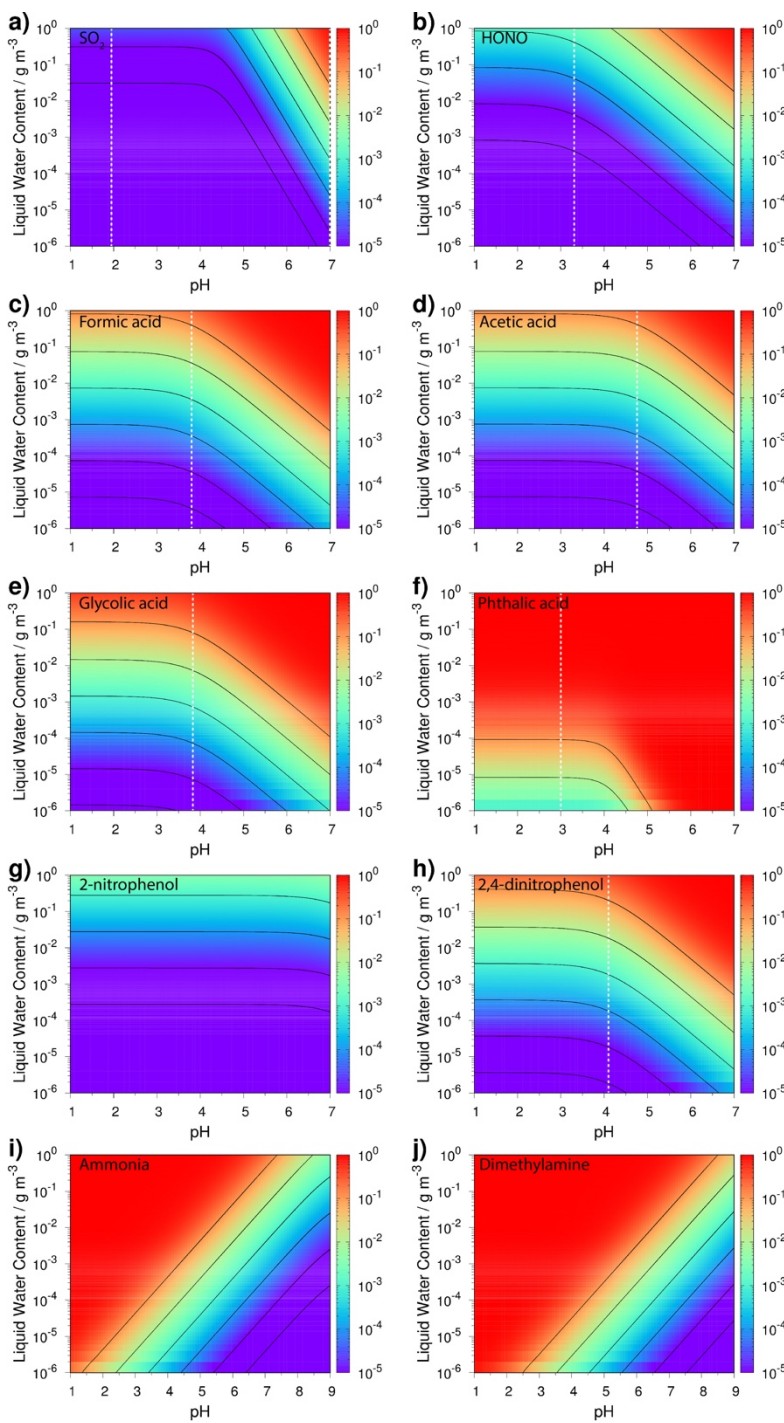

**Figure 2. Calculated aqueous-phase fraction $X_{A_{aq}}$ of 8 selected weak acids (a: SO₂, b: HONO, c: Formic acid, d: Acetic acid, e: Glycolic acid, f: Phthalic acid, g: 2-nitrophenol, h: 2,4-dinitrophenol) and bases (i: Ammonia, j: Dimethylamine) as a function of the LWC and acidity. The black lines are the isolines of the aqueous fractions of $10^{-i}$ (i = 1,..,6). The dashed white lines indicate $pK_a$ values of the corresponding acids (except for the two bases and for 2-nitrophenol due to the very high $pK_a$ of 7.2 (see Table S1)).**





Examples in Fig. 2 illustrate that acidity, along with the LWC, strongly influences the phase partitioning of weak acids and
bases into the aqueous phase. The partitioning into the aqueous phase is efficient for pH values well above the individual $pK_a$
values of each acidic compound as long as the LWC does not limit the uptake. High LWCs (0.1-1 g m$^{-3}$) typically associated
with cloud/fog conditions and, accordingly, less acidic media (pH > 4), favor phase partitioning towards the aqueous phase for
most of the weak acids as well as for ammonia. Less water-soluble acids (i.e., with lower $H$ values), such as dissolved SO$_2$
and HONO, display fractions above 0.1 only under less acidic conditions for typical cloud LWC values. Thus, even at colder
cloud temperatures than the 298 K used in Fig. 2, where $H_A^*$ is larger, SO$_2$ and HONO largely remain in the gas phase under
typical cloud acidity conditions. Hence, note that $X_{A_{aq}}$ values of SO$_2$ are typically in the range of 0.005 to 0.5 depending on
both the cloud acidity and temperature. Under typical aerosol conditions, the LWC restricts uptake and only very small
fractions of the less water-soluble acids can partition in the aqueous particle phase. Moreover, very weak acids, with pK$_a$ values
larger than 7 (e.g., 2-nitrophenol) show almost no acidity dependency in the plotted acidic range. On the other hand, for
stronger acids the LWC and acidity impact is even lower due to their lower and/or multiple $pK_a$ values. For example, phthalic
acid partitions in substantial amounts into the aqueous phase for a large range of acidity and LWC conditions. The implication
is that only very water-soluble and strong acids are expected to remain in acidic aerosol solutions. However, it is worth
mentioning again that this treatment neglects several other factors/processes affecting the partitioning of acids in the aqueous
phase, particularly under concentrated aqueous-aerosol conditions. Specifically, volatile acids (e.g., formic and acetic) often
show substantial deviations from this theory (see Nah et al. (2018)) for instance because of the formation of organic salts which
can increase their particle partitioning by two orders of magnitude (Meng et al., 2007). In practice, weak acid anions are often
measured in non-negligible fractions in the particle phase (Tanner and Law, 2003; Limbeck et al., 2005; van Pinxteren and
Herrmann, 2007; Bao et al., 2012; Nah et al., 2018; Teich et al., 2019).

*Non-ideal solutions*

At less than 100% relative humidity, aqueous aerosol solutions exist as a highly concentrated, complex mixture of electrolytes.
Interionic and ion-molecule interactions are critically important under those conditions, leading to thermodynamically non-
ideal behavior (Pitzer, 1991; Zaveri, 2005; Cappa et al., 2008; Zuend et al., 2008; Herrmann et al., 2015; Rusumdar et al.,
2016; Rusumdar et al., 2020). Therefore, parameters that have been developed for dilute aqueous solutions do not strictly apply
to aerosol-phase chemistry.

Nevertheless, several such principles, such as Henry's Law, have been shown experimentally to hold for the aqueous aerosol
phase (Kroll et al., 2005; Sumner et al., 2014), although it may be necessary to account for phenomena such as salting effects
(Kampf et al., 2013; Waxman et al., 2015). Factors such as ionic strength, the different chemical composition of the
concentrated solution, other favored chemical pathways, shifted chemical equilibria (salting-in/out, hydration, metal
complexes, dimer/polymer etc.) and more can significantly affect overall phase partitioning and reaction rates. The inclusion
of these factors into the calculation of the effective Henry's Law constant can explain increased or decreased aqueous-phase



partitioning of chemical compounds such as atmospheric carbonyl compounds (Kampf et al., 2013; Waxman et al., 2015) and organic monocarboxylic acids (Limbeck et al., 2005; Meng et al., 2007) compared to what may be expected based on aqueous solubility alone. Ionic strength effects are also believed to be critically important for acidity-producing in-particle chemical

reactions such as S(IV) oxidation (Martin and Hill, 1987a; Lagrange et al., 1993; Lagrange et al., 1994; Maaß et al., 1999; Ali et al., 2014; Cheng et al., 2016), although experimental data at the extremely high ionic strengths typical of atmospheric aerosols are limited. The first models treating both non-ideal solution effects as well as their feedbacks on occurring chemical processes in detail, have been developed in the last couple of years enabling advanced investigations, e.g. on the phase partitioning issues (Rusumdar et al., 2016).

## 2.2  Acidity buffering

The response of pH in the atmospheric aqueous phases to a perturbation in acidity can be strongly affected by the presence and ability of weak acids or bases to buffer against that change. A buffer is a mixture of a weak acid and its conjugate base (e.g., formic acid and formate) or a mix of a weak base and its conjugate acid (e.g., ammonia and ammonium). The buffering effect, a resistance to pH change, comes from changes in the equilibrium between concentrations, for example, of a weak acid

and a conjugate base. The Henderson-Hasselbach equation (Eq. 4)

$$pH = pK_a + \log\left(\frac{[A^-]}{[HA]}\right) \tag{4}$$

is used to calculate the pH of a buffer solution based on the acid dissociation constant ($K_a$) and the concentrations of the acid [HA] and its conjugate base [A$^-$]. Ion speciation curves for a wide range of atmospherically relevant weak acids are shown in Fig. 3. The magnitude of the buffering effect is greatest when the solution pH is equal to the $pK_a$ of the weak acid buffer

(intersection points of the speciation curves as shown in Fig. 3). Consider, for example, the case of formic acid ($pK_a$ = 3.75 at 298 K) and formate (see Fig. 3c). If protons are added (e.g., through addition of a strong acid such as sulfuric acid) to a solution containing formate or formic acid, and the solution pH is far above or below the 3.75 $pK_a$ of formic acid, each added proton will directly increase the H$^+$ concentration in solution. When the solution pH, however, is close to the formic acid $pK_a$ (where the concentrations of formic acid and formate are equal), many of the added protons will be consumed in converting formate

to formic acid, thereby slowing the pH decline of the solution. For diprotic acids, buffering occurs at each of the two acid dissociation steps. Carbonate buffering is a relevant example for atmospheric cloud and fog droplets. The $pK_a$ values for carbonic acid and bicarbonate are 6.35 and 10.33 at 298 K. A titration by acid addition beginning at pH 12, therefore, would show strong buffering at pH 10.3 and again at pH 6.4, the latter being much more relevant for atmospheric water. Moreover, in mineral dust and volcanic particles that can bear phosphate minerals such as apatite, dissolved phosphate can act as a buffer.

But, unlike carbonates, the phosphate buffer cannot be not lost owing to volatilization. Nevertheless, the buffering by phosphate in other kind of atmospheric aerosol particles is negligible because of the typically extremely low phosphate concentration.



**Figure 3.** Ion speciation of dissolved (a) $SO_2$, (b) HONO, (c) formic acid, (d) acetic acid, (e) phthalic acid, and (f) 2-nitrophenol as a function of pH.

The buffering capacity ($\beta$), a measure to quantitatively express the resistance of an aqueous solution towards acidity changes, is defined for a monoprotic acid by Eq. 5 (see Urbansky and Schock (2000) for details). The buffering capacity $\beta$ expresses the amount of an acid or base concentration addition ($d[C_{a/b}]$) needed to cause a certain change in pH ($d(pH)$).

$$\beta = \frac{d[C_{a/b}]}{d(pH)} = ln10 \cdot \left( \frac{K_W}{[H]^+} + [H^+] + \sum_i \frac{[C]_i \cdot K_{a,i} \cdot [H^+]}{(K_{a,i} + [H^+])^2} \right) \tag{5}$$





Eq. 5 and the plotted examples in Fig. 4 reveal that very high and very low acidity conditions show significantly increased buffering capacities. Moreover, added buffers in the solution lead to local maxima of $\beta$ between very acidic and very alkaline conditions. In the case of one monoprotic acid present in an aqueous solution, the maximum of the buffering capacity occurs close to the $pK_a$ value of the acid, as mentioned above (within approximately $\pm$ 1 pH unit). Furthermore, Eq. 5 shows that buffering capacity, i.e., the amplitude of the local maxima, depends on the concentration of the buffer compound. This agrees with findings in the field, e.g. in fog samples analyzed by Collett et al. (1999) (see discussion below). Furthermore, this

dependency implies a rather high buffering capacity in regions with high multiphase concentrations of weak inorganic and organic acids and bases or high amounts of particulate buffers such as carbonate components. However, the latter are most important in buffering the acidity of supermicron particles or fog/cloud droplets that activate on them.

**Figure 4.** **Buffering capacity $\beta$ of water, ammonia/ammonium and carbonate/bicarbonate/carbonic acid (top: from left to right) as well as formic and acetic acid (bottom: from left to right) as a function of pH. The atmospherically relevant range of cloud and aerosol pH is marked in yellow.**



Titrations of actual cloud and fog samples have exhibited buffering across a wide pH range, suggesting the importance of pH buffering by a variety of compounds with different $pK_a$ values. For example, Collett et al. (1999) report titrations of fog samples

collected at urban and rural locations in California's San Joaquin Valley. Observed buffering in rural fogs in the study could be nearly accounted for based on ammonia and bicarbonate concentrations present in the fog samples. By contrast, significant additional buffering ($\beta$ up to $10^{-4}$ mol $L^{-1}$) was observed in urban fogs over a broad pH range from 4 to 7. The amount of additional buffering was strongly correlated with concentrations of organic compounds in fogs from these environments, with relevant organic buffering agents likely including carboxylic and dicarboxylic acids and phenols.

The buffering phenomenon described above is often referred to as "internal buffering," since it derives from shifts in equilibrium concentrations of compounds present in solution. The exchange of material with the gas phase can also lead to "external buffering." Perhaps the most important form of external buffering is the uptake of additional ammonia from the gas phase in response to a drop in solution pH, as outlined by Liljestrand (1985) and Jacob et al. (1986). Corresponding buffering in atmospheric aerosols from semi-volatile partitioning also occurs, as shown by Meng et al. (2007), Weber et al. (2016) and

Song et al. (2019a) as well as recently by Zheng et al. (2020). However, it should be noted that the effect of aerosol pH buffering from semi-volatile gases on relevant chemical processes has not been studied comprehensively and still represents an issue for future research.

One important consequence of pH buffering in fog and cloud drops is an effect on rates of pH-sensitive aqueous reactions. The presence of (internal and/or external) acid buffering in cloud and fog droplets can slow droplet acidification and maintain

greater rates of reaction for strongly pH-dependent aqueous chemical pathways (e.g., the oxidation of S(IV) by ozone) which are favored by high pH.

## 3   Sources of acidity and alkalinity

Acidic and alkaline components of tropospheric aerosols result from primary gas and aerosol particle emissions as well as secondary gas-phase and aqueous-phase formation processes (Pöschl, 2005b; Seinfeld and Pandis, 2006; Seinfeld, 2015; Zhang

et al., 2015). The most important acidic chemical components of aerosols and cloud/fog droplets are sulfuric acid, nitric acid, nitrous acid, and hydrochloric acid, as well as organic mono- and dicarboxylic acids (e.g., formic acid, acetic acid, oxalic acid etc.) (Vet et al., 2014; Zhang et al., 2015). Then, the most important basic components of aerosols and cloud/fog droplets are ammonium, amines and alkali/alkaline earth metals (U.S.EPA, 2000; Vet et al., 2014; Zhang et al., 2015). The global contribution of different acid and base ions to precipitation has been assessed by Vet et al. (2014). As precipitation samples

provide both compositional and acidity information for the vertical column, these data represent a useful means to point out the spatial sources and sinks of gas and aqueous-phase acidity and alkalinity components.

Gaseous acids can be directly emitted into the troposphere from primary sources such as biomass combustion, traffic (fuel combustion), domestic heating, industrial burning, agriculture, soil, and vegetation (Chebbi and Carlier, 1996; Paulot et al., 2011; Spataro and Ianniello, 2014; Kawamura and Bikkina, 2016). Moreover, gaseous acids can be formed secondarily by





gas-phase oxidations of emitted acid precursor compounds such as $SO_2$, $NO_x$, and VOCs (Chebbi and Carlier, 1996; Paulot et al., 2011; Spataro and Ianniello, 2014; von Schneidemesser et al., 2015; Zhang et al., 2015; Kawamura and Bikkina, 2016). The gas-phase OH oxidation of $SO_2$ is an important source of gaseous sulfuric acid and, after condensation, of particulate sulfate (von Schneidemesser et al., 2015; Zhang et al., 2015). $SO_2$ is emitted from anthropogenic activities such as the combustion of sulfur-containing fuels and various natural sources such as volcanos (Smith et al., 2001; Smith et al., 2011;

Seinfeld, 2015). Moreover, it is formed from the oxidation of natural precursors such as dimethyl sulfide (DMS, $CH_3SCH_3$) emitted by oceanic phytoplankton (Seinfeld and Pandis, 2006). The gaseous oxidation pathway of $SO_2$ contributes also to newly formed particles (nucleation, (Zhang et al., 2015)) which are expected to be quite acidic. Furthermore, the gaseous oxidation of $NO_x$ and VOCs can lead to the formation of nitric/nitrous acid and organic acids (e.g., formic, acetic and oxalic acid) (Chebbi and Carlier, 1996; Paulot et al., 2011; Spataro and Ianniello, 2014; Zhang et al., 2015; Kawamura and Bikkina,

2016). By contrast, gaseous bases such as ammonia and amines are almost exclusively emitted into the troposphere, mainly from agriculture due to intensive stock farming and the use of $NH_3$-based fertilizer applications. Moreover, bases are released from biomass burning, vehicles, industrial processes, and as a consequence of volatilization from soils and oceans. (U.S.EPA, 2000; Behera et al., 2013). As shown in Fig. 5, subsequent to their emission or secondary formation, gaseous acids and bases can condense on existing aerosol particles or fog/cloud droplets and can then contribute to aerosol acidity.






**Figure 5.** Schematic of sources (red text) and conditions of acidity in different aqueous aerosol particles (green text) together with microphysical and chemical processes that are able to influence the acidity of tropospheric aerosols (Fig. created after Raes et al. (2000) and McMurry (2015)) The dashed gray line represents an aerosol number size distribution.

Acidic and alkaline aerosol components are also (i) primarily emitted by anthropogenic and natural processes (Zhang et al., 2015) or secondarily formed in aqueous aerosol solutions or at their interface (see Sect. 4 and 5 below). Important anthropogenic primary sources of acidic and alkaline aerosols (see Fig. 5) are urban combustion aerosols and agricultural aerosols, including e.g. agricultural ammonia from livestock farming. Important natural primary sources of acidic and alkaline aerosols are sea spray, desert dust, biomass burning and volcanic emissions. Besides the secondary acid formation in the gas-

phase, in-cloud oxidation of $SO_2$ contributes more than 50% globally to sulfate aerosol mass formation (Alexander et al., 2009) (see Sect. 4.1 for details). Thus, the aqueous-phase formation of sulfate from the oxidation of $SO_2$ is the largest source of acidity in the atmosphere. However, besides sulfate, other acidic components are also secondarily formed in aqueous aerosols



such as nitrate, chloride, formate, acetate, and oxalate (see (Chebbi and Carlier, 1996; Spataro and Ianniello, 2014; Ervens, 2015; Zhang et al., 2015; Kawamura and Bikkina, 2016)).

In the past, emissions of SO₂ in industrialized countries were the predominant cause of strong acidification of aerosol particles, cloud droplets and precipitation, typically known as the acid rain phenomenon (Adriano and Johnson, 1989; Seinfeld and Pandis, 2006). However, due to strongly reduced anthropogenic sulfur emissions in some parts of the world, a reduction in cloud/fog acidity has been observed over recent decades (see Pye et al. (2020)). As a consequence of the changing acid and base sources, the composition of continental aerosol particles and cloud/fog/rain droplets will most likely continue to evolve

toward compositions observed pre-industrially in rural continental areas, e.g. in North America and Western Europe. These environments are characterized by higher contribution of organic acids and chloride (less acid displacement) and less sulfate/nitrate mass (see precipitation composition data compiled by Vet et al. (2014)). In such a future environment, natural acidity sources become a much more important source for the acidity of tropospheric cloud/fog/rain droplets. On the other hand, no significant changes are expected for the acidity of marine droplets, except downwind of continents. Their main acidity

and alkalinity sources, such as the emission of DMS, marine NH₃ and sea salt particles containing chloride and base cations, are not expected to change significantly. However, it should be mentioned that the impact of climate change including higher temperatures and ocean acidification and related changes in the ocean biochemistry, may unequally affect the emission of DMS in different regions. The effects of climate change on DMS emission patterns are still under debate due to the complex interactions of marine biochemistry and atmosphere-ocean interactions (Six et al., 2013; Gypens and Borges, 2014; Dani and

Loreto, 2017; Hopkins et al., 2020).

## 4    Interactions of acidity and chemical processes: Inorganic systems

In this section, the feedbacks between particle/droplet acidity and key inorganic chemical subsystems, the sulfur(IV) oxidation and tropospheric halogen chemistry are discussed in detail.

### 4.1    Acidity and Sulfur Oxidation

In addition to its reaction with OH in the gas phase, SO₂ is oxidized via heterogeneous and multiphase reactions in clouds, fog, or aerosol particles to form particulate sulfate. Sulfate is a major component of PM₂.₅, especially in areas affected by emissions from burning coal or other sulfur-containing fossil fuels (Attwood et al., 2014). Because sulfate lifetime is on the order of days (Barth et al., 2000), sulfate contributes to regional haze and acid deposition, as well as local air pollution.

Once in the aqueous phase, SO₂ is hydrated and undergoes acid-base equilibrium to form other S(IV) species, bisulfite (HSO₃⁻)

(pK$_{a,R-1}$ = 1.9) and sulfite (SO₃²⁻) (pK$_{a,R-2}$ = 7.2). The hydration of SO₂ upon uptake alone, according to (R-1) already leads to the release of acidity:

$$SO_2 \cdot H_2O \overset{K_{a1}}{\rightleftharpoons} HSO_3^- + H^+ \tag{R-1}$$





$$HSO_3^- \overset{K_{a2}}{\leftrightharpoons} SO_3^{2-} + H^+ \qquad \text{(R-2)}$$

S(IV) oxidation in the aqueous phase to form S(VI) species (sulfate ($SO_4^{2-}$), bisulfate ($HSO_4^-$), and sulfuric acid ($H_2SO_4$)) leads

to further acidification. S(IV) oxidation can take place via a number of chemical pathways, many of which are pH-sensitive (Fig. 6). As a result of the equilibrium reactions described by R-1 and R-2, the effective solubility of $SO_2$ in aqueous solutions increases rapidly with increasing pH (see Eq. 2a). Partly for this reason, as well as because of their relatively small liquid water content ($\sim 10^{-9}$ cm$^3$ cm$^{-3}$), sulfate formation in aerosols is generally believed to be less significant than in clouds and fog (Schwartz, 1986). Only S(VI) formation in the gas-phase and in clouds is included in most large-scale atmospheric chemistry

models. Globally, in-cloud formation is thought to be the dominant sulfate production pathway ($\sim$60%), particularly over the oceans (generally >75%) (Barth et al., 2000; Barrie et al., 2001; Manktelow et al., 2007; Alexander et al., 2009; Faloona, 2009; Alexander et al., 2012). However, there is evidence that significant sulfate formation also occurs in polluted urban areas, during periods of high aerosol surface area and few clouds (Hering and Friedlander, 1982; Wang et al., 2014; He et al., 2018a). This suggests that aerosol chemistry is also an important source of sulfate under some conditions.

In the aqueous phase, S(VI) species exist in acid-base equilibrium according to:

$$H_2SO_4 \overset{K_{a3}}{\leftrightharpoons} HSO_4^- + H^+ \qquad \text{(R-3)}$$

$$HSO_4^- \overset{K_{a4}}{\leftrightharpoons} SO_4^{2-} + H^+ \qquad \text{(R-4)}$$

Since sulfuric acid is a very strong acid ($K_{a,R-4} \cong 1000$ mol L$^{-1}$ at 298 K (Graedel and Weschler, 1981)), almost no unionized $H_2SO_4$ exists in aqueous solution and $HSO_4^-$ is significant only at pH < 3. As a consequence, the conversion of S(IV) to S(VI)

in the aqueous phase increases the acidity of the cloud or aerosol particle not only by the initial acidification through the $SO_2$ reaction with water, but additionally through the dissociation of sulfuric acid. Some S(IV) oxidation reactions have other acidic byproducts such as halous acid species HX (with X = Cl, Br) or HONO, and thus may contribute additional acidity to the aerosol (Fig. 6). Figure 6 illustrates that sulfate oxidation under urban haze conditions can significantly contribute to the acidification of aerosols. After a short period of chemical processing, aerosols are expected to reach pH 4.5 or lower.

Particularly for haze particles with initial pH conditions above 4, a fast acidification can be modeled as a consequence of the higher initial S(IV) oxidation rates under less acidic conditions.

In the next subsections, we outline the major S(IV) oxidation pathways, their sensitivity to the pH of the aqueous medium, and their potential to alter pH via the formation of acid products.

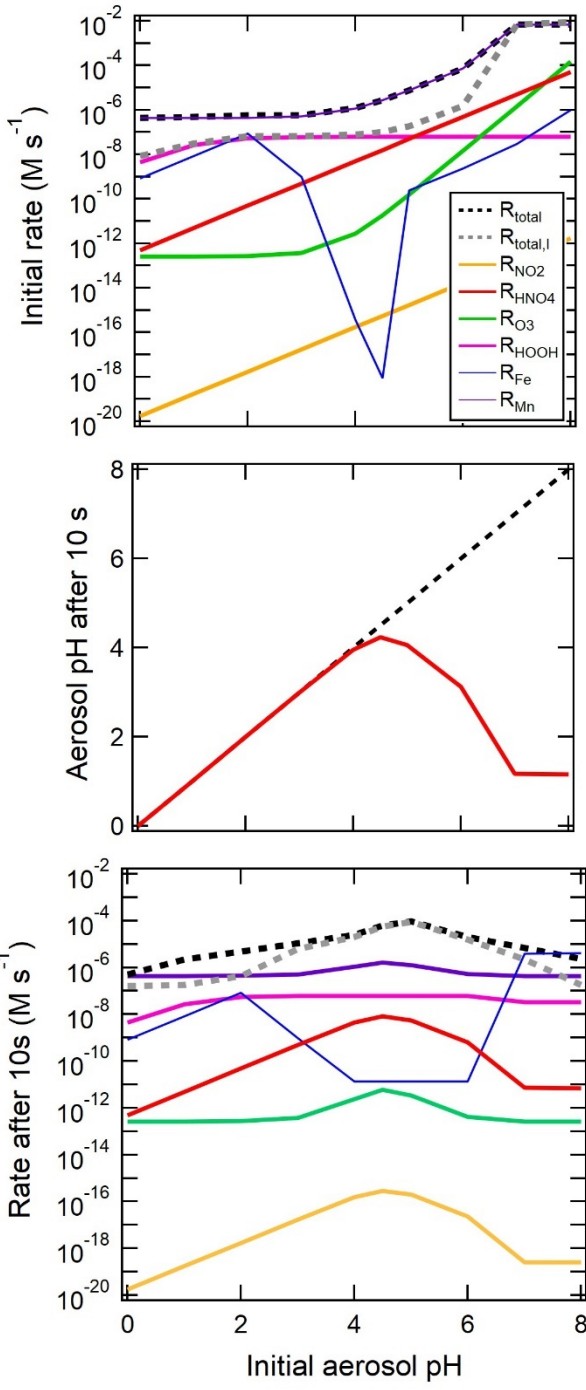

**Figure 6. S(IV) oxidation rates for Beijing winter haze conditions (following (Cheng et al., 2016)). Shown are initial rates (top), aerosol pH after 10 s of reaction at the initial rates (center), and S(IV) oxidation rates after 10s of reaction (bottom). Total S(IV) rate with and without taking into account ionic strength at the maximum reported limit is shown. Rates used were those recommended in this text.**



## 4.2  S(IV) oxidation through $O_3$, $H_2O_2$, ROOH, and HOX (with X = Cl, Br, I)

Due to the pH-dependent partitioning of S(IV) species and hence solubility of $SO_2$, most S(IV) oxidation mechanisms are highly pH-dependent. However, S(IV) oxidation by $H_2O_2$, is only weakly pH dependent. At pH values typical of cloud water (pH = 2 - 7 (Pye et al., 2020)), S(IV) oxidation by $H_2O_2$ is thought to dominate sulfate production (Faloona, 2009) although other oxidants can be important at higher pH values or if $H_2O_2$ is depleted (e.g., Shen et al. (2012)). In-cloud S(IV) oxidation by $H_2O_2$ proceeds via reaction with $HSO_3^-$, followed by addition of $H^+$ (R-5/6a).

$$HSO_3^- + H_2O_2 \rightleftharpoons SO_2OOH^- + H_2O \qquad \text{(R-5)}$$

$$SO_2OOH^- + H^+ \rightarrow 2\,H^+ + SO_4^{2-} \qquad \text{(R-6a)}$$

$$SO_2OOH^- + HX \rightarrow 2\,H^+ + SO_4^{2-} + X^- \qquad \text{(R-6b)}$$

Therefore, the intrinsic reaction rate decreases rapidly with increasing pH above pH 2 (McArdle and Hoffmann, 1983). This is balanced by the fact that the effective $SO_2$ solubility increases with increasing pH. As a result, the overall rate is relatively independent of pH, above pH ~ 1.5. The rate expression for S(VI) formation by S(IV) + $H_2O_2$ is given by McArdle and Hoffmann (1983); Lind et al. (1987); and Gunz and Hoffmann (1990):

$$R_{H_2O_2} = k_{6a} \frac{[H^+][HSO_3^-]}{1+K_5[H^+]} [H_2O_2] \qquad \text{(6a)}$$

with a recommended temperature-dependent rate constant $k_{6a} = 7.45 \times 10^7 \exp\left(-4000\left(\frac{1}{T} - \frac{1}{298}\right)\right)$ L mol$^{-1}$ s$^{-1}$ and $K_5 = 13$ mol L$^{-1}$.

Recently, Liu et al. (2020) investigated S(VI) formation by S(IV) + $H_2O_2$ in a flow reactor under aqueous aerosol conditions (pH = 2.5, high ionic strength, and 73-90% relative humidity) and in the presence of malonic acid. This study revealed that, under concentrated aqueous-aerosol conditions, the S(VI) formation rate can be significantly increased compared to dilute aqueous conditions like those in clouds. The study demonstrated that ionic strength and general acid catalysis promotes faster S(VI) formation via R-6b. This additional pathway is expected to contribute to S(VI) missing from model simulations of severe haze episodes (Hering and Friedlander (1982); Wang et al. (2014); He et al. (2018a)).

The rate expression given by Liu et al. (2020) is as follows:

$$R_{H_2O_2} = (k + k_{HX}[HX][H^+]^{-1})K_{a1}H_{SO_2}p_{SO_2}H_{H_2O_2}p_{H_2O_2} \quad (pH > 2) \qquad \text{(6b)}$$





with the following ionic strength dependencies of the reaction rate constant, Henry's law constants and dissociation constants (see Liu et al. (2020) and references therein).

k: $\qquad \log\left(\frac{k}{k^{I=0}}\right) = 0.36 \cdot I - \frac{1.018\sqrt{I}}{1+1.018\sqrt{I}}$ $\qquad$ ($I_{max} = 5$ molal) $\qquad$ (6c)

$H_{H_2O_2}$: $\qquad \frac{H_{H_2O_2}}{H_{H_2O_2}^{I=0}} = 1 - 1.414 \cdot 10^{-3} \cdot I^2 + 0.121 \cdot I$ $\qquad$ ($I_{max} = 5$ molal) $\qquad$ (6d)

$H_{SO_2}$: $\qquad \frac{H_{SO_2}}{H_{SO_2}^{I=0}} = (\frac{22.3}{T} - 0.0997) \cdot I$ $\qquad$ ($I_{max} = 6$ molal) $\qquad$ (6e)

$K_{a1}^*$: $\qquad \log\left(\frac{K_{a1}^*}{K_{a1}^{I=0}}\right) = 0.5 \cdot \sqrt{I} - 0.31 \cdot I$ $\qquad$ ($I_{max} = 6$ molal) $\qquad$ (6f)

and

$K_{a2}^*$: $\qquad \log\left(\frac{K_{a2}^*}{K_{a2}^{I=0}}\right) = 0.5 \cdot \sqrt{I} - 0.36 \cdot I$ $\qquad$ ($I_{max} = 6$ molal) $\qquad$ (6g)

In Eq. 6b, $k = k' \cdot 1.3 \times 10^{-2} \cdot e^{1960\left(\frac{1}{T}-\frac{1}{298}\right)} \cdot 6.6 \times 10^{-8} \cdot e^{1500\left(\frac{1}{T}-\frac{1}{298}\right)}$ (reaction rate constant of proton-catalyzed pathway R-6a), $k_{HX}$ (overall reaction rate constant of the catalysis pathway of a general acid HX (R-6b), $k_{malonic\ acid} = 5.61 \times 10^5\ mol^2\ kg^{-2}\ s^{-1}\ (at\ I = 3.9\ mol\ kg^{-1})$ , $k_{malonate} = 1.32 \times 10^5\ mol^2\ kg^{-2}\ s^{-1}\ (at\ I = 6.6\ mol\ kg^{-1})$ ), $K_{a1} = 1.3 \times 10^{-2} \cdot e^{1960\left(\frac{1}{T}-\frac{1}{298}\right)}$ (thermodynamic dissociation constant of R-5), $H_{SO_2} = 1.23 \cdot e^{3145.3\left(\frac{1}{T}-\frac{1}{298}\right)}$ (Henry's law constant of SO₂) and $H_{H_2O_2} = 1.3 \times 10^5 \cdot e^{7297.1\left(\frac{1}{T}-\frac{1}{298}\right)}$ (Henry's law constant of H₂O₂). Furthermore, $p_{SO_2}$ and $p_{H_2O_2}$ represent partial pressure of SO₂ and H₂O₂ in the gas phase, respectively. Note, the kinetic of the study by Liu et al. (2020) has been determined for NaCl/NaNO₃-malonate/malonic acid mixtures only which could restrict their applicability. Hence, further investigations for other aerosol composition mixtures (e.g., considering ammonium-sulfate salts and other general acids), lower pH conditions and higher ionic strengths are definitely needed to provide even more advanced rate expressions for concentrated aqueous-aerosol conditions.

Organic hydroperoxides (ROOH) can also oxidize HSO₃⁻ by a similar mechanism to H₂O₂, although at lower rates (Graedel and Goldberg, 1983; Lind et al., 1987; Drexler et al., 1991). The oxidation of HSO₃⁻ by methylhydroperoxide, CH₃OOH, has methanol as a product, with the overall reaction given as (Lind et al., 1987):

$HSO_3^- + CH_3OOH + H^+ \rightarrow SO_4^{2-} + 2\ H^+ + CH_3OH$ $\qquad$ (R-7)

with a third-order rate law:

$R_{CH_3OOH} = k_7[HSO_3^-][CH_3OOH][H^+]$ $\qquad$ (7)





with $k_7 = 1.7 \times 10^7 \exp\left(-3800\left(\frac{1}{T} - \frac{1}{298}\right)\right)$ $L^2\ mol^{-2}\ s^{-1}$.

The S(IV) oxidation rate for peroxyacetic acid is faster (Lind et al., 1987), and produces acetic acid as a byproduct, thereby further increasing the acidity of the aqueous phase:

$HSO_3^- + CH_3C(O)OOH + H^+ \rightarrow SO_4^{2-} + 2\ H^+ + CH_3COOH$ (R-8)

with a third-order rate law:

$$R_{CH_3C(O)OOH} = k_8[HSO_3^-][CH_3C(O)OOH][H^+]$$ (8)

with $k_8 = 5.6 \times 10^7 \exp\left(-3990\left(\frac{1}{T} - \frac{1}{298}\right)\right)$ $L^2\ mol^{-2}\ s^{-1}$.

The aerosol- and gas-phase abundances of organic hydroperoxides are poorly constrained, so S(IV) oxidation by ROOH may be more important than previously thought in aerosols containing secondary organic material (Ye et al., 2018b; Dovrou et al.,

2019; Wang et al., 2019a). Organosulfates have been proposed as minor products of the S(IV) + ROOH reactions with secondary organic material, with further implications for aerosol pH (Wang et al., 2019a).

In contrast to S(IV) oxidation by $H_2O_2$, the oxidation of S(IV) by reaction with $O_3$ becomes faster with increasing pH. Since S(VI) formation contributes to acidification of the aerosol, these processes are therefore potentially self-limiting, depending

on the buffering capacity of the aqueous medium (Fig. 6).

$SO_2 \cdot H_2O + O_3 \rightarrow HSO_4^- + O_2 + H^+$ (R-9a)

$HSO_3^- + O_3 \rightarrow HSO_4^- + O_2$ (R-9b)

$SO_3^{2-} + O_3 \rightarrow SO_4^{2-} + O_2$ (R-9c)

Each S(IV) species reacts with $O_3$, leading to a composite rate expression of:

$R_{O_3} = \left(k_{9a}[SO_3^{-2}] + k_{9b}[HSO_3^-] + k_{9c}[SO_2 \cdot H_2O]\right)(1 + F_iI)[O_3]$ (9)

Here, $F_i$ is an empirically determined factor accounting for the effect of ionic strength, $I$, on the rate. Lagrange et al. (1994) explored the effects of ionic strength on the oxidation of S(IV) by $O_3$ (up to 4 mol $L^{-1}$) and found that $F = 1.59 \pm 0.3$ for NaCl and $F = 3.71 \pm 0.7$ for $Na_2SO_4$. The rate constant for oxidation of $SO_3^{2-}$ by $O_3$ ($k_{9a} = 1.5 \times 10^9 \exp\left(-5280\left(\frac{1}{T} - \frac{1}{298}\right)\right)$ L $mol^{-1}\ s^{-1}$) is over three orders of magnitude larger than the rate constant for $O_3 + HSO_3^-$ ($k_{9b} = 3.7 \times$

$10^5 \exp\left(-5530\left(\frac{1}{T} - \frac{1}{298}\right)\right)$ L $mol^{-1}\ s^{-1}$) (Hoffmann and Calvert, 1985), which is more than ten times the rate constant for the reaction of $O_3$ with $SO_2 \cdot H_2O$ ($k_{9c} = 2.4 \times 10^4$ $L^2\ mol^{-2}\ s^{-1}$) when the respective maximum values are compared (Hoffman, 1986). Therefore, the overall rate of S(IV) oxidation by $O_3$ increases rapidly with increasing pH, and is most important above pH 5-6 (Chameides, 1984; Calvert et al., 1985; Turnock et al., 2019).

Sulfate can also form via reaction of S(IV) with $O_3$ on the surface of alkaline aerosols, e.g. freshly emitted sea salt aerosols

and some mineral dust aerosols (Sievering et al., 1992; Chameides and Stelson, 1993; Zhang and Carmichael, 1999; Li et al., 2006; Wu et al., 2011; Yu et al., 2017; Zhang et al., 2018b). At pH values typical of fresh sea salt aerosol (pH ≈ 8), the S(IV)





loss rate constant for oxidation by $O_3$ in these aerosols is $10^5$ times larger than in-cloud oxidation by $H_2O_2$, more than making up for their lower liquid water content (Sievering et al., 1992; Chameides and Stelson, 1993). However, like other S(IV) + $O_3$ mechanisms, these processes are potentially self-limiting, as noted above.

Reactions of S(IV) with hypohalous acids (HOBr, HOCl, and HOI; see reactions R-17/ R-18) contribute to sulfate formation in the marine boundary layer (Vogt et al., 1996; von Glasow et al., 2002a; Chen et al., 2016). These reactions act as a sink for reactive halogens by converting them to their acidic form (e.g., HOBr → HBr, see Sect. 4.8 for further details) (Chen et al., 2016).

### 4.3 Free radical pathways for S(IV) oxidation

The hydroxyl radical (OH) can oxidize S(IV) in the aqueous phase through a radical pathway involving $SO_3^-$, $SO_5^-$, $HSO_5^-$, and $SO_4^-$. This process is more likely to be important in cloud water than in aqueous aerosol due to the higher liquid water content of clouds and the relatively lower OH concentration in aqueous aerosols (Herrmann et al., 2010; McNeill, 2015). The high concentrations of organic material in aerosols can quench of radical and triplet species (Herrmann et al., 2010; McNeill, 2015; Wang et al., 2020). Furthermore, the reaction of OH with $SO_3^{2-}$ is somewhat faster than that of OH with $HSO_3^-$

($k = 4.6 \times 10^9$ L mol$^{-1}$ s$^{-1}$ vs. $2.7 \times 10^9$ L mol$^{-1}$ s$^{-1}$) (Buxton et al., 1996). This, along with the pH dependence of the water solubility of $SO_2$, suggests that S(IV) oxidation by OH is more efficient at higher pH and in clouds (and is potentially self-limiting). The production of $SO_4^-$ via this reaction pathway couples S(IV) oxidation to organosulfate production (Perri et al., 2010), although this is a minor pathway (McNeill et al., 2012).

Laboratory studies have demonstrated sulfate production on the surface of acidic aerosols, via direct electron transfer from

$HSO_3^-$ to $O_2$, followed by a free-radical chain oxidation of bisulfite to sulfate (Hung and Hoffmann, 2015), however the significance of this pathway is not confirmed by field and modeling studies (Shao et al., 2019). Catalytic oxidation of S(IV) by $NO_3$ (Exner et al., 1992; Rudich et al., 1998; Feingold et al., 2002), also believed to take place via a free radical mechanism, may be important in the remote troposphere. Recent experimental studies suggest that photolysis of particulate nitrate and hydrolysis of $NO_2$ to form nitrate and HONO (Li et al., 2018) may accelerate oxidation of S(IV) under Beijing conditions by

generating $NO_2$ and OH radicals (Gen et al., 2019). However, the consumption of OH radicals by organic constituents present in aerosols were ignored in this study likely leading to an overestimation of the effect.

Another suggested S(IV) oxidation pathway is the reaction of excited triplet states of photosensitizers (PS*) with S(IV) species (see R-10). This pathway potentially involves produced sulfur-containing radicals and/or excited transient species (see e.g., Wang et al. (2020) and Loeff et al. (1993)). Currently, it is also discussed as potential S(IV) oxidation pathway under polluted

aerosol conditions (Wang et al., 2020).

$$S(IV) + PS^* \rightarrow S(VI) + products \quad (S(IV) = SO_2 \cdot H_2O + HSO_3^-) \tag{R-10}$$

The exact reaction pathway is still uncertain particularly with respect to the involved sulfur-containing radicals or excited transient species. Some studies (Loeff et al., 1993; Wang et al., 2020) already determined chemical reaction rate constants for





certain PS* species such as acetophenone, flavone, xanthone, 4-(benzoyl)benzoic acid and anthraquinone-1-sulfonate. The

second-order reaction rate constants of PS* with S(IV) species measured in the laboratory are between $6.0 \times 10^7$ and $1.0 \times 10^9$ mol L$^{-1}$ s$^{-1}$. Kinetic measurements of the reactive PS* quenching by S(IV) using of ambient filter extracts taken during Chinese winter haze conditions revealed a rate constant of $1.3 \times 10^8$ mol L$^{-1}$ s$^{-1}$ (Wang et al. (2020)). Note, the kinetic investigations of Wang et al. (2020) assumed that the initial reaction step is the rate-limiting step in this reaction sequence and the reaction rate constant is pH-independent. So, based on Wang et al. (2020), the rate expression is as follows:

$$R_{PS^*} = k_{10}[PS^*][S(IV)] \tag{10}$$

Due to the presently strong uncertainties in the existing kinetic data and mechanistic understanding of R-10, a recommendation of a proper kinetic reaction rate constant is rather difficult. Thus, we preliminary recommend the chemical rate constant of $k_{10} = 1.3 \times 10^8$ mol L$^{-1}$ s$^{-1}$. Finally, it should be noted that great care is needed for estimating the rate of R-10 because of lacking knowledge about the present PS* concentrations in ambient aerosols and cloud droplets and the very rapid quenching and

deactivation triplet species by water, dissolved oxygen and organic/inorganic aerosol constituents which might lead to very low PS* concentrations which can strongly limit or inhibit this pathway similarly to the S(IV) oxidation by free radical which can be effectively scavenged by particle constituents other than S(IV) as outline above in the present section.

### 4.4 S(IV) oxidation catalyzed by transition metal ions

The oxidation of S(IV) by O$_2$ as catalyzed by transition metal ions (TMI, mainly Fe(III) and Mn(II); see R-11/R-12)

(Humphreys, 1964; Martin and Hill, 1987a; Martin and Hill, 1987b; Brandt and van Eldik, 1995; Alexander et al., 2009; Harris et al., 2013) is an efficient pathway for S(VI) formation, especially under conditions where photochemistry is limited, e.g. wintertime at high latitudes (Simpson et al., 2019).

$$S(IV) + \tfrac{1}{2}\,O_2 \xrightarrow{\text{Fe(III)}} S(VI) \tag{R-11}$$

$$S(IV) + \tfrac{1}{2}\,O_2 \xrightarrow{\text{Mn(II)}} S(VI) \tag{R-12}$$

The solubility and speciation of the TMI (Pye et al., 2020), as well as the reaction rates, all depend on pH. As primary pollutants, TMI concentrations are higher in aerosols than in cloud water, but this effect is limited by the pH-dependent solubility of the active species. The TMI-S(IV) reactions (R-11/R-12) are also reported to be inhibited by ionic strength (Martin and Hill, 1987a; Martin and Hill, 1987b), although this dependence is only known under the relatively dilute conditions which are accessible in bulk solutions. This introduces considerable additional uncertainty to estimates of the aerosol-phase TMI

catalyzed S(IV) oxidation rate.

TMI-mediated S(IV) oxidation has been proposed to proceed through radical intermediates (Grgić and Berčič, 2001), at least for pH > 3.6 (Martin et al., 1991). A detailed discussion of the mechanisms can be found in Brandt and van Eldik (1995). A pH-dependent synergistic effect has been reported when multiple transition metal ions are present in solution (Ibusuki and Takeuchi, 1987; Martin and Good, 1991; Harris et al., 2013). Martin et al. (1991) observed that water-soluble organic material

inhibits Fe(III)-catalyzed S(IV) oxidation for pH ≥ 5. Given this pH range, the effect is not expected to be significant for





atmospheric aerosols, although interactions with organics, for example complexation with oxalate, may impact TMI chemistry in other ways (e.g., Okochi and Brimblecombe (2002); Passananti et al. (2016)).

Given the current focus on sulfate formation in atmospheric aerosols, our recommendations for kinetics of S(IV) oxidation by TMI favor studies which included the ionic strength and pH effects. For Fe(III)-catalyzed S(IV) oxidation, the expression from

Martin and Hill (1987b) and Martin et al. (1991) is as follows:

$$R_{Fe,10a} = \begin{cases} \dfrac{k_{11a}[Fe(III)][S(IV)]10^{-2\sqrt{I}/(1+\sqrt{I})}}{[H^+](1+K_{11}[S(VI)]^{2/3})} & \text{for pH} < 3.6 \\ k_{11b}[Fe(III)]^2[S(IV)] & \text{for } 3.6 \le \text{pH} \le 5 \\ k_{11c}[S(IV)] & \text{for } 5 < \text{pH} \le 6 \\ k_{11d}[S(IV)] & \text{for pH} > 6 \end{cases} \tag{11a}$$

Here, $k_{11a} = 6\,\text{s}^{-1}$, $K_{11} = 150\,(\text{mol L}^{-1})^{-2/3}$, $k_{11b} = 10^9\,\text{L}^2\,\text{mol}^{-2}\,\text{s}^{-1}$, $k_{11c} = 10^{-3}\,\text{s}^{-1}$ and $k_{11d} = 10^{-4}\,\text{s}^{-1}$. However, the dependence of Eq. 11a on ionic strength (I) is only known up to 1 mol L$^{-1}$ and unfortunately the rate law is valid for a limited range of conditions only ([Fe$^{3+}$] > 10$^{-7}$ mol L$^{-1}$, [S(IV)] < 10$^{-5}$ mol L$^{-1}$, [S(VI)] < 10$^{-4}$ mol L$^{-1}$, I < 10$^{-2}$ mol L$^{-1}$).

Moreover, note that the ionic strength effect was verified at pH = 2 and T = 25 °C only. Additionally, the study implied that the effect of higher S(IV) and S(VI) concentrations may be more important than the ionic strength effect (see Martin et al. (1991) for details). Due to the limited range of conditions where the expression of Martin and Hill (1987b) and (Martin et al., 1991) are valid and existing gaps in the understanding of this reaction, thus, we recommend the rate expression by (Hoffmann and Calvert, 1985).

$$R_{Fe,11b} = k_{10e}[Fe(III)][SO_3^{2-}] \qquad \text{(for pH} < 5) \tag{11b}$$

with $k_{11e} = 1.2 \times 10^6\,\text{L mol}^{-1}\,\text{s}^{-1}$.

The rate for Mn(II)-catalyzed S(IV) oxidation from Martin and Hill (1987a) is recommended:

$$R_{Mn} = \begin{cases} k_{12a}[Mn(II)][S(IV)] & \text{for S(IV)} < 10^{-4}\,\text{mol L}^{-1} \\ k_{12b}[Mn^{2+}]^2 & \text{for S(IV)} > 10^{-4}\,\text{mol L}^{-1} \end{cases} \tag{12}$$

where $k_{12a} = k_{12a,0}10^{-4.07\sqrt{I}/(1+\sqrt{I})}\,\text{L mol}^{-1}\text{s}^{-1}$ and $k_{12b} = k_{12b,0}10^{-4.07\sqrt{I}/(1+\sqrt{I})}\,\text{L mol}^{-1}\text{s}^{-1}$

with $k_{12a,0} = 10^3\,\text{L m}^{-1}\text{s}^{-1}$ and $k_{11b,0} = 680\,\text{L mol}^{-1}\text{s}^{-1}$. Note that, while Martin and Hill (1987b) and Martin and Hill (1987a) observed strong inhibition with increasing ionic strength, $k_{12a}$ is only reported for ionic strength up to 1 mol L$^{-1}$. Overall, TMI -catalyzed reactions are still not very well understood, and further studies of these reactions particularly under aerosol conditions are needed.

A synergistic effect has been reported in laboratory studies when Fe(III) and Mn(II) are both present in solution (Martin, 1984; Ibusuki and Takeuchi, 1987; Martin and Good, 1991; Grgić et al., 1992), but more work must be done to reconcile the rates of R-13 from those studies with single-ion studies, and the effect of ionic strength is not known.

$$S(IV) + \tfrac{1}{2}O_2 \xrightarrow{\text{Fe(III) + Mn(II)}} S(VI) \tag{R-13}$$




The recommended rate of R-13 is from Ibusuki and Takeuchi (1987), who investigated the effect as a function of pH and
temperature:

$$R_{TMI-Syn} = \begin{cases} k_{13a}[H^+]^{-0.74}[Mn(II)][Fe(III)][S(IV)] & \text{for } 2.6 \leq pH \leq 4.2 \\ k_{13b}[H^+]^{0.67}[Mn(II)][Fe(III)][S(IV)] & \text{for } 4.2 < pH \leq 6.5 \end{cases} \tag{13}$$

where $k_{13a} = 3.72\times10^7$ L mol$^{-1}$ s$^{-1}$ and $k_{13b} = 2.51\times10^{13}$ L mol$^{-1}$ s$^{-1}$.

For the sake of completeness, a more comprehensive literature overview on reaction rate constants related to TMI-catalyzed
S(IV) oxidation kinetics is given in Radojevic (1992) and Brandt and van Eldik (1995).

## 4.5 NO$_2$ and HNO$_4$

NO$_2$ can oxidize HSO$_3^-$ in the aqueous phase (Lee and Schwartz, 1983) through adduct formation, followed by decomposition,
to eventually form SO$_3^-$ and the weak acid HONO. The thermodynamic driving force for this process is small (Spindler et al.,
2003). The reaction favors basic conditions and therefore is unlikely to be significant for most atmospheric aerosols, and self-
limiting. Early studies Lee and Schwartz (1983) reported relatively high reaction rates which decreased rapidly with decreasing
pH. Spindler et al. (2003) demonstrated, based on coupled gas- and aqueous phase measurements together with the direct
measurement of NO$_2$ in aqueous solution, that the reaction between NO$_2$ and S(IV) proceeds first by an adduct formation
equilibrium (R-14a and R-14b) followed by the adduct's unimolecular decomposition (R-15a and R-15b) to products
nitrite/HONO and SO$_3^-$.

$$NO_{2(aq)} + SO_{3(aq)}^{2-} \underset{k_{-14a}}{\overset{k_{14a}}{\rightleftharpoons}} [NO_2 - SO_3]^{2-} \tag{R-14a}$$

$$[NO_2 - SO_3]^{2-} \xrightarrow{k15a} NO_2^- + SO_{3(aq)}^- \tag{R-15a}$$

$$NO_{2(aq)} + HSO_{3(aq)}^- \underset{k_{-14b}}{\overset{k_{14b}}{\rightleftharpoons}} [NO_2 - HSO_3]^- \tag{R-14b}$$

$$[NO_2 - HSO_3]^{2-} \xrightarrow{k15b} H^+ + NO_2^- + SO_{3(aq)}^- \tag{R-15b}$$

This mechanism was invoked to explain the formation of 'artifact HONO' in a wet denuder when both NO$_2$ and SO$_2$ are present
in the gas phase. The kinetic data of Spindler et al. (2003) were experimentally determined by measuring NO$_2$ in aqueous
solution in a laser photolysis-broadband optical absorption experimental set-up and newly kinetically analyzed for the present
review. The measurement was performed at pH = 4.5 and pH = 10 to investigate either the HSO$_3^-$ or the fully deprotonated
form $SO_3^{2-}$. From the T-dependent rate constants in Table S2, of the forward ($k_{14a}$, $k_{14b}$) and backward reaction ($k_{-14a}$, $k_{-14b}$) the
equilibrium constants ($K_{14a}$, $K_{14b}$) were calculated and the Arrhenius expressions were derived as follows.





At pH 10.0:

$k_{14a}(T) = (1.4 \pm 0.2)\ 10^7\ L\ mol^{-1}\ s^{-1}$   $(288\ K \leq T \leq 328\ K)$

$k_{-14a}(T) = (3.5 \pm 0.5)\ 10^6\ exp[-(2440 \pm 710)\ K\ /\ T]\ s^{-1}$

$K_{14a}(T) = (1.9 \pm 15)\ exp[-(-2700 \pm 1600)\ K\ /\ T]\ L\ mol^{-1}$

At pH 4.5:

$k_{14b}(T) = (8.5 \pm 1.9)\ 10^{12}\ exp[-(4670 \pm 2010)\ K\ /\ T]\ L\ mol^{-1}\ s^{-1}$

$k_{-14b}(T) = (3.8 \pm 0.5)\ 10^7\ exp[-(3560 \pm 680)\ K\ /\ T]\ s^{-1}$

$K_{14b}(T) = (2.2 \pm 0.1)\ 10^5\ exp[-(2270 \pm 150)\ K/T]\ L\ mol^{-1}$ $(298\ K \leq T \leq 328\ K)$

Finally, from the measurements of 'artifact HONO' in the Spindler et al. (2003) publication the unimolecular rate of decomposition for the adduct was determined as $k_{15a}(T) = (8.4 \pm 0.1)\ 10^{-3}\ s^{-1}$ (T = 298 K).

The most significant difference between the results of Spindler et al. (2003) and earlier studies is that the mechanism identified by Spindler et al. (2003) (adduct formation with a slow adduct decomposition) limits the potential for S(VI) formation via this mechanism under environmental conditions. Here, from the viewpoint of aqueous-phase thermochemistry, it should also be noted that such high rate constants for a prompt bimolecular reaction with a concerted single electron transfer from $HSO_3^-$ to $NO_2$ would not be feasible. The one-electron reduction potentials of $NO_{2(aq)}$ and $HSO_{3\ (aq)}^-$ are very similar with $E°\ (HSO_3^-) = 0.84\ V$ vs. NHE (Huie and Neta, 1984; Wardman, 1989) and $E°\ (NO_2^-/NO_2) = 1.04 \pm 0.02\ V$ vs. NHE (Armstrong et al., 2013).

The oxidation of S(IV) by $NO_2$ in aerosol water was previously proposed to be important during wintertime haze episodes in Beijing (Cheng et al., 2016; Wang et al., 2016). The significance of this S(IV) oxidation pathway rests on (a) the hypothesis that aerosols in Beijing have unusually high pH (Wang et al., 2016), which is not supported by thermodynamic models, and (b) the mechanism and relatively fast kinetic parameters of earlier studies by (Lee and Schwartz, 1983) and Clifton et al. (1988) without considering the more recent findings of Spindler et al. (2003).

Recent isotopic studies provide further evidence that this reaction is not important in Beijing (Au Yang et al., 2018; He et al., 2018a; Shao et al., 2019; Li et al., 2020a) in line with the aforementioned mechanistic and thermodynamic considerations.

The importance of the $NO_2 + HSO_3^-$ reaction has also been highlighted for fogs in China with pH > 5 (Xue et al., 2016; Xue et al., 2019). However, as with the aerosol aqueous chemistry, this sulfate production pathway should be self-limiting due to its production of $H^+$.

Peroxynitric acid ($HNO_4$), a product of the gas-phase reaction of $HO_2$ and $NO_2$, also oxidizes $HSO_3^-$, primarily in cloud water, with a rate constant of $3.3 \times 10^5\ L\ mol^{-1}\ s^{-1}$ (Amels et al., 1996; Warneck, 1999; Dentener et al., 2002). The reaction rate increases with increasing aqueous pH due to increased solubility of S(IV) and $HNO_4$. Besides the acidifying effect of S(IV) to S(VI) conversion, the reaction yields nitric acid ($HNO_3$) as an acidic byproduct. The significance of this pathway depends on gas-phase $HO_x$ and $NO_x$ levels and the relative abundance of other competing S(IV) oxidants.



## 4.6  Overall S(IV) oxidation considerations

To compare the potential atmospheric relevance of the different S(IV) to S(VI) conversion pathways with respect to different environmental and acidity regimes in aerosols, haze and clouds, initial S(IV) oxidation rates of the different pathways discussed up to here were calculated. Figure 7 shows the resulting calculated S(IV) oxidation rates of these reaction pathways in mol L$^{-1}$ s$^{-1}$ for continental urban haze and rural aerosol conditions as well as continental urban and rural cloud conditions. These rates were calculated with the rate expressions from the subsections above (Eq.s 6a, 7, 8, 9, 10, 11b, 12) and are based on the typical conditions as summarized in Table 1. For the NO$_2$, kinetic rates were calculated applying the pseudo-steady-state approximation ($k_{PSSA,HSO3-}$ = 1.3×10$^1$ L mol$^{-1}$ s$^{-1}$, $k_{PSSA,SO32-}$ = 2.7×10$^2$ L mol$^{-1}$ s$^{-1}$). For HNO$_4$, the reaction rate was calculated with a rate constant of 3.3×10$^5$ L mol$^{-1}$ s$^{-1}$ (Amels et al., 1996; Warneck, 1999; Dentener et al., 2002). For Fe(III) and Mn(II), the rate expressions by Hoffmann and Calvert (1985) and Martin and Hill (1987a) were applied, respectively. Note that the synergistic rates of Ibusuki and Takeuchi (1987) (Eq. 13) were not used due to the still large uncertainties of this oxidation pathway.





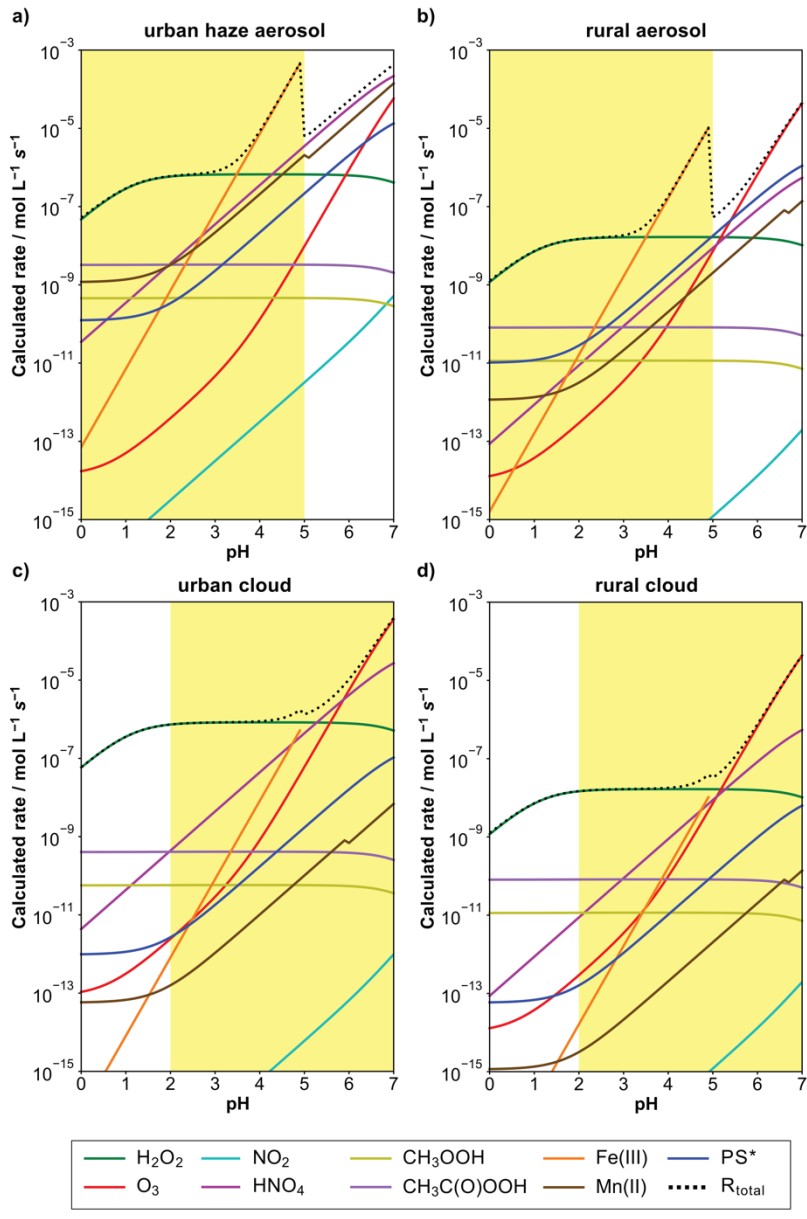

**Figure 7. Calculated S(IV) oxidation rates of different reaction pathways in mol L⁻¹ s⁻¹ for urban winter haze (a) and rural aerosol (b) conditions as well as urban (c) and rural (d) cloud conditions at 298 K. Applied conditions are given in Table 1 and the rate expressions used were those given in this text. The atmospherically relevant acidity range in the different cases is marked in yellow.**

For diluted cloud conditions, the S(IV) oxidation by dissolved $H_2O_2$, $O_3$, $HNO_4$ and potentially the iron-catalyzed pathway are the most important oxidation pathways (see Fig. 7c and 7d). The reaction with dissolved $H_2O_2$ is the major oxidation pathway under acidic cloud conditions. Under less acidic cloud conditions (pH > 5), the other reaction pathways are able to contribute



significantly to the S(VI) formation. Moreover, Fig. 7 displays that the oxidation rates of other oxidants such as $NO_2$, excited triplet states of photosensitizers (PS*) and organic hydroperoxides ($CH_3COOH$, $CH_3C(O)OOH$) are unimportant under cloud conditions mainly because of their low in-cloud concentrations.

Differently, under more concentrated haze and deliquesced aerosol conditions, the molar concentrations of TMIs are significantly higher. Thus, the contributions of TMI-catalyzed S(IV) oxidation pathways are elevated against cloud conditions. From the calculation output in Fig. 7a and 7b, it can be seen that the S(IV) oxidation by dissolved $H_2O_2$ is still predominant below pH ≤ 3. However already at quite low acidity conditions with pH ≈ 3.5, the TMI-catalyzed pathways can become the main oxidation route for S(IV). Note that the synergistic rate of Ibusuki and Takeuchi (1987) ((Eq. 13) were not included in

the current study, so even higher contributions of TMI-catalyzed S(IV) oxidation pathways can be possible. Besides the TMI-catalyzed S(IV) oxidation pathways, also S(IV) oxidations by dissolved $HNO_4$ and $O_3$ as well as, to some extent, PS* can be important under polluted haze and rural aerosol conditions when pH is above pH > 5, respectively. Importantly, the current comparison clearly shows that the $NO_2$-driven oxidation route even under very high $NO_x$ conditions (66 ppb) applied in the urban haze case still remains of minor importance. Only by the combination of applying unusually high aerosol pH

values, artificially low $H_2O_2$ concentrations and unrealistically fast kinetic parameters from earlier studies by Clifton et al. (1988), $NO_2$ rates can fall into the range of other key oxidants discussed here.

In conclusion, the outcomes of this overall comparison are in agreement with findings of isotope field investigations (see e.g., Harris et al. (2013); Au Yang et al. (2018); He et al. (2018a); Shao et al. (2019); Li et al. (2020a)) which have implicated that mainly $H_2O_2$, $O_3$, and TMI-catalyzed pathways are responsible for the S(IV) to S(VI) conversion in atmospheric aqueous-

phase cloud and aerosol solutions. However, due to the uncertainties still existing with regard to kinetics and mechanisms further acidity-dependent investigations are warranted.

## 4.7   Sequestering of S(IV) as HMS

$HSO_3^-$ or $SO_3^{-2}$ can react with a variety of aldehydes to form hydroxyalkylsulfonates (Olson and Hoffmann, 1989). Of particular interest has been S(IV) reaction with HCHO to produce hydroxymethanesulfonate (HMS, $HOCH_2SO_3^-$) (Munger et al., 1986).

The formation of HMS is strongly dependent on drop acidity, increasing rapidly at higher pH values due to increased partitioning of S(IV) to $HSO_3^-$ and $SO_3^{2-}$ (Rao and Collett, 1995). Furthermore, the reaction rate increases with increasing pH due to the fact that the rate coefficient for $SO_3^{2-}$ ($k = 2.5×10^7$ L $mol^{-1}$ $s^{-1}$) is more than four orders of magnitude higher than that for $HSO_3^-$ ($k = 790$ L $mol^{-1}$ $s^{-1}$) (Boyce and Hoffmann, 1984; Olson and Hoffmann, 1989). At pH values > 6, HMS formation becomes so fast that it can limit aqueous sulfate production in large droplets where mass transport limits $SO_2$ uptake

from the gas phase (Reilly et al., 2001). Since oxidation of HMS is slow (Hoigne et al., 1985; Kok et al., 1986; Barlow et al., 1997a, b), its formation effectively protects S(IV) from oxidation to S(VI) by non-radical oxidants such as $H_2O_2$, $O_3$ and others. Whiteaker and Prather (2003) demonstrated the utility of HMS measurements in single particles as a tracer for fog processing. Recent field and modeling studies have suggested that HMS production may also be an important contributor to fine particle sulfur content under polluted haze conditions (Moch et al., 2018; Song et al., 2019b; Ma et al., 2020; Moch et al., 2020).





Sulphur in particles may exist in the form of other sulphonates (R-C-SO$_3^-$) besides organosulphates (R-C-O-SO$_3^-$) (Le Breton et al., 2018; Brüggemann et al., 2020).

## 4.8 Acid-driven production of tropospheric reactive halogens: Multiphase halogen activation

Of the many acid-catalyzed reactions in the atmosphere, the acid-catalyzed formation of reactive halogens (Br, Cl, I) in the
troposphere has the potential to render acidity as an influencer of the oxidative capacity of the atmosphere, although its influence has yet to be fully quantified. Reactive halogens and halogen reservoir species are of the form Br$_y$ (= Br + 2 Br$_2$ + HOBr + BrO + HBr + BrNO$_2$ + BrNO$_3$ + IBr +BrCl), Cl$_y$ (= Cl + 2 Cl$_2$ + HOCl + ClO + HCl + ClNO$_2$ + ClNO$_3$ + ICl + BrCl + ClOO + OClO + 2 Cl$_2$O$_2$), and I$_y$ (I + 2 I$_2$ + HOI + IO + OIO + HI + HIO$_3$+ INO + INO$_2$ + INO$_3$ + 2 I$_2$O$_2$ + 2 I$_2$O$_3$ + 2I$_2$O$_4$). Tropospheric reactive halogens can impact the oxidation capacity of the atmosphere by acting as an effective sink for ozone
(O$_3$), e.g. bromine explosion events in the Arctic, and nitrogen oxides (NO$_x$ = NO + NO$_2$) and by influencing the partitioning of HO$_x$ (= OH + HO$_2$) (Oltmans et al., 1989; Simpson et al., 2015; Schmidt et al., 2016; Sherwen et al., 2016; Hoffmann et al., 2019a). Reactive halogens also directly impact the lifetime of reduced trace gases such as methane (CH$_4$) and non-methane volatile organic compounds (VOCs), dimethylsulfide (DMS), and mercury in the atmosphere (Barnes et al., 2006; Saiz-Lopez and von Glasow, 2012; Ariya et al., 2015; Simpson et al., 2015). Sources of tropospheric reactive halogens include oxidation
of organohalogens (e.g., CH$_3$Br and CH$_3$I) (Saiz-Lopez et al., 2012b; Saiz-Lopez and von Glasow, 2012), deposition of ozone to the ocean surface to yield HOI and I$_2$ (Carpenter et al., 2013), release from sea salt aerosols (Parrella et al., 2012; Schmidt et al., 2016; Sherwen et al., 2017) and, to a minor extent, transport from the stratosphere (Schmidt et al., 2016; Wang et al., 2019b). Liberation of halogens to their reactive form via acid-catalyzed reactions on sea salt aerosols (see Fig. 8) is the largest source of reactive bromine in the troposphere (Vogt et al., 1996; von Glasow et al., 2002a; Pechtl et al., 2007; Pechtl and von
Glasow, 2007; Parrella et al., 2012; Chen et al., 2017). As shown in Fig. 8, the formation and processing of reactive halogens strongly depends on the aqueous-phase conditions, i.e. the LWC and the acidity of the solution.



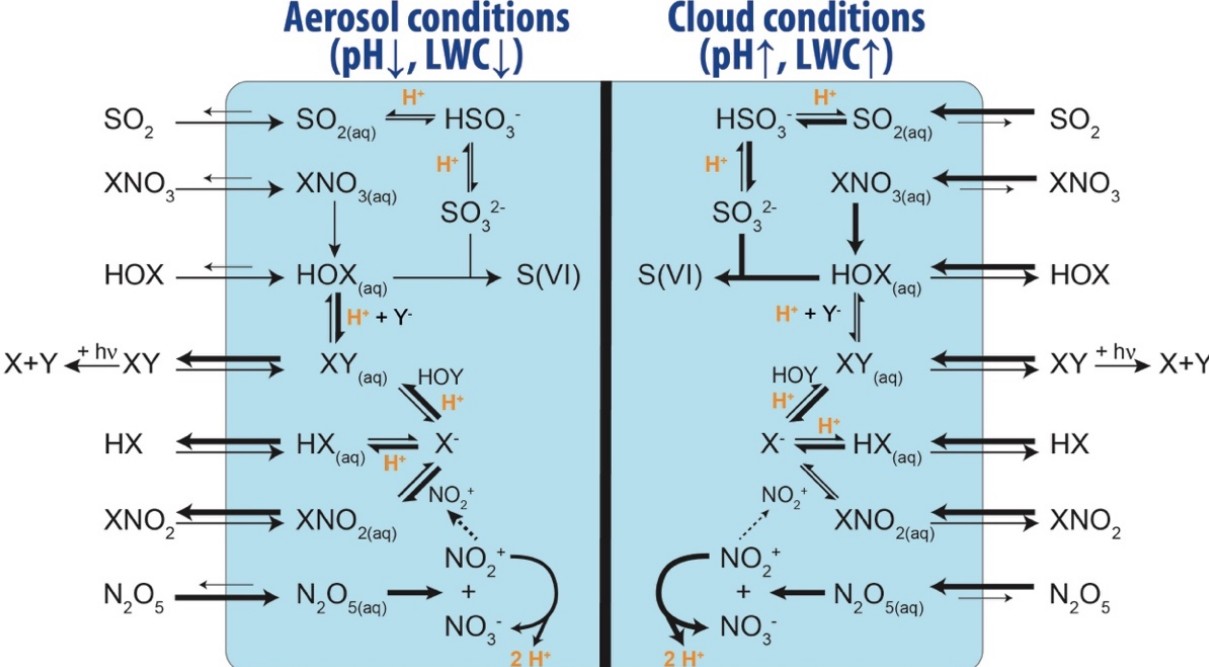

**Figure 8: Simplified scheme of the reactive halogen chemistry and their differences between diluted less acidic cloud conditions and more concentrated and acidic aerosol conditions. Differences in the chemical rates und uptake fluxes are indicated by lighter and thicker arrows.**

Formation of reactive halogens (Br, Cl, I) from sea salt aerosols proceeds in pristine environments via the uptake of hypohalous acid species (HOX, where X = Br, Cl, or I) from the gas phase (von Glasow et al., 2002b) or in more polluted environments via the hydrolysis of $N_2O_5$ forming $ClNO_2$ (Finlayson-Pitts et al., 1989; Roberts et al., 2009; Sarwar et al., 2014) as well as via the hydrolysis of $XNO_3$ forming HOX (Schmidt et al., 2016; Hoffmann et al., 2019b), see reactions 16a-16c:

$$HOX_{(aq)} + Y^- + H^+ \rightarrow XY_{(aq)} + H_2O \tag{R-16a}$$

$$N_2O_{5(aq)} + Cl^- \rightarrow ClNO_{2(aq)} + NO_3^-{}_{(aq)} \tag{R-16b}$$

$$XNO_{3(aq)} + H_2O_{(aq)} \rightarrow HOX_{(aq)} + HNO_{3(aq)} \tag{R-16c}$$

where X = Br, Cl, or I. If two different halogens are involved, Y denotes the second halogen atom. The formed XX or XY species then either reacts further or partitions to the gas phase, where it is photolyzed and participates in gas-phase oxidation chemistry to ultimately regenerate HOX or $XNO_3$. Non-linear reactive halogen production proceeds via uptake of one molecule HOX or $XNO_3$, carrying one halogen atom, yielding two halogen atoms released back to the gas phase (see Fig. 8). Note this is an acid-driven process which consumes $H^+$ in the aqueous-particle phase without recycling it and also one halogen anion $Y^-$ is consumed. When Br is a participant, this auto-catalytic reaction cycle can lead under high bromide concentrations to so-called bromine explosion events characterized by high concentrations of BrO (Evans et al., 2003) resulting from the gas-phase reaction of Br with $O_3$.



Changing atmospheric acidity due to changes in anthropogenic emissions of acid precursor gases may influence the formation of reactive halogens via reaction (R-16a) (Keene et al., 1998). However, lower acidity conditions might also result in stronger aqueous-phase partitioning of hydrogen halides which might partly compensate for the reduced acidity effect via reaction

(R-16a). Changes in sulfur dioxide ($SO_2$) may contribute to sources or sinks of reactive halogens. The formation of sulfate, from the oxidation of $SO_2$, is typically the largest source of acidity in the atmosphere (see Sect. 3). However, reactions of HOX with dissolved S(IV) ($HSO_3^-$ + $SO_3^{2-}$) in aqueous aerosols can convert halogens to their less-reactive acid form (HX) via R-17 and R-18 (Fogelman et al., 1989; Troy and Margerum, 1991; von Glasow et al., 2002a; Chen et al., 2017; Liu and Abbatt, 2020). Here, especially the reaction with HOI can be very significant (Pechtl and von Glasow, 2007; Bräuer et al., 2013;

Hoffmann et al., 2019a).

$$HSO_3^- + HOX \rightarrow SO_4^{2-} + HX + H^+ \tag{R-17}$$

$$SO_3^{2-} + HOX \rightarrow SO_4^{2-} + HX \tag{R-18}$$

The rate expression for S(VI) formation by S(IV) + HOX is given by:

$$R_{HOX,1} = k_{17,HOX}[HSO_3^-][HOX] \tag{17}$$

$$R_{HOX,2} = k_{18,HOX}[SO_3^{2-}][HOX] \tag{18}$$

with recommended rate constants for HOCl of $k_{17,HOCl} = 2.8 \times 10^5$ L mol$^{-1}$ s$^{-1}$ (Liu and Abbatt, 2020) and $k_{18,HOCl} = 7.6 \times 10^8$ L mol$^{-1}$ s$^{-1}$ (Fogelman et al., 1989), and for HOBr of $k_{17,HOBr} = 2.6 \times 10^7$ L mol$^{-1}$ s$^{-1}$ (Liu and Abbatt, 2020) and $k_{18,HOBr} = 5.0 \times 10^9$ L mol$^{-1}$ s$^{-1}$ (Troy and Margerum, 1991), respectively. Unfortunately, reaction rate constants for HOI with dissolved S(IV) ($HSO_3^-$ + $SO_3^{2-}$) have not been measured yet. However, following the augmentation of Pechtl et al. (2007),

the reaction rate constants of HOI with $HSO_3^-$ and $SO_3^{2-}$ should be even faster than the reaction rate constants of HOCl and HOBr or is likely diffusion-limit controlled.

Finally, the overall impact of changes in anthropogenic emissions of $SO_2$ or other acid-gas precursors on tropospheric reactive halogen production remains unknown. Because of the impact of reactive halogens on the radiative forcing of the powerful greenhouse gas ozone (Saiz-Lopez et al., 2012a) as well as aerosol particle composition (Hoffmann et al., 2016; Lee et al.,

2019), their chemistry can be of crucial importance for climate predictions. Therefore, more laboratory investigations, chamber studies and accompanied modelling efforts are needed to determine chemical reaction rate constant of crucial halogen processes, such as the oxidation of S(IV) by HOI, and to better characterize the overall reactive cycling of halogens including its sensitivity to aerosol particle and cloud acidity.

### 4.9 Discussion and outlook: atmospheric multiphase chemistry of inorganic species

Multiple reactive pathways for the conversion of S(IV) to S(VI) have been discussed here. Many of these processes are limited in atmospheric aerosols by acidic conditions and the presence of particle-phase organics, which quench highly reactive radical and triplet species. Studies from the past four decades have shown that, under polluted conditions, such as found in urban areas worldwide or in the North China Plain (NCP), only relatively stable oxidants or TMI catalysis may lead to the required rate of



S(IV) to S(VI) conversion to explain the observed S(VI) budgets (Jacob and Hoffmann, 1983; Chameides, 1984; Saxena and Seigneur, 1987; Seigneur and Saxena, 1988; Pandis et al., 1992; Amels et al., 1996; Berglund and Elding, 1996). That being said, our understanding of atmospheric multiphase sulfate production, especially in the aerosol phase, is still incomplete, despite more than a century of studies on aqueous sulfur oxidation. S(IV) conversion explaining the aerosol sulfate budgets encountered today, especially under urban or semi-urban polluted conditions still need further elucidation from the basic aqueous-phase processes to concrete field measurements. This includes the role of acidity in these processes which could be decisive to whether or not a process can really be important in the environment.

Areas of focus should include:

a) Laboratory studies of S(IV) oxidation by all pathways under atmospheric aerosol conditions, i.e. in aerosol flow tube reactors, to assess the impact of high ionic strength and other factors specific to the aerosol phase

b) Advanced sulfur-isotope measurements of ambient aerosol and cloud water samples to identify driving sulfur oxidation pathways under various atmospheric conditions

c) Advanced knowledge of TMI-catalyzed S(IV) oxidation pathways, including investigation of synergy effects and the role of other metal catalysts present in aqueous atmospheric solutions besides Fe and Mn. The impact of acidity and ionic strength on both the speciation of TMIs, i.e. their presence in free and complexed form, and the specific chemical reaction rates of single-TMIs have to be studied.

d) Kinetic and mechanistic investigations on other potential oxidants, especially comparatively stable oxidants such as ROOHs and HOI

e) Investigations of pH-dependent in-situ formation of key S(IV) oxidants such as $H_2O_2$ and ROOH resulting from TMI-$HO_x$-DOM chemistry.

## 5 Interactions of acidity and chemical processes: Organic systems

Acidity in aerosol particles can strongly enhance secondary organic aerosol (SOA) formation (Jang and Kamens, 2001; Jang et al., 2002; Jang et al., 2003; Iinuma et al., 2004; Jang et al., 2004; Liggio and Li, 2006; Surratt et al., 2007b). These early observations triggered immense research interest in investigating aqueous-phase reactions leading to the accumulation of organic particle constituents. These so-called 'accretion reactions' are often acid-driven, or even acid-catalyzed. In the following, the most important organic compound families and the influence of acidity on their aqueous-phase chemistry are discussed. In this section we discuss the role of acidity on the gas-particle partitioning of semi-volatile organic compounds through its influence on hydration of carbonyls and dicarbonyls. We then discuss in detail the impact of acidity on the multiphase oxidation of organic material. Oxidative organic chemistry can be influenced by acidity because this influences the reactant speciation, such as in acid and diacid oxidations by radicals and non-radical oxidants such as dissolved ozone. Finally, we discuss accretion reactions.



## 5.1 Acidity and hydration of carbonyl compounds

Carbonyl compounds (i.e., aldehydes or ketones), omnipresent in the tropospheric gas and aqueous phase, result from primary emissions or are secondary oxidation products. The photolysis of carbonyl compounds can be important for both their degradation in the troposphere and gas-phase oxidant production. Water-soluble carbonyl compounds may partition into the
aqueous phase of deliquesced aerosols and cloud/fog droplets. Once in the aqueous phase, carbonyl compounds can undergo hydration, leading to conversion of carbonyl group into gem-diol moieties. As hydration processes are typically acid- or base-catalyzed, the acidity of an aqueous solution can affect the hydration and consequently all other processes linked to it. With regard to phase partitioning, the hydration equilibria increase the effective partitioning of the carbonyl-containing compound towards the aqueous phase (Sumner et al., 2014). Moreover, compared to the carbonyl group, the diol functionality is
photochemically inactive. Thus, partitioning to the aqueous phase and subsequent hydration can, in part, protect carbonyl compounds from photolysis and shut off possible photochemistry of the carbonyl group (George et al., 2015; Herrmann et al., 2015; McNeill and Canonica, 2016). However, hydrated aldehydes are often characterized by a somewhat lower reactivity with radical oxidants such as OH compared to the unhydrated carbonyl species (Schuchmann and von Sonntag, 1988).

This sub-section summarizes the present knowledge on the acidity dependence of carbonyl hydration constants, and
implications for the chemical conversions of carbonyl compounds in atmospheric aqueous media.

### 5.1.1 The influence of acidity on hydration constants and its implications

The reversible hydration and dehydration of the carbonyl group of an aldehyde or ketone in the aqueous phase is illustrated in Fig. 9 (Bell and Darwent, 1950; Bell, 1966; Ogata and Kawasaki, 1970; Lowry and Richardson, 1976).


**Figure 9: General mechanism of the acid- (A) or base- (B) catalyzed formation of diols resulting from the hydration of the carbonyl group.**





Simple aldehydes such as formaldehyde (Zavitsas et al., 1970; Li et al., 2011; Rivlin et al., 2015), acetaldehyde (Ahrens and
Strehlow, 1965; Kuschel and Polarz, 2010) and glyoxal (Liggio et al., 2005a; Loeffler et al., 2006) tend to self-oligomerize,
e.g. via hemiacetal formation or aldol condensation, which further influences the hydration equilibrium. .

The ratio of the hydrated and dehydrated fraction under equilibrium conditions is described by equilibrium constant $K_{hyd}$,
defined as follows for a dilute aqueous solution:

$$K_{hyd} = \frac{[diol\ compound]}{[carbonyl\ compound]} = \frac{k_{hyd}}{k_{dehyd}} \tag{19}$$

Where $k_{hyd}$ is the rate constant for hydration and $k_{dehyd}$ is the rate constant for dehydration. In general, the hydration constants
$K_{hyd}$ decrease with decreasing electron-withdrawing power of the substituent in a substituted organic acid (Clayden et al.,
2012). The equilibrium constants of simple aldehydes or ketones generally show no pH dependence but are dependent on
temperature.

For most carbonyls, the hydration reaction with $H_2O$ under neutral conditions is slow. In the presence of hydrogen ions,
hydroxyl ions, undissociated acid molecules and anion bases, the hydration reaction proceeds faster. The overall hydration rate
considering all acid and base dependencies can be calculated by means of Eq. 20 (Ogata and Kawasaki, 1970; Lowry and
Richardson, 1976):

$$k_{hyd} = k_0 + k_{H+}[H_3O^+] + k_{OH}[OH^-] + \left(k_a + k_b \frac{[B]}{[HA]}\right) \tag{20}$$

The catalytic constants of the hydration rate in equation (11) are described as follows: $k_0$ for the solvent influence, $k_{H+}$ for the
effect of the $H_3O^+$ ion, $k_{OH}$ as influence of the $OH^-$ ion, $k_a$ and $k_b$ as general acid or base contribution (Ogata and Kawasaki,
1970). An overview of the acid or base catalytic constants for the hydration of formaldehyde and acetaldehyde by a few
different organic acids is presented in Table 2. As can be seen from the data compiled in Table 2, the presence of acids clearly
influences the hydration rate of the carbonyl group.

Experimentally determined values for $K_{hyd}$ and our recommended values are presented for several atmospherically relevant
simple aldehydes, ketones and α-oxocarboxylic acids in the Supporting Information Tables S3 and S4. The general trend in
$K_{hyd}$ is: glyoxal (2nd hydration) > formaldehyde > methylglyoxal (CHO-group hydration) > glyoxylic acid > glyoxal
(1st hydration) > glyoxylate > glycolaldehyde > pyruvic acid > biacetyl > acetalydehyde > propanal > butanal > pivealdehyde
> pyruvate > acetone. These data are discussed in more detail in the Supporting Information.

It is important to note that the hydrolysis of simple aldehydes and ketones and dicarbonyls is unaffected by pH. For
multifunctional carbonyl compounds, the hydration equilibrium constant of the carbonyl group is strongly influenced by the
electronic effects of the adjacent group. The hydration of carbonyl groups in compounds that also contain pH sensitive
moieties, such as α-oxocarboxylic acids, is highly influenced by the acidity of the surrounding environment.

Besides the hydration and dissociation equilibria, condensation (dimerization or polymerization) equilibria as well as keto-
enol-equilibria could influence these compounds (Fig. 10). The equilibria are related by $K_{hyd.1} \times K_{diss.2} = K_{diss.1} \times K_{hyd.2}$, while
the apparent dissociation constant is given by





$$K_{Hyd} = \frac{[H^+] \times K_{diss,1} + K_{hyd,2} \times K_{diss,1}}{[H^+] + K_{diss,1}} \tag{21}$$

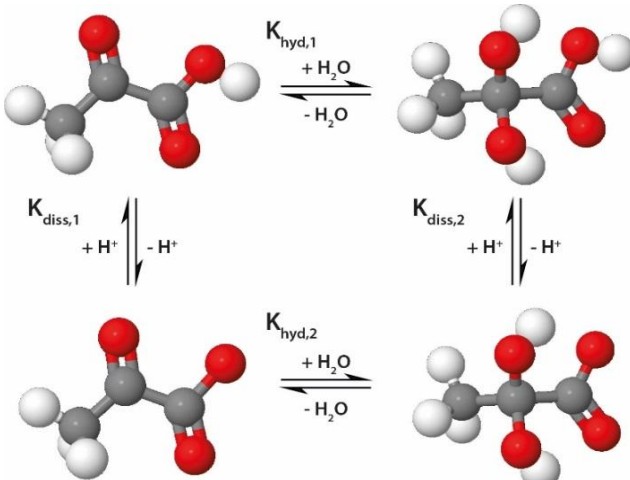

**Figure 10: Scheme describing the four equilibria of pyruvic acid, a representative α-keto-carboxylic acid, in aqueous solution.**

An overview of the determined $K_{hyd}$ values for atmospherically relevant α-oxocarboxylic acids is given in Table S4 and the existing data are outlined for each of the listed chemical compounds separately in the Supplement. The most prominent α-oxocarboxylic acid compounds in the atmosphere are glyoxylic acid and pyruvic acid. Special emphasis in the recent literature was put on pyruvic acid. The recent data on pyruvic acid is summarized in Fig. 11.

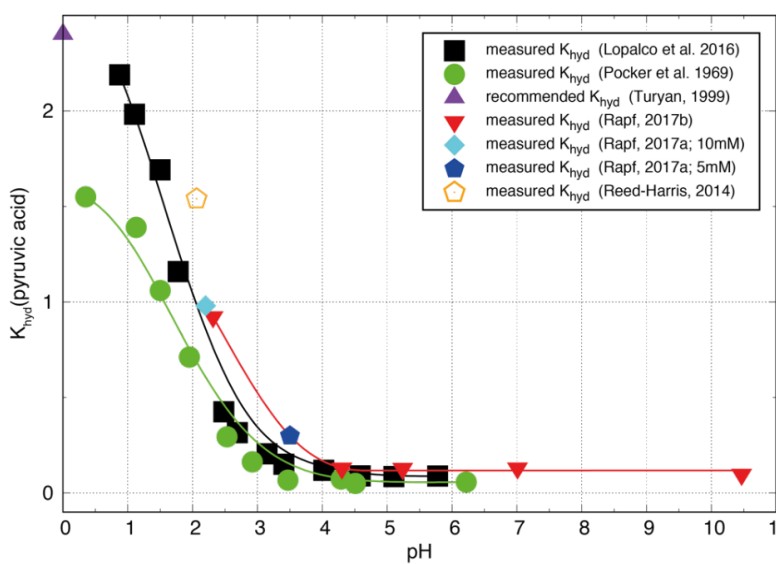


**Figure 11: Measured pH dependency of the apparent hydration equilibria (see Eq. 21) of pyruvic acid in aqueous solution. Experimental data are shown from the literature (Pocker et al., 1969; Reed Harris et al., 2014; Lopalco et al., 2016; Rapf et al., 2017a; Rapf et al., 2017b).**





As shown in Fig. 11, $K_{\text{hyd}}$ for pyruvic acid increases rapidly with decreasing pH for pH < 3. Note that the formation of the
hydrated pyruvic acid (2,2-dihydroxypropanoic acid) is also dependent on the water concentration (Pocker et al., 1969; Maron
et al., 2011) – which may have implications for aqueous aerosol chemistry.

The impact of acidity and its feedback on the hydration, as well as their impact on the photochemistry of pyruvic acid have
been examined by spectroscopic investigations performed at TROPOS. These investigations have shown that the molar
absorption coefficient spectra of pyruvic acid are rather different under low and high acidity conditions. Measured absorption
coefficient spectra of pyruvic acid at pH = 0 and pH = 9 (Fig. 12) shows higher absorption coefficients under pH = 9
conditions.

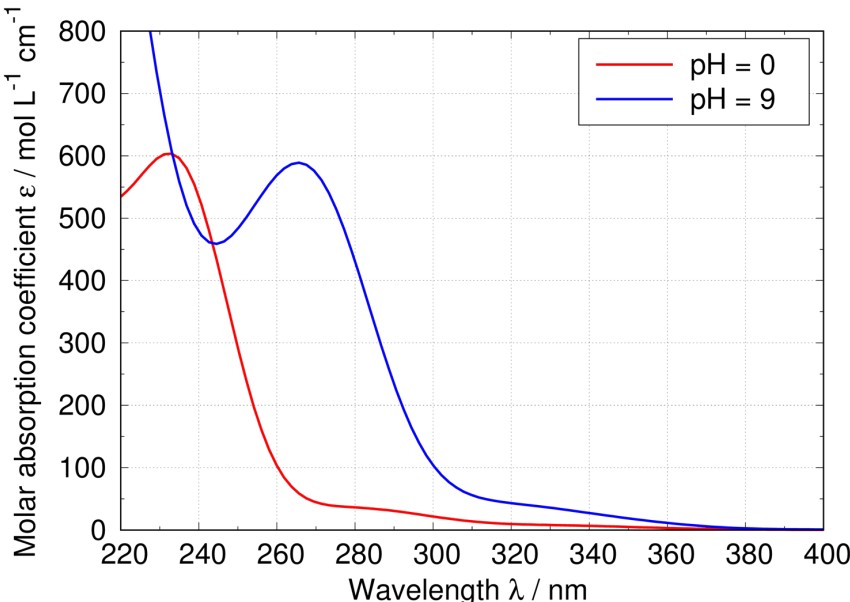

Fig. 12. Measured UV absorption coefficient spectra of pyruvic acid in water under acidic (pH = 0, red line) and alkaline
(pH = 9, blue line) conditions.


At 300 nm wavelength, the measured absorption coefficient is about 4 times larger at pH = 9 than at pH = 0. Under high pH
conditions (pH = 9), a large fraction of pyruvic acid is present in its unhydrated form and, consequently, higher absorption
coefficients are observed compared to very acidic conditions, where pyruvic acid is mainly present in its photochemically
inactive hydrated form. This difference has implications for photochemistry of pyruvic acid which will become less efficient
in more acidic solutions compared to less acidic ones. Such effects should be implemented into aerosol liquid water chemistry
models. Finally, hydration processes can be characterized by both temperature and acidity dependencies particularly for
α-/β-keto-carboxylic acids such as pyruvic acid. These dependencies need to be included in future models to be able to
accurately investigate their impact on the partitioning and occurring multiphase chemistry. For that reason, more laboratory
studies are needed to extend the available database for other atmospherically-relevant functionalized α-/β-keto-carboxylic





acids. In concrete terms, more studies appear desirable for glyoxylic acid, mesoxalic acid, oxalacetic acid and oxalglycolic
       acid.

## 5.2    pH-sensitive organic accretion reactions

Organic accretion reactions are considered to be a source of high-molecular-weight organic material in atmospheric aerosols,
playing a key role in the formation of secondary organic aerosol material (Barsanti and Pankow, 2004, 2005, 2006). These
       reactions are typically multistep, bond-forming reactions, and are highly pH-sensitive. Many organic accretion reactions are
       acid- or base-driven or, in some cases, even catalyzed. Others require acidity, e.g. in the form of the participation of a proton
       ($H^+$) in the reaction, but are not technically 'acid catalyzed' because the proton is incorporated into the formed reaction product
       (e.g., ring-opening of epoxides, cf. Sect. 5.3)). Examples of atmospherically important accretion reactions include (i) aldol
condensation, (ii) hemiacetal and acetal formation, and (iii) esterification of carboxylic acids which will be treated in the
       following subsections.

       The current kinetic and mechanistic knowledge on tropospheric accretion reactions has been summarized not too long ago in
       a review by Herrmann et al. (2015) and a book by Barker et al. (2016). Accordingly, the present subsection only briefly outlines
       the mechanisms and emphasizes their dependence on acidity. For further specific details on organic accretion reactions please
see Herrmann et al. (2015), Zhao et al. (2016) and references therein.

### 5.2.1    Aldol condensation and ammonium catalyzed reactions

*Overview*

The Aldol (short form of 'aldehyde alcohol') condensation is a carbon-carbon-bond formation requiring the participation of
an enolizable carbonyl compound (e.g., ketones or aldehydes with an α-hydrogen) (Loudon, 1995). Under acidic conditions,
       the enol reacts with a protonated carbonyl compound to form the aldol addition product. The product may be dehydrated to
       form the aldol condensation product, a conjugated enone compound. The acid- or base-catalyzed nature of the enol formation,
       as well as the role of the protonated carbonyl reactant under acidic conditions, make this family of reactions pH-sensitive. This
       pathway has been suggested as a source of light-absorbing secondary organic material (i.e., brown carbon) in atmospheric
aerosols (Laskin et al., 2015; Nozière et al., 2015), which has been discussed in more detail for acetaldehyde, glyoxal as well
       as methylglyoxal (Laskin et al., 2015; Nozière et al., 2015), which was discussed in more detail for acetaldehyde, glyoxal as
       well as methylglyoxal (Noziere and Esteve, 2005; Nozière et al., 2007; Noziere and Esteve, 2007; Noziere and Cordova, 2008;
       De Haan et al., 2009a; Shapiro et al., 2009; Bones et al., 2010; Sareen et al., 2010; Li et al., 2011; Yu et al., 2011; Kampf et
       al., 2012; Nguyen et al., 2012; Laskin et al., 2015; Lin et al., 2015; Maxut et al., 2015; Nozière et al., 2015; Van Wyngarden
et al., 2015; Aiona et al., 2017; Rodriguez et al., 2017). Several studies which focused on sulfuric acid-catalyzed aldol
       formation have shown that this chemistry occurs efficiently only under strongly acidic conditions (Duncan et al., 1999; Kane





et al., 1999; Imamura and Akiyoshi, 2000; Nozière and Riemer, 2003; Esteve and Noziere, 2005; Noziere and Esteve, 2005; Liggio and Li, 2006; Noziere et al., 2006; Casale et al., 2007; Noziere and Esteve, 2007; Krizner et al., 2009).

*Surface films*

Van Wyngarden et al. (2015) reported on the formation of surface films from $H_2SO_4$-propanal mixtures with or without glyoxal and/or methylglyoxal. These films tended to form faster when the acidity was increased up to 48 wt % $H_2SO_4$, but with an acidity of 76 wt % $H_2SO_4$, the film formation slowed down or even stop in all mixtures except propanal/glyoxal.

*Mechanistic and kinetic considerations*

Yasmeen et al. (2010) suggested that the favorable mechanism under acidic conditions pH < 3.5 is the acetal formation, while
the aldol condensation only occur at a pH = 4-5, which is in contrast to the above-mentioned conditions. Ammonium-catalyzed or amine-catalyzed aldol condensation proceeds under higher, but still acidic, pH values typical for tropospheric aerosol particles (Noziere and Cordova, 2008; De Haan et al., 2009a; Noziere et al., 2010; Sareen et al., 2010; Li et al., 2011; Sedehi et al., 2013; Powelson et al., 2014; Aiona et al., 2017). The pH-dependent rate constants for aldol condensation reactions of glyoxal and methylglyoxal with ammonium sulfate and amines have been further investigated in several studies (Noziere et
al., 2009; Sareen et al., 2010; Yu et al., 2011; Kampf et al., 2012; Sedehi et al., 2013; Powelson et al., 2014; Yi et al., 2018). Noziere et al. (2009) observed that the pathway via iminium ion is faster at higher pH, whereas the aldol condensation is favored at lower pH values, which suggests a pH dependency incorporation of N-containing products. The results of pH dependency appear to be contradictory. On the one hand, Sareen et al. (2010) reported an enhanced product formation by decreasing the pH and concluded an acid-catalyzed aldol formation of the light-absorbing product at 280 nm. On the other
hand, Yu et al. (2011) reported an exponential increase of the formation rate of condensation products, e.g. imidazole by increasing the pH and concluded an ammonium-catalyzed reaction. Similarly, Kampf et al. (2012) observed a higher production rate of the imidazole bicycle with increasing pH values. Furthermore, Sedehi et al. (2013), showed a strong pH-dependence with an increasing reaction rate proportional to the concentration of the deprotonated amine or in other words an increase of the pH value. Nevertheless, the pH-dependent character of the reaction of ammonium sulfate or amine reaction
with glyoxal is stronger than for methylglyoxal. A study by Yi et al. (2018), describes an acceleration of the pH-dependent reaction of ammonium sulfate or amine in the presence of glycolaldehyde, whereas no cyclic compounds (e.g. imidazole) were formed in this reaction. Powelson et al. (2014), Grace et al. (2019) and (Li et al., 2019) reported the formation of heterocyclic compounds under similar conditions. Hawkins et al. (2018) reported an increasing formation of pyrazine-based chromophores in an aqueous mixture containing methylglyoxal and ammonium sulfate by increasing the pH from 2 to 9. This indicates that
the nitrogen nucleophile is more important than the acid-catalyzed aldol condensation, which is consistent with the observation of Kampf et al. (2012), Kampf et al. (2016) and Yi et al. (2018). A theoretical analysis of glyoxal condensation in the presence of ammonia conduct by Tuguldurova et al. (2019) describes two different imidazole formation pathways: the imine pathway and the aminoethanetriol pathway. These authors reported that the imine concentrations are very low due to the high-energy barriers for imine formation. Although a pH decrease due to amino alcohol dehydration leads to higher imine concentrations,
it also leads to higher ammonium cation formation, which is another difficulty for ammonium addition to the carbonyl group.





The second proposed pathway, which includes the intermediate aminoethanetriol, has a lower energy barrier and appears to be kinetically favourable due to the higher concentrations. Finally, imidazole formation is determined by the glyoxal concentration, the ratio of glyoxal/ (amine or ammonium), the composition of the solvent, and the pH value.

All in all, aldol condensations are today generally regarded as demanding to drastically acidic conditions to be really important in particle and multiphase chemistry.

### 5.2.2 Hemiacetal and acetal formation

Hemiacetal and acetal formation are the addition of an organic molecule containing either one or two hydroxyl groups (e.g. alcohols) to a carbonyl compound leading to the formation of one(hemiacetal) or two (acetal) ether-type C−O−C bonds. This type of accretion reaction is significant for aqueous secondary organic aerosol formation involving glyoxal, methylglyoxal, acetaldehyde, formaldehyde, and other common atmospheric carbonyl-containing compounds (Schweitzer et al., 1998; Tobias and Ziemann, 2000; Jang et al., 2002; Kalberer et al., 2004; Hastings et al., 2005; Liggio et al., 2005b, a; Loeffler et al., 2006; Zhao et al., 2006; De Haan et al., 2009a; De Haan et al., 2009b; Shapiro et al., 2009; Sareen et al., 2010; Yasmeen et al., 2010; Li et al., 2011). Hemiacetal formation is initiated by protonation of a carbonyl group, followed by nucleophilic addition of the alcohol (Loudon, 1995). After the deprotonation of the attacking alcohol, the hemiacetal is formed. Promoted by acidity, the hemiacetal can react further to a full acetal, by protonation of the alcohol group of the hemiacetal to eliminate water again under the formation of a carbocation. This carbocation can react in a subsequent reaction with another alcohol molecule to form the full acetal by deprotonation of the hydroxyl group. Hemiacetal and acetal formation are reversible. In addition to the aldol condensation product, Liggio et al. (2005a, 2005b); and Liggio and Li (2006) reported on an acetal formation during the reactive uptake of glyoxal and pinonaldehyde on acidic aerosols. It has been reported by De Haan et al. (2009b) that glyoxal is more prone to undergo the acetal formation, while methylglyoxal reacts mainly by the aldol condensation reaction mentioned above, whereas the contribution of oligomer formation was strongly dependent on the relative humidity and hence the particulate water concentration. Holmes and Petrucci (2006, 2007) observed the formation of hemiacetals in the oligomerization process of levoglucosan induced by the Fenton chemistry. Noziere et al. (2009, 2010) observe a pH dependent ammonium-catalyzed acetal formation from glyoxal and acetaldehyde. The hydration and the subsequent acetal formation involving methylglyoxal is strongly dependent on the pH value and occur at a pH < 3.5 (Yasmeen et al., 2010). Maxut et al. (2015) investigated also the ammonium catalyzed imidazole formation with glyoxal in neutral aqueous solution concluded that the contribution of acetal oligomer formation pathway is small. Grace et al. (2019) referred to the study by De Haan et al. (2011) and Kampf et al. (2016), who reported that aldol condensation type reactions are more important than acetal or hemiacetal formation under atmospheric conditions. In summary, hemiacetal and acetal formation requires acidic conditions, but the contribution of this reaction pathway is small compared to aldol formation under atmospheric conditions.



### 5.2.3 Esterification of carboxylic acids

Esterification is a reversible, acid- or base-catalyzed condensation reaction of carboxylic acids and hydroxyl group containing molecules under the formation of an C(O)−O−C type bond (Ingold, 1969; Larson and Weber, 1994) .The acid-catalyzed

mechanism can be described as follows: The carbonyl group of the undissociated carboxylic acid can be protonated under acidic conditions to form a carbocation. The carbocation then is subject to a nucleophilic attack by a hydroxyl group-containing molecule. The resulting intermediate further reacts by tautomerization (proton shift in the molecule), which subsequently decays in an equilibrium reaction to a protonated ester and a water molecule (Loudon, 1995).

The base-catalyzed mechanism includes the reaction of a carboxylate group (resulting from the deprotonation of carboxylic

acid group) and a hydroxyl group-containing molecule, such as alcohol. First, a proton transfer from the alcohol to the carboxylate occurs. Second, the deprotonated hydroxyl group reacts in a nucleophilic attack with the carbon atom of the carboxylic acid forming a metastable intermediate, which subsequently decays to an ester molecule and a hydroxide ion in an equilibrium reaction. In addition to being pH-sensitive, esterification reactions are also strongly dependent on the water content. The majority of esters are hydrolyzed in the presence of water. Both the formation and the hydrolysis of esters are

slow processes under tropospheric conditions. Moreover, the hydrolysis rate of esters will increase with increasing acidity (Mabey and Mill, 1978). Altieri et al. (2006) and (2008) reported the esterification mechanism occur in the cloud processing of pyruvic acid (pH = 2.7-3.1) and methylglyoxal (pH = 4.2-4.5). The oxidation leads to carboxylic acids and proceeds through α- or β-hydroxy acid to esters or oligoesters, similarly to the proposed mechanisms for oligomers in the aerosol phase (Gao et al., 2004; Tolocka et al., 2004; Surratt et al., 2006a; Surratt et al., 2007a). Since then, ester formation by oxidation of organic

matter in the troposphere has been the subject of many laboratory investigations (Hamilton et al., 2006; Surratt et al., 2006b; Szmigielski et al., 2007; Altieri et al., 2008; Galloway et al., 2009; Zhang et al., 2011; Birdsall et al., 2013; Kristensen et al., 2013; Strollo and Ziemann, 2013; Claflin and Ziemann, 2019; Mekic et al., 2019) and field studies (Raja et al., 2008; Raja et al., 2009; Kristensen et al., 2013). The work from (Birdsall et al., 2013) suggested that esterification by condensation of carboxylic acids with hydroxyl group containing molecules is not efficient enough to explain the oligoesters under realistic

aerosol acidities. In a recent study by Zhao et al. (2019), heterogeneous oxidation processes near the gas-particle interface open up a further formation pathway of ester-like structures, namely the dimerization of organic oxygen-containing radicals leads dominantly to ester formation. In summary, it should be noted that this accretion reaction in the atmospheric aerosol-phase depends more on the hygroscopicity, than the acidity of the aerosols (Zhao et al., 2006; De Haan et al., 2009a), since the hydrolysis competes with the ester formation.

## 5.3 Epoxide reactions

*Isoprene -derived epoxides*

In the last decade, acid-catalyzed ring-opening reactions of epoxides (see Fig. 13) in aqueous aerosols have emerged as a significant source of secondary organic aerosol material. In the aqueous phase, protonation of the epoxide by an acid ($H_3O^+$ or



NH$_4^+$ (Minerath et al., 2008; Minerath and Elrod, 2009; Eddingsaas et al., 2010; Nguyen et al., 2014; Noziere et al., 2018))
occurs in concert with nucleophilic addition. Typically, the participating nucleophiles are H$_2$O, HSO$_4^-$, and SO$_4^{2-}$, although
amines (Stropoli and Elrod, 2015) and alcohols (Surratt et al., 2010; Piletic et al., 2013) can also add. Nucleophilic attack by
H$_2$O results in hydrolysis and polyol formation, thus explaining the presence of isoprene-derived tetrols in particles (Kourtchev
et al., 2005; Xia and Hopke, 2006; Liang et al., 2012; Zhang et al., 2013). The hydrolysis of epoxides catalyzed by NH$_4^+$ can
only be important in less acidic aerosol solutions due to the orders of magnitude lower rate coefficients (Noziere et al., 2018).
Addition of HSO$_4^-$ or SO$_4^{2-}$ to the protonated epoxide in the aerosol phase is a more efficient pathway for organosulfate (OS)
formation than radical mechanisms (McNeill et al., 2012; Schindelka et al., 2013). While the formation of polyols via
hydrolysis of epoxides may be acid catalyzed (Eddingsaas et al., 2010), OS formation can consume H$^+$ (Riva et al., 2019;
Brüggemann et al., 2020) (e.g. see Fig. 13).


**Figure 13. Schematic of the OS and polyol formation via acid-catalyzed ring-opening reactions of epoxides.**

Isoprene epoxydiol (IEPOX), a photooxidation product of isoprene (Paulot et al., 2009; Surratt et al., 2010), is calculated to
contribute 34% of global SOA mass (Lin et al., 2012), and 28% of organic aerosol mass in the SE USA (Marais et al., 2016).
The reactive uptake of IEPOX to aqueous media is strongly pH-dependent, with the reactive uptake coefficient decreasing
rapidly with increasing pH for pH > 1 (Gaston et al., 2014). For this reason, the rate of IEPOX SOA formation is slow in cloud
water (McNeill, 2015), but given the relatively large liquid water content of clouds and the relatively large water solubility of
IEPOX, it could be significant in more acidic cloud droplets (pH 3-4) (Tsui et al., 2019).

*Terpene-derived epoxides*
In regions with lower isoprene but higher monoterpene emissions, e.g., in the boreal forests, monoterpene-derived OSs formed
via different proposed pathways can also contribute to SOA mass in atmospheric aerosols (see Brüggemann et al. (2020) for
details). Their importance for SOA is still not well characterized. Formation of monoterpene-derived OS has been observed in





chamber experiments and measured in field samples (Iinuma et al., 2004; Iinuma et al., 2007; Ye et al., 2018b; Brüggemann et al., 2019; Cui et al., 2019). However, there are only a few measurements of monoterpene-derived OSs in boreal forests areas.

OS formation via acid-catalyzed ring-opening reactions of several monoterpene epoxides (β-pinene oxide, limonene oxide, and limonene dioxide) has been kinetically and mechanistically investigated (Cortes and Elrod, 2017). Investigations demonstrated that monoterpene epoxides react faster than IEPOX in aqueous solution and might even react in less acidic solutions. However, this study also revealed that the formed OS compounds are not long-lived compounds under aqueous aerosol conditions and may quickly react further mainly through hydrolysis. Therefore, Cortes and Elrod (2017) concluded that other OS formation mechanisms, than the acid-catalyzed ring-opening mechanism of monoterpene epoxides, are needed to explain the formation of more long-lived OS from monoterpenes. In agreement with these findings, recent chamber studies on the OS formation from α-pinene oxidation (Duporte et al., 2016; Duporte et al., 2020) showed that the OS yield, including the subsequent formation of OS dimers and trimers, decreases with increasing relative humidity. Furthermore, these studies revealed that effective formation rates of OS from α-pinene are 2 orders of magnitude higher under very acidic aerosol conditions and that the OS formation under slightly acidic aerosols conditions is limited. Further sensitivity studies showed a strong dependency of the OS formation on the available sulfate supporting an acid-catalyzed processing of monoterpene epoxides yielding OS. However, it should be noted that regions with high monoterpene emissions are usually not associated with high sulfate aerosol loadings and quite acidic aerosols, hence, their contribution to SOA might be limited and important only in mixed environments.

*Other epoxides*

Other atmospheric epoxides have been proposed to contribute to SOA formation, including 2-methyl-3-buten-2ol (MBO) (Mael et al., 2015), methacrylic acid epoxide (MAE) (Lin et al., 2013; Birdsall et al., 2014), and epoxides from toluene oxidation (Baltaretu et al., 2009; McNeill et al., 2012). However, none of these species have the relatively large gas-phase production rate and water solubility of IEPOX, so they probably lead to small SOA mass contributions.

## 5.4 Oxidation reactions of acids and diacids

Acidity changes the speciation of dissociating organic compounds in the atmospheric aqueous phase. More specifically, acidity decreases the degree of dissociation for organic acids, i.e. lowers the fraction of a compound in its deprotonated form. The protonated and deprotonated forms of a dissociating compound are characterized by different molecular properties (e.g., different bond-dissociation energies (BDEs)). Therefore, key aqueous-phase oxidants, such as the radicals OH, $NO_3$ or the non-radical oxidant $O_3$, may react via different possible reaction pathways and kinetics with the protonated and deprotonated forms. Accordingly, acidity can strongly affect the chemical processing of dissociating organic compounds.



Within this subsection, the potential effect of acidity on the chemical processing of dissociating organic compounds in atmospheric aqueous solutions is summarized. The discussion will focus primarily on acids and their respective anions, however, acidity may influence reactivity and partitioning for any dissociating species (including, e.g., imidazoles or phenols).

### 5.4.1    Reaction pathways of dissociating organic compounds with different oxidants

Similar to the gas phase radical oxidants such as OH and $NO_3$ can react with dissociating organic compounds via H-abstraction.
Oxidation of dissociated organic compounds may also proceed through an electron transfer reaction (ETR). For unsaturated organic compounds, radical addition to the C=C double bond represents a third possible reaction pathway. Overviews on atmospheric aqueous-phase radical oxidants are available by Buxton et al. (1988); Herrmann (2003); Herrmann et al. (2010); Herrmann et al. (2015).

In Fig. 14, the three types of radical-initiated reactions are schematically displayed for carboxylic acids, the most prominent
dissociating organic compound class in tropospheric aerosols and clouds. The H-abstraction related reactivity of an organic molecule strongly depends on the BDE of the abstractable hydrogen atoms. Carbon-hydrogen bonds (C-H) are typically characterized by lower BDEs (e.g., BDE = $410 \pm 5$ kJ mol$^{-1}$ for acetone (Herrmann, 2003)) than other bonds such as oxygen-hydrogen bonds (e.g., O-H in acids, BDE = 445 kJ mol$^{-1}$ (Luo, 2002)). However, please note that the given BDEs are gas-phase BDEs and that BDEs can be slightly altered by an aqueous solvent. As a consequence of the weaker carbon-hydrogen bonds,
the H-abstraction reaction is currently expected to predominantly proceed at the carbon chain of dissociating organic compounds and not on the hydroxyl group, e.g. of the acid group. The H-abstraction leads to carbon-centered radicals that further react with oxygen leading to the formation of peroxy-radicals and subsequently to functionalized organic compounds with possibly changed dissociation properties. Dissociated organic compounds can also react with radical oxidants via ETR in the aqueous phase, e.g. by removing an electron from a deprotonated and ionized acid group. Particularly for more selective
radical oxidants such as $NO_3$ (or others such as $Cl_2^-$, $Br_2^-$), ETR is often preferred over H-abstraction. The reaction rate constants of $NO_3$ for ETRs are generally larger than those for H-abstraction. Overall, the different contributions of ETR and H-abstraction pathways modify also the product distributions as a function of pH.

The third mentioned pathway of radical oxidants, the addition reaction, occurs for unsaturated aliphatic and aromatic compounds. This reaction type is typically the fastest radical reaction pathway and proceeds almost at the aqueous-phase
diffusion limit (see Sect. 5.4.2), except for double bonds where the electron density is strongly reduced by electron-drawing substituents, such as halogen atoms.


**Figure 14. Schematic of the initial reaction steps for the most important radical oxidation pathways of dissociating organic compounds, exemplified for carboxylic acids.**

Besides the radical oxidation reactions, dissociating organic compounds can be also oxidized by ozone. In aqueous solutions, the decomposition of ozone is strongly affected by the acidity due to its strong chemical interaction with the water matrix (see Herrmann et al. (2015) and references therein). Ozone is known to be an electrophilic and selective oxidant for organic compounds, with particular selectivity for C=C double bonds. Therefore, ozone reacts primarily with both unsaturated aliphatic compounds and aromatic compounds. $O_3$ is also known to react slowly with saturated aliphatic compounds such as hydrated organic acids and carbonyl compounds. Rate constants for these reactions have recently been compiled in Herrmann et al. (2015).

Similar to radical oxidants, ozone reactions are expected to proceed via (i) H-abstraction (e.g. from the hydrated carbonyl groups of carboxylic acids, see Schöne and Herrmann (2014)), (ii) addition onto C=C double bonds (e.g., in case of unsaturated aliphatic compounds and aromatic compounds, see Mvula and von Sonntag (2003) and Leitzke and von Sonntag (2009)) and, finally, (iii) ETR (see Mvula and von Sonntag (2003)). However, current knowledge of the above-mentioned ozone oxidation mechanisms remains quite limited. An overview of proposed oxidation mechanisms of aqueous-phase ozone can be found, e.g. in (Hoigne and Bader, 1983a, b; Mvula and von Sonntag, 2003; Leitzke and von Sonntag, 2009; von Sonntag and von Gunten, 2012; Schöne and Herrmann, 2014) and references therein.



### 5.4.2 Comparison of kinetic data of dissociated and undissociated organic compounds

To examine the effect of acidity on the chemical processing of dissociating organic compounds, kinetic data for their oxidation by OH, NO$_3$ and O$_3$ have been newly compiled for the present review following several published review articles and data compilations (Buxton et al., 1988; Neta et al., 1988; Ross et al., 1998; Herrmann, 2003; Herrmann et al., 2010; Herrmann et al., 2015; Bräuer et al., 2019). These data are presented in Tables S5-S7 and Fig. S2-S5 in the Supporting Information. It should be noted that the Tables and Figures in the Supporting Information only show kinetic data for dissociating organic compounds where data for both their protonated and deprotonated form are available. For comparing differences in the kinetic reactivity data of protonated and deprotonated organic compounds, a reactivity ratio $\kappa_R$ has been calculated. The calculated ratios are defined as the quotient of the kinetic reaction rate constants of deprotonated and protonated form (see below) with the respective oxidant (OH, NO$_3$, O$_3$):

$$\kappa_R(i) = \frac{k_{deprotonated}^{298K}}{k_{protonated}^{298K}} \qquad (i = \text{OH}, \text{NO}_3, \text{O}_3) \tag{22}$$

In brief, values of $\kappa_R(i)$ above 1 imply that reaction of the deprotonated form will proceed more readily than the reaction of the protonated acid. Furthermore, in case of an acid, a $\kappa_R(i)$ above 1 implies that at higher pH, with an increased abundance of the deprotonated form, the overall reaction rate of a compound will be increased, i.e. oxidations are favored under decreasing acidic conditions (cf. Sect. 5.4.3).

### 5.4.2.1 OH radical oxidations

Overall, for OH reactions, the impact of acidity on the chemical kinetics is often quite small and crucial only for some specific compounds. Thus, with respect to OH reactions, acidity will mostly alter the lifetime of dissociating compounds mainly because of its impact on the partitioning and the consequently affected aqueous-phase concentrations and not so much because of changes in the OH kinetics (see also Sect. 5.4.3). Figure 15 shows the calculated ratios ($\kappa_R(OH)$) of OH reactions with several dissociating organic compounds. The calculated reactivity ratios are typically close to unity, i.e. a similar reactivity exists for the undissociated molecule and its corresponding anion. Larger ratios are calculated for a few small carboxylic and dicarboxylic acids. For formic acid and malonic acid, the $\kappa_R(OH)$ ratios are larger than 10, and for oxalic acid, they are larger than 100. For mono- and dicarboxylic acids, Fig. 15 shows decreasing ratios with increasing carbon chain. For larger carboxylic acids, calculated ratios are scattered around unity. This result indicates that the reaction mechanism of the carboxylic acid and the corresponding carboxylate is similar for the larger acids and H-abstraction is the dominant reaction pathway. With longer carbon chain, the impact of the acid functionality decreases and, thus, almost no acidity dependence exists. This also seems to be true for functionalized carboxylic acids. On the other hand, the substantially larger ratios of the smaller carboxylic acids demonstrate a much faster OH degradation of the smaller carboxylates compared to their protonated acids. This implies that the carboxylate group and their steric effects on the surrounding C-H bonds can facilitate an easier H-abstraction causing higher H-abstraction rate constants for smaller carboxylates compared to corresponding protonated acids. Besides, former



studies have shown that the ETR pathway contributes minor. Schuchmann et al. (1995) reported a contribution of less than 5%. Thus, the ETR pathway should be not responsible for the reactivity difference.

Acetic acid and acetate show the lowest OH reactivity among the considered unsubstituted monocarboxylic acids, with $1.70 \times 10^7$ L mol$^{-1}$ s$^{-1}$ and $7.30 \times 10^7$ L mol$^{-1}$ s$^{-1}$ (Chin and Wine, 1994), respectively. The weakest bond H-atoms in these molecules are part of the methyl-group. Those primary C-H bonds are much stronger than secondary or ternary C-H bonds. This explains why acetic acid is less reactive towards OH compared to higher non-substituted primary C-H bonds than monocarboxylic acids containing more CH$_2$ groups, or secondary C-H bonds. This also explains why the reactivity difference of acetic acid/ acetate is more distinct and the impact of the carboxylate group is higher compared to longer chain acids.

Moreover, its worthy to mention that the comparison of kinetic differences of protonated and deprotonated carboxylic acids given in Fig. 15 is more comprehensive compared to the work of Zhao et al. (2016). In Zhao et al. (2016), the OH oxidation kinetics of formic, glyoxylic, pyruvic acid, lactic acid, malic acid, oxalic acid has been compared. Based limited number of kinetic data and, particularly, because of the selected compounds, this study concluded that the oxidation of carboxylate forms is much more rapid compared to that of free carboxylic acid, indicating an acidity dependence in the reactivity of carboxylic

acids. However, from the present study, it can be concluded that this not true, except for smaller organic acids that are characterized by higher $\kappa_R(OH)$ ratios, due to their special structure properties, such as less or even no abstractable carbon-bonded H-atoms as in case of oxalic acid. Hence, an acidity dependence in the reactivity exists only for these smaller carboxylic acids and almost no acidity dependence exists for other carboxylic acids with a longer carbon chain. Therefore, the statements of Zhao et al. (2016) regarding the pH dependence in the reactivity of saturated carboxylic acids are by far too overgeneralized.






140

**Figure 15. Calculated reactivity ratios $\kappa_R(OH)$ of different dissociating organic compounds. The $\kappa_R(OH)$ ratio of the dianion and protonated diacid is indicated by (di). The applied aqueous-phase OH reaction rate constants are provided in Table S5 in the SI. Different colors indicate different compounds classes (such as unsubstituted saturated monoacids (red), unsubstituted unsaturated monoacids (purple), unsubstituted saturated diacids (blue), substituted saturated monoacids (green), substituted saturated diacids**
145 **(brown), phenols (light green), aromatic acids (pink), imidazoles (orange)). The dotted bars mark the ratio of the dianions.**



The rate constants for unsaturated aliphatic organic acids (protonated and deprotonated) reacting with OH are quite high, in the range of about $10^9$-$10^{10}$ L mol$^{-1}$ s$^{-1}$. The available kinetic data indicate that the considered protonated C4 unsaturated acids (methyl crotonic acid, methacrylic acid) react almost exclusively via OH-addition because of the high bond strengths for the

C-H bond in the methyl and the O-H bond in the carboxyl groups. For the smaller protonated unsaturated acids, the protonated acid group likely inhibits the OH-addition because the -COOH group lowers the electron density of the double bond and, therefore, leads to lower reaction constants. However, it can be concluded from the available kinetic dataset that the kinetic acidity effect on unsaturated aliphatic organic acids reacting mostly via OH addition generally should be small.

Aromatic compounds may also dissociate in case of side chains. Both protonated and deprotonated aromatic compounds are

characterized mostly by very high OH reaction rate constants up to about $10^{10}$ L mol$^{-1}$ s$^{-1}$ due to the preference of OH radicals to add onto the aromatic ring (Adams et al., 1965). The probability of an H-abstraction from the OH group or an ETR with deprotonated acid groups is minor. Therefore, the calculated reactivity ratios of OH reactions with aromatic compounds are around unity (0.7-2.7). This implies there is only a small kinetic acidity effect for OH reactions with aromatic compounds. A special class of aromatic compounds considered in Fig. 15 are imidazoles. In contrast to the acids that have been the primary

focus of discussion, the partitioning of imidazoles, as a base, increases with higher acidity and, therefore, greater aqueous-phase chemical processing is feasible under more acidic particle conditions. Reaction rate constants of protonated imidazoles (Rao et al., 1975; Felber et al., 2019) are approximately a factor of 2 higher than their deprotonated forms (see Fig. 15). This indicates a minor kinetic effect of acidity.

### 5.4.2.2   NO$_3$ radical oxidations

In comparison to the OH radical, the NO$_3$ radical is commonly characterized by a lower reactivity, especially for plain H-abstraction reactions (Herrmann and Zellner, 1998; Herrmann et al., 2015). However, this could be compensated through the high reactivity of NO$_3$ in single electron transfer reactions. As specifically for the acids, the compiled kinetic data demonstrate that NO$_3$ radical reaction rate constants for the reactions with undissociated and dissociated acids are often quite different (see Fig. 16). Reaction rate constants of NO$_3$ radical with saturated protonated aliphatic mono- and di-carboxylic

acids are typically in the range of $k = 10^4-10^6$ L mol$^{-1}$ s$^{-1}$, where the higher values correspond to rate constants for reactions of functionalized acids. In contrast, saturated deprotonated aliphatic mono- and di-carboxylic acids are oxidized by NO$_3$ with rate constants typically in the range of $k = 10^6-10^8$ L mol$^{-1}$ s$^{-1}$. Accordingly, the calculated reactivity ratios $\kappa_R(NO_3)$ (see Fig. 16) are often above 10 and in some cases up to $10^4$. As a consequence, acidity is a very important parameter when the reactivity of NO$_3$ in the atmospheric aqueous phase is to be described.

Compared to saturated aliphatic acids, unsaturated aliphatic acids show higher NO$_3$ reactions rate constants ($k = 10^7-10^8$ L mol$^{-1}$ s$^{-1}$). This is due to a possible addition of the NO$_3$ radical to the carbon double bond, which proceeds faster than the H-abstraction. $\kappa_R(NO_3)$ of unsaturated acids, such as acrylic and methacrylic acid, is about 6.4 and 1.8, respectively





(see Fig 16). The NO₃ addition reaction on the C=C double bond is more important than the ETR for both the undissociated and dissociated acid. Hence, the calculated reactivity ratios are smaller compared to NO₃ reactions of saturated acids.

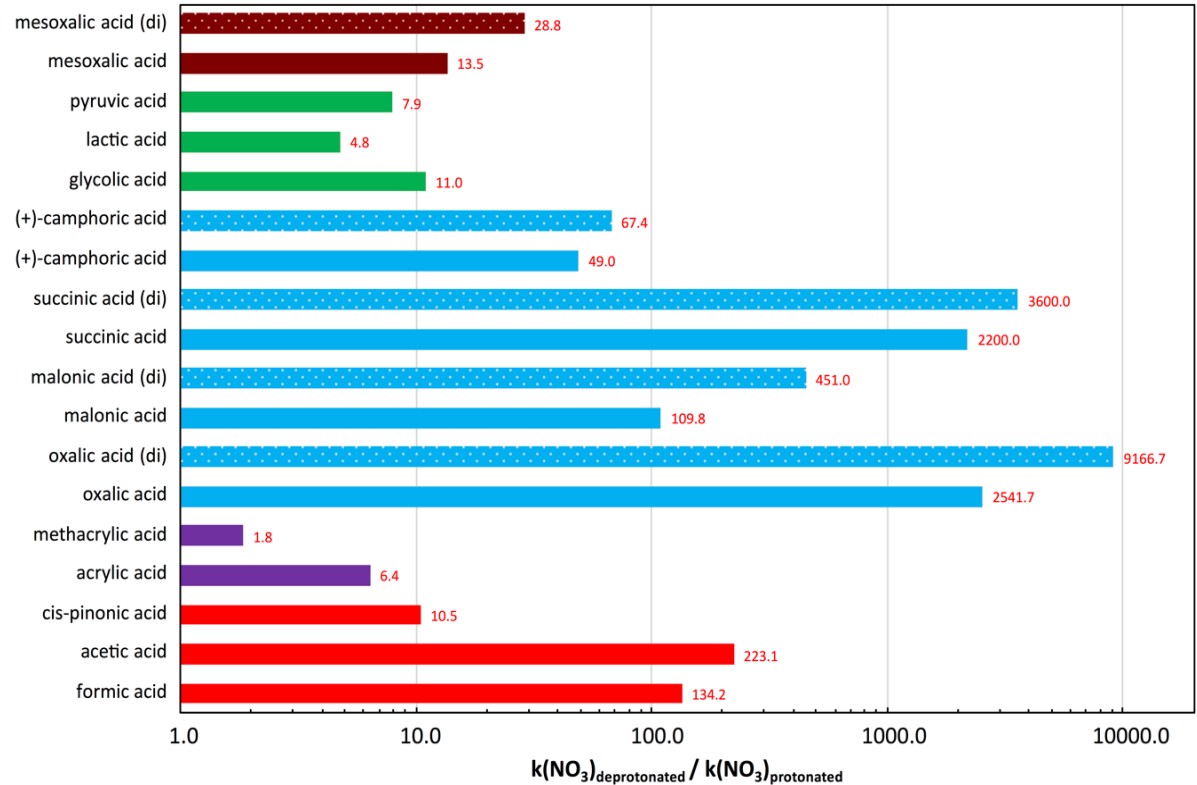

**Figure 16: Calculated reactivity ratios $\kappa_R(NO_3)$ of different carboxylic acids. The $\kappa_R(NO_3)$ ratio of the dianion and protonated diacid is indicated by the add-on (di) behind the acid name. The applied aqueous-phase NO₃ reaction rate constants are provided in Table S6 in the SI. Different colors indicate different compounds classes (such as unsubstituted saturated monoacids (red), unsubstituted unsaturated monoacids (purple), unsubstituted saturated diacids (blue), substituted saturated monoacids (green), substituted saturated diacids (brown)). The dotted bars mark the ratio of the dianions.**

Overall, it can be concluded for NO₃ in aqueous atmospheric systems, particularly clouds conditions, that acidity can substantially affect the chemical NO₃-initiated processing of organic compounds. Less acidic conditions will enhance the degradation of dissociating compounds via NO₃ because of more rapid oxidation and increased partitioning (see Sect. 5.4.3 for further details). This acidity effect could be important in urban mixed regimes, where higher NOₓ regimes mix with marine, continental dust or soil aerosols which are typically less acidic. However, due to of the sparse kinetic database for NO₃ radical reactions in the aqueous phase, more kinetic and mechanistic laboratory investigations are needed with special emphasis on acidity effects.



### 5.4.2.3 O₃ oxidations

In comparison to OH, ozone ($O_3$) is commonly known to be an electrophilic and very selective oxidant for organic compounds, covering a very wide range of reactivities. It should be noted here that lower ozone rate constants might be compensated by much higher concentrations of ozone compared to OH (Tilgner and Herrmann, 2010; Schöne and Herrmann, 2014). Comparison of the $O_3$ oxidation rates for protonated and deprotonated forms of dissociating compounds (Fig. 17) shows that $O_3$ oxidation kinetics depend significantly on acidity, especially for phenolic compounds. For saturated carboxylic acids, carboxylates demonstrate roughly a factor of 10 higher reactivity towards $O_3$ as compared to the protonated acids. This higher reactivity can be explained by the higher electron-withdrawing properties of the carboxylate. Therefore, BDEs of the carbon-bonded H-atoms are smaller and more easily abstractable. Furthermore, ETR can also occur.

As expected, compared to saturated carboxylic acids, unsaturated carboxylic acids have significantly higher reactivities with $O_3$, i.e. more than 4 orders of magnitude higher reaction rate constants. In case of unsaturated carboxylic acids, addition to the C=C double bond will establish an important reaction pathway for both the protonated and the deprotonated unsaturated acid. Nevertheless, the calculated reactivity ratios $\kappa_R(O_3)$ (see Fig. 17) show that the deprotonated unsaturated acid reacts more rapidly (1.3-25 times faster) with ozone than its protonated form. Possible reasons could be the same as for saturated organic acids. The deprotonation likely leads to a reduction in the electron density at the carbon-carbon double bond enabling an easier $O_3$ addition, i.e. a more rapid oxidation. From inductive effect theory, it is known that the COOH group is electron-withdrawing and COO⁻ is electron-donating. Thus, the obtained behavior is feasible.

A comparison of the kinetic data of maleic and fumaric acids demonstrates that the molecular structure (symmetry/bonds) strongly affects ozone reactivity. These isomers are characterized by different physical and chemical properties (dipole moments, pKₐ, reactivity). The differences in the molecular structure lead to a higher $O_3$ reactivity of fumaric acid. The C=C double bond in a fumaric acid molecule is less shielded from the two acids groups which simplifies $O_3$ addition onto the double bond. Thus, a 6 times higher reactivity of the protonated fumaric acid results compared to maleic acid.

Similar $\kappa_R(O_3)$ values are also found for aromatic acids containing unsaturated carbon side chains, e.g. cinnamic acid. On the other hand, for hydroxylated acids such as p-hydroxy benzoic acid, significantly higher reactivity differences are found with decreasing acidity. Under highly acidic conditions (pH = 2), p-hydroxy benzoic acid shows a reaction rate constant with $O_3$ of $2.0 \cdot 10^2$ L mol⁻¹ s⁻¹ (Beltran et al., 2000) whereas it reacts much more rapidly ($6.4 \cdot 10^7$ L mol⁻¹ s⁻¹, (Beltran et al., 2000)) under alkaline conditions (pH = 9). This increase can be explained by increasing deprotonation of the hydroxy group leading to formation of the phenolate form and a higher contribution of the ETR with decreasing acidity. The significantly higher reactivity of the fully deprotonated form implies that the oxidation rate at a pH of 6 is still dominated by the reaction of $O_3$ with the fully deprotonated form.



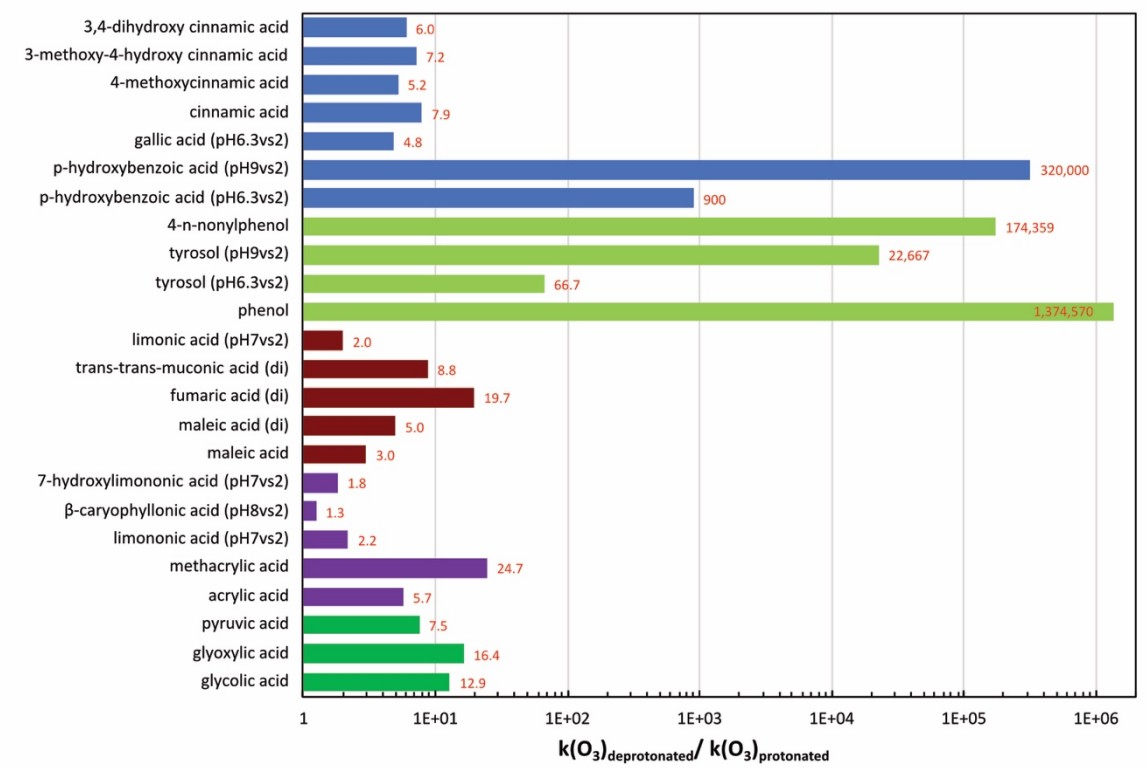

**Figure 17: Calculated $\kappa_R(O_3)$ of different dissociating organic compounds. The $\kappa_R(O_3)$ ratio of the dianion and protonated diacid is**
**indicated by the add-on (di) behind the acid name. The applied aqueous-phase $O_3$ reaction rate constants are provided in Table S7 in the SI. Different colors indicate different compounds classes (substituted saturated monoacids (green), unsaturated monoacids (purple), unsaturated diacids (brown), phenols (light green), aromatic acids (blue)).**

Rather high acidity dependencies of the reactivity data exist for phenolic compounds, too (see Fig. 17), with a large increase

of the ozone reaction rate of up to six orders of magnitude. However, it should be mentioned that the present kinetic literature

data are based on extrapolations (see Hoigne and Bader (1983b) for details). Due to the huge reactivity difference, the ozone

oxidation of phenol can be dominated by the reaction with phenolate even at neutral or slightly acidic conditions. Moreover,

charge transfer to ozone can lead to the formation of OH radical (Mvula and von Sonntag, 2003) which can initiate further

oxidation reactions. Thus, less acidic conditions can enhance the aqueous oxidation of phenolic compounds by dissolved ozone

and additionally promote further oxidation due to the initiated OH chemistry.

Overall, the present $O_3$ kinetic data analyses have demonstrated the crucial role of acidity for ozonolysis processes and, hence,

the chemical processing of dissociating compounds in tropospheric aqueous solutions. The possible formation of OH radicals

following initial ozone reactions can further enhance the oxidation capacity of the atmospheric aqueous phase. Further

laboratory measurements and modelling studies are urgently needed to improve current knowledge.





### 5.4.3 Overall considerations for the oxidation of dissociating species and the role of acidity

The overall reaction rate of a compound at a given pH depends on both the individual reaction rate constants (see above) and the degree of dissociation of the compound (which, in turn, is determined by its $pK_a$ value(s)). The rate constants of the individual free acids and their dissociated anions represent only extreme values, and the overall processing rate for a weak acid or base will often fall between these two values, depending on the pH and the $pK_a$. In order to illustrate this point, Fig. 18 shows the dependence of the overall reaction rate constant through a typical tropospheric pH range of 0 to 9 for a few selected

mono- and diacids. The overall reaction rate constant is calculated by means of the individual reaction rate constants of the protonated and deprotonated forms and their respective speciation fraction. Please note that the overall second order reaction rate constants consider the dissociation speciation of the carboxylic acids but not their effective solubility. Thus, the overall chemical reaction rate will depend on both the aqueous oxidant concentration and on the total aqueous compounds concentration. The latter largely depend on the microphysical conditions present.

Briefly, Fig. 18 demonstrates that the overall reaction rate constant can be largely pH dependent. This is particularly true for compounds with large reactivity ratios, i.e. those for which the anion is more reactive than the unionized form. For such compounds, the overall rate constant typically increases with increasing pH and more efficient oxidation can be expected under less acidic conditions. In view of decreasing inorganic acid aerosol content, together with decreasing acidity in clouds in some parts of the world (Pye et al., 2020), this might imply both stronger partitioning and more efficient oxidation of organic acids

(lower chemical aqueous-phase lifetime) in the troposphere under future conditions.

Overall, based on a compiled kinetic dataset for oxidation by OH, $NO_3$ and $O_3$ of both protonated and deprotonated organic compounds, investigations of the reactivity ratio $\kappa_R$ showed that, for OH reactions, the impact of acidity on the chemical kinetics is often quite small and only important for some specific compounds. For $NO_3$ reactions, particularly in cloud droplets, acidity can substantially affect the chemical $NO_3$-initiated processing of organic compounds and less acidic conditions will

enhance the degradation of dissociating compounds via $NO_3$ because of more rapid oxidation from an increased likelihood for Electron-Transfer-Reactions (ETRs). Furthermore, the present $O_3$ kinetic data analyses demonstrate the crucial role of acidity for ozonolysis processes, especially for phenolic compounds.





**Figure 18. Overall condensed-phase second order rate constant $k_{2nd}$ as a function of pH of selected mono- (top) and dicarboxylic (bottom) acids for different radical (OH/NO₃) and non-radical (O₃) oxidants.**



## 6  Atmospheric implications

In the review of Pye et al. (2020), a detailed compilation of acidity data measured in cloud and fog water around the globe
showed decreasing trends across North America and Europe, mainly driven by a decreased sulfate and nitrate aerosol content
due to reduced anthropogenic emissions of $SO_2$ and $NO_{x,y}$. The reduction of fossil fuel combustion emissions in a changing
world and its related feedback on acidity will have several implications on chemistry-related topics discussed in the present
review. As a similar trend in acidity of aqueous aerosols particles has not yet been widely predicted by thermodynamic models
and as observations of such a trend for aerosol particles are scarce (see Pye et al. (2020)), this section will mainly focus on
implications of changes in the acidity of cloud and fog on multiphase chemistry.

As a result of reductions in anthropogenic emissions of acid precursors in many western industrialized countries, the relative
contributions of other sources to the acidification in fog and cloud droplets will continue to grow in importance over the next
few decades unless emissions of ammonia from agricultural fertilization are simultaneously reduced. Other direct and indirect
acid sources are, for example, (1) the emission of dimethyl sulfide (DMS), (2) emission of $SO_2$ from volcanic activity, (3) sea
spray aerosol related emission of $HCl/Cl^-$ and (4) the emission and secondary multiphase formation of organic acids such as
formic and oxalic acid.

On the one hand, at pH ranges between 4 and 6, weaker acids tend to partition into less acidic cloud/fog waters more effectively
and, thus, contribute more substantially to acidity in less acidic droplet waters. On the other hand, less acidic cloud and fog
water pH values are in the typical range of the $pK_a$ values of weak acids such as acetic acid ($pK_a = 4.75$), so that they can
efficaciously buffer acidification by stronger acids in this acidity range (see Sect. 2.2). As a consequence, the increased
aqueous-phase partitioning enables higher chemical processing rates of weak acids such as $SO_2$, HONO and organic carboxylic
acids. Both lower acidity and stronger buffering can support faster S(IV) to S(VI) conversions due to the higher efficiency of
other chemical pathways such as ozone oxidation (Li et al., 2020b) and therefore reduce the tropospheric lifetime (and in-
cloud lifetime) of $SO_2$. Therefore, under future conditions with a lower overall $SO_2$ budget, the increased secondary sulfate
mass formation probabilities may compensate at least partly the reduced sulfate formation potential. In the case of organic
acids, the increased in-cloud partitioning allows them greater opportunities for chemical processing, leading to higher
formation yields of functionalized organic acids which tend to partition even stronger towards the aqueous phase of particles
and droplets. Hence, higher in-cloud SOA formation yields can be expected as a consequence of the lower acidification of
cloud and fog waters by anthropogenic sulfate.

Having affected in-cloud chemistry processes, the decreasing $SO_2$ budget will also presumably influence the isoprene-related
SOA formation, particularly the OS formation. Here, several projection studies (Pye et al., 2013; Marais et al., 2016;
Budisulistiorini et al., 2017; He et al., 2018b; Zhang et al., 2018a) have proposed a reduced IEPOX-derived SOA formation
under reduced $SO_2$ emissions. Also, studies for the SE-US (Pye et al., 2013; Marais et al., 2016; Budisulistiorini et al., 2017)
implied that a reduction of $SO_2$ by 25-48% led to reduction of the IEPOX-derived SOA formation of about 35-70%. This effect
is mainly related to the changes in aerosol acidity but could be further modulated by the resulting changes in particle viscosity





and phase separation that result from extensive conversion of inorganic to organic sulfur expected with declines in $SO_2$ (Riva et al., 2019). For the Pearl River Delta region, a reduction of ~ 45% of the IEPOX-derived SOA was reported by He et al. (2018b) due to an aerosol sulfate reduction by 25%. Finally. all studies clearly demonstrated that a $SO_2$ emission decline in polluted regions could significantly lower the isoprene-related SOA. Similar effects can also be assumed for other acid-catalyzed or acidity-dependent processes.

Another chemical subsystem that will be likely affected by reduced anthropogenic acid precursor emissions in the future is TMI solubilization (see Pye et al. (2020) and references therein). The smaller possible acidification of aqueous interfacial layers on crustal aerosols can lower the acid-driven solubilization of TMI, particularly in regions where dust particles are mixed with urban pollutants. The decreased formation of soluble and, hence, bioavailable TMIs can (1) cause lower nutrient inputs into oceans impacting the ocean biological activity there, (2) decrease the chemical $HO_x$ radical cycling in both aqueous particles and droplets and (3) might also affect the TMI-related S(IV) oxidation.

Decreasing atmospheric acidity may also impact the acidity-driven production of reactive halogens, with potential implications for ozone and OH. Observations of sea-salt aerosol bromide and chloride deficits over the northeastern Pacific Ocean revealed that depletions in bromide and chloride relative to seawater were correlated with particle acidity (see e.g., Newberg et al. (2005)).

In order to explore the expected changes in the tropospheric multiphase chemistry and their overarching impacts in a changing environment, further field measurements, laboratory studies and accompanied modeling are needed to both monitor occurring changes and improve air quality and climate model predictions.



## 7 Conclusions and outlook


In the present review, we have outlined different aspects and chemical subsystems where acidity affects multiphase chemistry and, in turn, acidity is affected by tropospheric multiphase chemistry. Although many advances have been made in our understanding of acidity-driven and acid-catalyzed chemical processes, there are still many open issues which need to be addressed in order to further advance our understanding of the complex role acidity in the atmosphere. Besides the implications

caused by changing acidity conditions in the atmosphere (cf. Sect. 6), the present review has also identified chemistry-related research targets and needs for further investigation that are outlined below. Specifically, these chemistry-related future research needs and objectives are:

(1)    Advance our understanding of the activation mechanisms and multiphase chemical processing of reactive halogen species in different acidity environments and quantify their presence in and above the tropospheric boundary layer

(2)    Undertake more sophisticated field, laboratory and model investigations on the importance of different S(IV) oxidation pathways under various acidity conditions in different environments, particularly under conditions typical of aerosols (lower pH, high ionic strength)

(3)    Advance our understanding of aqueous-phase organic accretion reactions including their dependence on acidity in order to assess their role for the secondary aerosol formation in clouds and aqueous particles

(4)    Perform advanced kinetic and mechanistic studies on acidity-dependencies of aqueous-phase organic oxidations

(5)    Quantify the role of organosulfates (OS)/ organonitrates (ON)/ nitrooxy organosulfates (NOS) as potential acidity reservoir or sink and characterize the role of acidity for their formation and fate in aerosols

(6)    Investigate the impact of particle acidity in the formation and early growth of tropospheric nanometer-sized particles from highly-oxidized molecules (HOMs)

(7)    Improve size/time-resolved cloud/fog measurement techniques and develop in situ measurements techniques for directly determining aerosol acidity, burdens which still limits our knowledge about the impact of acidity on the multiphase chemistry in different droplet and aerosol sizes and their feedbacks on chemical composition

(1) While much effort has been devoted to investigations of reactive halogen chemistry in pristine open ocean regions and
partly coastal areas (see Sect. 4.8 for details), the impact of reactive halogens on atmospheric chemistry in developing countries is less examined. Especially in developing economies such as China and India, where a substantial amount of the air pollution is related to coal combustion and other biomass burning, a significant fraction of the aerosol matter consists of halogens related to such sources (Goetz et al., 2018; Gani et al., 2019). Further studies are needed to investigate the role of halogen chemistry in strongly polluted environments which are characterized by very acidic particles compared to marine environments. Under
very polluted conditions, high acidity linked with high $NO_{x,y}$ can cause active halogen radical chemistry that might influence the tropospheric cleaning capacity. However, the role of multiphase halogen chemistry in such environments is still not well investigated. For example, recent studies of Wang et al. (2019b) have demonstrated that current models are not able to





reproduce high observed Cl₂ in daytime in continental regimes. Furthermore, recent model studies of Zhu et al. (2019) also reported overestimates of free-tropospheric BrO during the extratropical winter–spring. Moreover, our understanding of

halogen activation processes in Arctic regions needs further improvement. Arctic regions are undergoing unprecedented climate changes, most likely with substantial changes in the aerosol composition that can affect the aerosol acidity and consequently halogen activation processes. So, further research is needed to focus on how Arctic climate changes will impact halogen chemistry in this dynamic and sensitive environment.

(2) Comparisons of model findings with field measurements in polluted regions have shown that current models often

underestimate the S(VI) formation rates or cannot reproduce the findings of sulfur-isotope measurements regarding the responsible oxidation pathways. Hence, the current chemical kinetic and mechanistic understanding of the S(IV) to S(VI) conversion processes, including their acidity dependency, is still incomplete to adequately predict the budgets of S(VI) in cloud, fog, and especially aqueous aerosol conditions. As multiphase chemistry models rely on detailed acidity-dependent kinetic and mechanistic knowledge, further laboratory studies are indispensable to improve model predictions of S(VI),

particularly under conditions of high ionic strength (e.g. aerosol chamber or aerosol flow tube studies).

(3) Non-oxidative aqueous organic chemical process, such as accretion reactions (aldol condensation, hemiacetal as well as acetal formation and the esterification of carboxylic acids), are affected by acidity/basicity of the solution and are expected to be important formation pathways of aqSOA. However, the potential role of such acidity-related processes in tropospheric aqueous solutions is still not fully explored yet since mechanistic and particularly kinetic data on acid-catalyzed accretion

reactions in aerosols are still sparse (see Herrmann et al. (2015)). So, these acidity-related processes should be a key objective of future laboratory and chamber studies towards a better representation of such processes in detailed chemistry mechanisms and models.

(4) More advanced kinetic and mechanistic studies on acidity-dependencies of aqueous-phase oxidations of dissociating organic compounds, such as functionalized carboxylic acids, are needed to better describe such processes in future multiphase

models and to, finally, elucidate their impacts. Compared to the huge number of functionalized organic dissociating organic compounds that are formed by various multiphase reaction pathways, the body of investigated aqueous-phase oxidations are still rather small. Furthermore, existing estimation methods for aqueous-phase kinetic reaction rate constants are either characterized by large uncertainties particularly for functionalized compounds (Bräuer et al., 2019) or don't exist because of the sparse dataset available. Even though the present review showed that acidity could play an important role for ozonolysis

processes of dissociating compounds in tropospheric aqueous solutions, nevertheless, the kinetic and mechanistic knowledge of such oxidation processes including the possible formation of OH radicals is currently rather limited. Further kinetic and mechanistic laboratory investigation are urgently needed to minimize the current enormous knowledge limitations and uncertainties, for example, with regards to the production of organic mono- and dicarboxylic acids as well as their functionalized derivatives.

(5) Organosulfates (OSs) are ubiquitous constituents of atmospheric aerosol particles that not only contribute substantially to OM, but may also bind a considerable portion of the sulfate content of atmospheric particles (Riva et al., 2019; Brüggemann



et al., 2020). For this reason, these compounds have the potential to reduce the free sulfate, and consequently the $H^+$ formation potential, with major implications for aerosol acidity. Overall, OSs may be a temporary acidity reservoir due to the binding and release of sulfate, or a sink of $H^+$ if $H^+$ is incorporated into the OS and unavailable for further processing. While progress has been achieved in the understanding of OS formation pathways in the last decade, the scientific understanding of their chemical processing in aqueous aerosols is still uncertain. In order to elucidate the role of OSs to act as crucial acidity reservoir/buffer in atmospheric particles, more detailed knowledge of their chemical processing is highly desirable. Thus, the chemical transformations of OSs through hydrolysis and oxidations by atmospheric radicals (e.g., OH, $NO_3$, Cl etc.) require more kinetic and mechanistic laboratory investigations as reaction data are usually not yet available. Additionally, similar investigations on the formation and transformation must be performed for organonitrates (ONs) and nitrooxy organosulfates (NOSs) which could also potentially act as nitrate/sulfate reservoirs, and thus affect the acidity.

(6) Historically, sulfuric acid has been considered the driver of new particle formation events (e.g. Sipila et al. (2010)). However, several studies (see Lee et al. (2019) and references therein) have shown that the gas-phase oxidation of biogenic and anthropogenic VOCs, via autoxidation reactions, can produce highly-oxidized organic molecules (HOMs). HOMs are characterized by extremely low saturation vapor pressures and can effectively condense on nanometer-sized aerosol, contributing there to the early growth of particles. Due to their high degree of oxygenation, including several chemical functional groups (e.g., peroxide, hydroperoxide, carbonyl, and carboxylic acid) that are sensitive to acidity, they can undergo chemical reactions there which can enhance their uptake and contribution to the particle growth. As a result, one possible chemical pathway would be the formation of HOOS (highly oxidized OSs). Having first been reported by Mutzel et al. (2015), they potentially serve as an example to better explain the early growth of freshly formed particles. Such acidic aerosols might provide a chemical environment where extremely low volatility compounds can condense and react, leading to the formation of HOOSs. Elucidating the role of acidity for the formation of SOA in such small particles and their importance for early nanoparticle growth will be a crucial objective for future field and chamber studies.

(7) Because aerosols are ubiquitous in the atmosphere, and clouds cover 60% of the Earth, understanding their contribution to tropospheric composition and how it is evolving, is crucial. Here, the present review has shown the role of acidity in determining aqueous chemistry and vice versa, yet several issues demand more advanced field, kinetic laboratory, chamber and modeling studies. Armed with this new fundamental knowledge, better predictions can be made of aerosol and cloud/fog chemistry on tropospheric oxidizing capacity, air quality, climate, and human health. For example, to better characterize the effect of varying acidity on chemical processing and the identification of potential changes due to changing anthropogenic emissions, further advances in measuring the chemical composition of cloud/fog in a both size- and time-resolved manner is needed. The lack of measurement and analytical techniques for directly determining aerosol acidity in situ is even more urgent.



*Data availability.* All data used for the figures are provided in the Tables and supplement.

*Supplement.* The supplement related to this article is available online at:

The supplement includes additional documentation on available hydration equilibrium constants $K_{hyd}$ and compiled kinetic data of OH, $NO_3$ and ozone reactions of dissociating organic compounds. In addition, figures further displaying differences in reaction rate constants of protonated and deprotonated organic compounds are shown. Data used as input, to create the partitioning and speciation plots (Fig. 1 and 2) are available in the supplement.

*Author contributions.* VFM and HH coordinated the manuscript preparation and AT was the lead author. HOTP and AN designed the overall scope of this study. AT, TS, BA, MB, JLC, KLF, HH, AN, HOTP, and VFM wrote the manuscript. All authors prepared text, figures, and tables in collaboration.

*Competing interests.* The authors declare no conflicts of interest.


*Disclaimer.* The U.S. Environmental Protection Agency through its Office of Research and Development collaborated in the research described here. The research has been subjected to Agency administrative review and approved for publication but may not necessarily reflect official Agency policy. The views expressed in this article are those of the authors and do not necessarily represent the views or policies of the U.S. Environmental Protection Agency.


*Acknowledgements.* We thank the EPA for funding and hosting a workshop on the State of Acidity in the Atmosphere: Particles and Clouds. We thank Ken Elstein, Brooke Hemming, and Randa Boykin for their assistance during the workshop. This work has received funding from the European Union's Horizon 2020 research and innovation program through the EUROCHAMP-2020 Infrastructure Activity under grant agreement no. 730997. Support for TROPOS-ACD by EFRE
(Contract 3000582010) is acknowledged.

*Financial support.* We thank the EPA for funding and hosting the workshop *The State of Acidity in the Atmosphere: Particles and Clouds* and Ken Elstein, Brooke Hemming, and Randa Boykin for their assistance during the workshop. AN was supported by the project PyroTRACH (ERC-2016-COG) funded by H2020-EU.1.1. – Excellent Science – European Research Council (ERC), project ID 726165. The work by JC was supported by grant number NSF-AGS- 1650786.





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



**Table 1:** **Composition conditions applied for the calculation of the S(IV) oxidation rates of different reaction pathways for urban haze and rural aerosol conditions as well as urban and rural cloud conditions (bottom) at 298 K.**

| Reactant | Concentration | | | |
|---|---|---|---|---|
| | **Urban haze** | **Rural aerosol** | **Urban cloud** | **Rural cloud** |
| $SO_2$ / ppb | $40.0^\#$ | $1.0^\dagger$ | $5.0^\infty$ | $1.0^\dagger$ |
| $H_2O_2$ / ppb | $0.1^\ddagger$ | $0.1^\dagger$ | $1.0^\infty$ | $0.1^\dagger$ |
| $O_3$ / ppb | $1.0^\#$ | $30.0^\dagger$ | $50.0^\infty$ | $30.0^\dagger$ |
| $NO_2$ / ppb | $66.0^\#$ | $1.0^\dagger$ | $10.0^\dagger$ | $1.0^\dagger$ |
| $HNO_4$ / ppb | $0.01^\dagger$ | $0.001^\dagger$ | $0.01^\dagger$ | $0.001^\dagger$ |
| $CH_3OOH$ / ppb | $0.1^\dagger$ | $0.1^\dagger$ | $0.1^\dagger$ | $0.1^\dagger$ |
| $CH_3C(O)OOH$ / ppb | $0.1^\dagger$ | $0.1^\dagger$ | $0.1^\dagger$ | $0.1^\dagger$ |
| $Fe(III)$ / mol $L^{-1}$ | $1.1\times10^{-3}$ $^\#$ | $1.0\times10^{-3}$ $^\infty$ | $1.0\times10^{-5}$ $^\S$ | $1.0\times10^{-6}$ $^\S$ |
| $Mn(II)$ / mol $L^{-1}$ | $2.55\times10^{-3}$ $^\#$ | $1.0\times10^{-4}$ $^\infty$ | $1.0\times10^{-6}$ $^\S$ | $1.0\times10^{-7}$ $^\S$ |
| PS* / mol $L^{-1}$ | $1.9\times10^{-11}$ | $6.3\times10^{-11}$ | $1.2\times10^{-12}$ | $3.6\times10^{-13}$ |
| Ionic strength mol $L^{-1}$ | $1.0^\dagger$ | $1.0^\dagger$ | $1.0\times10^{-4\dagger}$ | $1.0\times10^{-4\dagger}$ |

Remarks: $^\infty$ based on Seinfeld and Pandis (2006), $^\#$ based on Cheng et al. (2016), $^\S$ estimated from data given in Deguillaume et al. (2005), $^\ddagger$ based on (Ye et al., 2018a), $^\dagger$ estimated, daytime particle/cloud mean concentrations based on simulations using CAPRAM mechanism (Bräuer et al., 2019; Hoffmann et al., 2020). Further simulation details are given in the SI.





**Table 2: Influence of acidity on the hydration rate of formaldehyde and acetaldehyde in buffered solutions.**

| Acidic species | Acid dissociation constant of the catalyst acid-base pair $K_a$ / (unitless) | Acid catalytic constant $k_a$ / L mol$^{-1}$ s$^{-1}$ | Base catalytic constant $k_b$ / L mol$^{-1}$ s$^{-1}$ |
|---|---|---|---|
| *Formaldehyde*[*] | | | |
| $H_3O^+$ | 55.5 | 2.7 | 0.0051 |
| Formic acid | $1.77 \times 10^{-4}$ | 0.070 | 0.013 |
| Phenylacetic | $4.88 \times 10^{-5}$ | - | 0.015 |
| Acetic acid | $1.75 \times 10^{-5}$ | 0.043 | 0.022 |
| Trimethylacetic acid | $8.9 \times 10^{-6}$ | 0.025 | 0.022 |
| Water | $1.8 \times 10^{-16}$ | 0.0051 | 1600 |
| *Acetaldehyde*[#] | | | |
| $H_3O^+$ | 55.5 | 930 | 0.00014 |
| Formic acid | $1.77 \times 10^{-4}$ | 1.74 | 0.065 |
| Phenylacetic | $4.88 \times 10^{-5}$ | 0.91 | 0.054 |
| Acetic acid | $1.75 \times 10^{-5}$ | 0.47 | 0.157 |
| Trimethylacetic acid | $9.4 \times 10^{-6}$ | 0.33 | 0.161 |
| Water | $1.8 \times 10^{-16}$ | 0.00014 | $8 \times 10^4$ |
| [*]Bell and Evans (1966), [#]Kurz (1967); Kurz and Coburn (1967); Ogata and Kawasaki (1970) | | | |