# Peer review of "Acidity and the multiphase chemistry of atmospheric aqueous particles and clouds"

_Atmospheric Chemistry and Physics, 2021_

## Referee Comment (RC1)

This review article written by Tilgner et al. nicely summarizes the fundamentals and recent literature on how acidity affects chemical partitioning and reactions occurring in the atmospheric aqueous phases. The topic is important and timely. The review covers all the important aspects of this topic, and the manuscript is extremely carefully written. I strongly recommend publication in ACP. I only have a number of minor comments listed below.

Minor comments
- A class of aqueous-phase reactions that are strongly pH-dependent is hydrolysis. Hydrolysis can be a major fate of many environmental pollutants in aquatic chemistry. While the manuscript contains a few discussions related to hydrolysis (e.g., sections 5.2.3 and 5.3), I suggest the authors considered having a section to systematically discuss hydrolysis. E.g., what functional groups can undergo hydrolysis and how; pH-dependence; the importance of hydrolysis in the atmosphere compared to other reactions already described in the manuscript.
- Figure 2-f. I suggest the authors added pKa2 for phthalic acid. I wonder why the aqueous fraction (XAaq) does not immediately increase at the pKa1 (~3) of phthalic acid? This trend seems to be different from the other monoacids.
- Figure 7a) and b) - the kinks on the total S(IV) oxidation rates at pH 5 appear very unnatural, and there are no explanations provided. My understanding is that the figure is based on the recommended S(IV) oxidation by Fe(III), as indicated in equation 11. I recommend that the authors explain why the Fe(III) oxidation drastically loses importance at pH > 5.
- Line 1207 "The deprotonation likely leads to a reduction in the electron density at the carbon-carbon double bond enabling an easier O3 addition, i.e. a more rapid oxidation. From inductive effect theory, it is known that the COOH group is electron-withdrawing and COO- is electron-donating. Thus, the obtained behavior is feasible." - This couple of statements seem to be conflicting with each other. Please clarify.
- Line 1246 "Please note that the overall second order reaction rate constants consider the dissociation speciation of the carboxylic acids but not their effective solubility. Thus, the overall chemical reaction rate will depend on both the aqueous oxidant concentration and on the total aqueous compounds concentration. The latter largely depends on the microphysical conditions present." - I agree with the author's statements. Can they comment on how important O3 and NO3 oxidation reactions are, considering their relatively small Henry's law constant? Or may their Henry's law constants are not so small?
-

Technical comment
- Figure 4. Can the authors consider adding pKa values of each compound, in a similar manner as in Figure 2?
- Figure 5 caption. The authors defined the red text and green text in the figure, leaving the blue ones unexplained. Maybe add that blue text describes physical changes of droplets?
- Line 358 - "sulfate oxidation" should be "S(IV) oxidation"

- Line 572 - "newly kinetically analyzed for the present review" sounds awkward. Please rephrase.
- Line 599 - Lee and Schwartz 1983 the citation format is inconsistent.
- Line 796 - extra "." at the end
- Line 802 - "substituted organic acid" - should it be "substituted aldehyde"?
- Line 820  - "hydrolysis" should be "hydration"
- Figure 12 caption - This caption uses "Fig. XX", while others all use "Figure XX"
- Line 886 - the phase "which was discussed in more detail for acetaldehyde, glyoxal, as well as methylglyoxal" is repeated.
- Line 908 - "On the one hand"
- Line 917 - Li et al. 2019 - citation format is inconsistent.
- Line 1065 - "Overall, the different contribution of ETR and H-abstraction pathways modify also the product distribution as a function of pH." Does the author have any references for this statement?
- Line 1138 - "Therefore, the statements of Zhao et al. (2016) regarding the pH dependence in the reactivity of saturated carboxylic acids are by far too overgeneralized)" - the same authors have a recent paper showing that larger organic acids indeed do not exhibit much pH dependence. Please cite.
  https://doi-org.login.ezproxy.library.ualberta.ca/10.1021/acs.est.0c03331
- Line 1190 - "due to of the"
- Line 1249 - "depend" should be "depends"
- Line 1386 - Just wanted to double check if the authors have defined "OM".

---

## Author Comment (AC1)

**CC1**: 'Comment on acp-2021-58', Krzysztof Rudzinski, 09 Feb 2021

The authors thank Krzysztof Rudzinski for his active contribution in the review process and his comments which helped to improve the manuscript. In the following, we provide a response to your comment. Comments by the reviewer are given in black normal font, and our response to the comments is shown in blue. Newly added and modified text in the revised manuscript and supporting information (SI) is given in *italics.*

**CC1-C1:** This is a very interesting paper tackling an important aspect of atmospheric chemistry.

I would like to draw the authors' attentiion to our paper on isoprene oxidation coupled with manganese catalyzed autoxidation of sulfite (1), which is matching section 4.4 . That paper shows the influence of acitidy on the rate of autoxidation and on the rate of isoprene conversion based on experimental observation and detailed kinetic modelling. The kinetic model used includes autoxidation of Mn(II), autoxidation of S(IV) catalyzed by Mn and oxidation of isoprene. It explains the observed influence of pH on the reactions (with and without isoprene present) better than empirical rate equaitions shown in section 4.4.
Our another paper (2) describes the aqueous-phase transformation of isorene in the presence of HONO. Also in this case the acidity of solutions had significant effect on the isoprene conversion demonstrated by experiments and explained by a model. However, in this case the role of acidity was simple - conversion of isoprene required undissociated HONO present.
The authors might consider using the quoted papers as a minor illustration to their valuable lecture.
with best regards

Krzysztof J. Rudziński

(1) Rudziński KJ, Gmachowski L, Kuznietsova I (2009) Reactions of isoprene and sulphoxy radical-anions – a possible source of atmospheric organosulphites and organosulphates Atmos Chem Phys 9:2129-2140 doi:10.5194/acp-9-2129-2009
(2) Rudziński KJ, Szmigielski R, Kuznietsova I, Wach P, Staszek D (2016) Aqueous-phase story of isoprene – A mini-review and reaction with HONO Atmos Environ 130:163-171 doi:http://dx.doi.org/10.1016/j.atmosenv.2015.12.027

**CC1-R1:** According to the posted comment, we have cited the work of Dr. Rudziński on the S(IV) oxidation catalyzed by Mn(II) in section 4.4. The other mentioned publication (reference 2) has not been included as the indirect pH-dependency results just from the presence of $HONO/NO_2^-$ under different acidity conditions.

*"A detailed discussion of the mechanisms can be found in Brandt and van Eldik (1995) and Rudziński et al. (2009)".*

**References:**

Brandt, C., and van Eldik, R.: Transition-Metal-Catalyzed Oxidation of Sulfur(Iv) Oxides - Atmospheric-Relevant Processes and Mechanisms, Chem. Rev., 95, 119-190, https://doi.org/10.1021/cr00033a006, 1995.
Rudziński, K. J., Gmachowski, L., and Kuznietsova, I.: Reactions of isoprene and sulphoxy radical-anions – a possible source of atmospheric organosulphites and organosulphates, Atmos. Chem. Phys., 9, 2129-2140, https://doi.org/10.5194/acp-9-2129-2009, 2009.

---

## Author Comment (AC2)

We thank reviewer#1 for the many constructive comments and suggestions which, we think, helped a lot to improve the manuscript. In the following, we provide a point-by-point response to all comments. In order to improve readability, we numbered each reviewer comment and its corresponding response in the style R1-C1 for reviewer comments (C1 means comment 1 of reviewer 1) and R1-A1 for answers to the reviewer comment (A1 means answer to comment 1 of reviewer 1), respectively. Comments by the reviewer are given in black normal font, and our response to the comments is shown in blue. Newly added and modified text in the revised manuscript and supporting information (SI) is given in *italics.* Finally, we would like to mention that the manuscript was checked by an EPA internal review which also caused certain small changes in the revised manuscript (please see the manuscript with tracked changes).
* * *
**RC1**: 'Comment on acp-2021-58', Anonymous Referee #1, 27 Feb 2021

This review article written by Tilgner et al. nicely summarizes the fundamentals and recent literature on how acidity affects chemical partitioning and reactions occurring in the atmospheric aqueous phases. The topic is important and timely. The review covers all the important aspects of this topic, and the manuscript is extremely carefully written. I strongly recommend publication in ACP. I only have a number of minor comments listed below.

Minor comments

**R1-C1:**  A class of aqueous-phase reactions that are strongly pH-dependent is hydrolysis. Hydrolysis can be a major fate of many environmental pollutants in aquatic chemistry. While the manuscript contains a few discussions related to hydrolysis (e.g., sections 5.2.3 and 5.3), I suggest the authors considered having a section to systematically discuss hydrolysis. E.g., what functional groups can undergo hydrolysis and how; pH-dependence; the importance of hydrolysis in the atmosphere compared to other reactions already described in the manuscript.

**R1-A1:**  We fully agree with the reviewer that hydrolysis processes can be strongly pH-dependent and an important tropospheric multiphase process. Therefore, hydrolysis processes and their pH-dependency were already comprehensively discussed in former reviews, e.g., by Herrmann et al. (2015), Ng et al. (2017) and Brüggemann et al. (2020). As a consequence, hydrolysis processes are not explicitly treated in the present paper and only addressed in sections 5.2 and 5.3. However, according to the reviewer's comment, we have looked again through our manuscript and included some links to the aforementioned papers in sections 5.2 and 5.3. The revised manuscript now refers the reader more directly to the comprehensive reviews for a more detailed overview on this specific topic.

The revised text parts are as follows:

*"For further specific details on organic accretion reactions and other linked important pH-dependent reactions of organic compounds, such as hydrolysis reactions, please see Larson and Weber (1994) Herrmann et al. (2015), Zhao et al. (2016), Ng et al. (2017) and Brüggemann et al. (2020) and references therein."*

*"Both the formation and the hydrolysis of esters are slow processes under tropospheric conditions (see Herrmann et al. (2015) for further details). Moreover, the hydrolysis rate of esters will increase with increasing acidity (Mabey and Mill, 1978; Herrmann et al., 2015)."*

**R1-C2:**  Figure 2-f. I suggest the authors added pKa2 for phthalic acid. I wonder why the aqueous fraction (XAaq) does not immediately increase at the pKa1 (~3) of phthalic acid? This trend seems to be different from the other monoacids.

**R1-A2:** We thank the reviewer for this careful look at the Figure. In fact, there was a problem in the GNUPLOT script of phthalic acid. Thus, thus Figure 2 was revised and for Figure 2-f the missing $pK_{a2}$ value was added. The revised Figure 2 is now as follows:

[Figure]

**R1-C3:**  Figure 7a) and b) - the kinks on the total S(IV) oxidation rates at pH 5 appear very unnatural, and there are no explanations provided. My understanding is that the figure is based on the recommended S(IV) oxidation by Fe(III), as indicated in equation 11. I recommend that the authors explain why the Fe(III) oxidation drastically loses importance at pH > 5.

**R1-A3:**  We thank the reviewer for the comment. In fact, the S(IV) oxidation rates appear a bit unnatural. For Figure 7, we have applied the recommended S(IV) oxidation by Fe(III) of Hoffmann and Calvert (1985) reported in equation 11b. This rate expression is only valid for pH conditions < 5 causing this unnatural pattern. However, the efficiency of iron(III)-catalyzed oxidation of S(IV) to S(VI) strongly depends on speciation of iron(III), i.e., the concentration of inorganic and organic complexing agents (Deguillaume et al. (2005)) which was not considered in the rate inter-comparison. At higher pH values, the $pK_a$ values of important complexing agents are exceeded. So, they are present in their dissociated form enabling to a stronger iron(III) complexation and inhibiting the iron-catalyzed S(IV) oxidation. This strong inhibiting effect on iron(III)-catalyzed S(IV) oxidation is well-known for example for organic acids such as oxalate (see e.g. Grgić et al. (1998)). Thus, the iron(III)-catalyzed S(IV) oxidation drastically loses importance at pH > 5 ($pK_a$ values of organic acids are typically < 5).

Based on the reviewer suggestion, the manuscript text was extended to better explain the pattern in Figure 7a) & b) and to explain why the iron(III)-catalyzed S(IV) oxidation drastically loses importance at pH values above 5 in atmospheric solutions where substantial amounts of dissolved organic matter are ubiquitous and a stronger partitioning of dissociating organic compounds occurs.

The revised description and discussion of Figure 7 in the text now reads as follows:

[revised manuscript text omitted]

**R1-C5:** Line 1246 "Please note that the overall second order reaction rate constants consider the dissociation speciation of the carboxylic acids but not their effective solubility. Thus, the overall chemical reaction rate will depend on both the aqueous oxidant concentration and on the total aqueous compounds concentration. The latter largely depends on the microphysical conditions present." - I agree with the author's statements. Can they comment on how important O3 and NO3 oxidation reactions are, considering their relatively small Henry's law constant? Or may their Henry's law constants are not so small?

**R1-A5:** This subsection was intended to illustrate the pH-dependency of the overall rate constant based on several examples. The reviewer questions regarding the importance of the different radical oxidants is well justified. However, besides the solubility (Henry's law constant) also the gas-phase oxidant concentration is important in order to judge whether an oxidant is crucial or not. Moreover, these concentrations strongly depend on the environment.

Because of the acidity focus of this review, we have not accentuated on the oxidant importance topic in this review and the comparison of different oxidants. However, former studies did this already, please see Tilgner & Herrmann (2010) and Schöne & Herrmann (2014). Moreover, kinetic reviews by Herrmann et al. (2010) and Herrmann et al. (2015) reported already that aqueous-phase concentrations of OH and $NO_3$ are quite similar but the aqueous-phase concentration of ozone is typically 5 order of magnitude higher. So, the lower reactivity of ozone can be fully compensated.

Therefore, Herrmann et al. (2015) concluded already that "$O_3$ reaction rate constants have to be on the order of about $10^3$–$10^5$ $M^{-1}$ $s^{-1}$ to be competitive with chemical conversions initiated by OH radicals". We hope this is addressing this specific well-taken reviewer comment.

Technical comments

**R1-C6:** Figure 4. Can the authors consider adding pKa values of each compound, in a similar manner as in Figure 2?

**R1-A6:** We thank the reviewer for this valuable comment and we agree that the addition of the $pK_a$ values much improves the interpretation and the applicability of the buffering capacity plots. Accordingly, we have included the $pK_a$ values of each buffer compound with a purple dashed line and updated the caption of Figure 4. The revised Figure 4 and corresponding caption is as follows:

[Figure]

*Figure 4.    Buffering capacity $\beta$ of water, ammonia/ammonium and carbonate/bicarbonate/carbonic acid (top: from left to right) as well as formic and acetic acid (bottom: from left to right) as a function of pH. The atmospherically relevant range of cloud and aerosol pH is marked in yellow, and the $pK_a$ values of the corresponding buffers are marked with dotted pink lines.*

**R1-C7:** Figure 5 caption. The authors defined the red text and green text in the figure, leaving the blue ones unexplained. Maybe add that blue text describes physical changes of droplets?

**R1-A7:** The reviewer is right. We have added an explanation to the caption which reads now as follows:

*"Figure 5.  Schematic of sources (red text) and conditions of acidity in different aqueous aerosol particles (green text) together with microphysical and chemical processes that are able to influence the acidity of tropospheric aerosols (Fig. created after Raes et al. (2000) and McMurry (2015)). The blue text describes microphysical processes of CCNs and cloud/fog droplets. The dashed gray line represents an aerosol number size distribution based on McMurry (2015). "*

**R1-C8:** Line 358 - "sulfate oxidation" should be "S(IV) oxidation"

**R1-A8:** The reviewer is right. The text has been revised.

**R1-C9:** Line 572 - "newly kinetically analyzed for the present review" sounds awkward. Please rephrase.

**R1-A9:** The sentence was rephrased and reads now as follows:

*"For this review, the kinetic data of Spindler et al. (2003) have been again kinetically analyzed in more detail."*

**R1-C10:** Line 599 - Lee and Schwartz 1983 the citation format is inconsistent.

**R1-A10:** The citation format has been changed.

**R1-C11:** Line 796 - extra "." at the end

**R1-A11:** The extra "." was removed.

**R1-C12:** Line 802 - "substituted organic acid" - should it be "substituted aldehyde"?

**R1-A12:** We thank the reviewer for carefully reading our manuscript and finding this typo. Accordingly, "substituted organic acid" has been replaced by "substituted aldehyde or ketone". The revised sentence reads:

*"In general, the hydration constants $K_{hyd}$ decrease with decreasing electron-withdrawing power of the substituent in a substituted aldehyde or ketone (Clayden et al., 2012)."*

**R1-C13:** Line 820 - "hydrolysis" should be "hydration"

**R1-A13:** "hydrolysis" was replaced by "hydration".

**R1-C14:** Figure 12 caption - This caption uses "Fig. XX", while others all use "Figure XX"

**R1-A14:** The formatting was revised.

**R1-C15:** Line 886 - the phase "which was discussed in more detail for acetaldehyde, glyoxal, as well as methylglyoxal" is repeated.

**R1-A15:** The repeated part was removed.

**R1-C16:** Line 908 - "On the one hand"

**R1-A16:** "On the one hand," was replaced by "On one hand,"

**R1-C17:** Line 917 - Li et al. 2019 - citation format is inconsistent.

**R1-A17:** The citation format has been revised.

**R1-C18:** Line 1065 - "Overall, the different contribution of ETR and H-abstraction pathways modify also the product distribution as a function of pH." Does the author have any references for this statement?

**R1-A18:** This is a very good question. Actually, to the best knowledge of the authors, there is currently no detailed product study available on $NO_3$-radical reactions of carboxylic acids that investigated the product distribution under different pH conditions. However, aqueous-phase H-abstraction reactions of an undissociated acid by $NO_3$ are expected to lead to a functionalization of the acid. This chemical fate is well-known from several OH studies where the H-abstraction channel is predominant (see e.g., Leitner and Doré (1997), Tan et al. (2012), Otto et al. (2017)). On the other hand, ETR reactions by $SO_4^-$ and $NO_3$-radical leads to a decarboxylation of the acid (see Chawla and Fessenden (1975), and Exner et al. (1993)). This means, the ETR leads to a formation of $CO_2$ and a smaller carbon chain compound with one carboxyl group less. All in all, the products formed via the ETR and H-abstraction pathways are different.

Based on the well-taken reviewer hint, we have extended the discussion in the manuscript which reads now as follows:

*"On one hand, aqueous-phase H-abstraction reactions to leads mostly to a functionalization of the acid (see e.g., Leitner and Doré (1997); Tan et al. (2012); Otto et al. (2017)). On the other hand, ETR reactions of dissociated acids leads to a decarboxylation of the acid (Exner et al., 1994; Chawla and Fessenden, 2002) resulting in a formation of $CO_2$ and a smaller carbon chain compound."*

**R1-C19:** Line 1138 - "Therefore, the statements of Zhao et al. (2016) regarding the pH dependence in the reactivity of saturated carboxylic acids are by far too overgeneralized)" - the same authors have a recent paper showing that larger organic acids indeed do not exhibit much pH dependence. Please cite. https://doi-org.login.ezproxy.library.ualberta.ca/10.1021/acs.est.0c03331

**R1-A19:** A sentence was added referring to this recent paper. The added sentence reads as follows:

*"For the sake of completeness, a more recent study by the same authors (Amorim et al., 2020) show that larger organic acids indeed do not exhibit much pH dependence."*

**R1-C20:** Line 1190 - "due to of the"

**R1-A21:** "of " has been removed.

**R1-C21:** Line 1249 - "depend" should be "depends"

**R1-A21:** "depend" was replaced by "depends".

**R1-C22:** Line 1386 - Just wanted to double check if the authors have defined "OM".

**R1-A22:** "OM" replaced by organic matter (OM).

**References:**

[revised manuscript text omitted]

---

## Author Comment (AC3)

We thank reviewer#2 for the many constructive comments and suggestions which helped to improve the manuscript. In the following, we provide a point-by-point response to all comments. In order to improve readability, we numbered each reviewer comment and its corresponding response in the style R2-C1 for reviewer comments (C1 means comment 1 of reviewer 2) and R2-A1 for answers to the reviewer comment (A1 means answer to comment 1 of reviewer 2), respectively. Comments by the reviewer are given in black normal font, and our response to the comments is shown in blue. Newly added and modified text in the revised manuscript and supporting information (SI) is given in *italics.* Finally, we would like to mention that the manuscript was checked by an EPA internal review which also caused certain small changes in the revised manuscript (please see the manuscript with tracked changes).
* * *
**RC2**: 'Comment on acp-2021-58', Anonymous Referee #2, 21 Apr 2021

The paper at hand by Tilgner and co-workers is a review on acidity in the atmosphere, focusing on aspects of multiphase chemistry in aerosols and clouds. The comprehensive review is for the most part well-written and explains basic and complex concepts well. The author line comprises experts in the field. The topic of this review paper is ever-topical and relevant for the Atmospheric Sciences and thus a great fit for ACP. I can recommend publication after the below comments, provided together with some editing suggestions, are addressed.

**Specific Comments**

**R2-C1**:  *l. 62, 64 – "phosphorus"*

**R2-A1:**   The typos were revised.

**R2-C2**:   l. 67-70 – Sentence structure is a bit unclear.

**R2-A2:**   According to the reviewer comment, the sentence was revised and reads now as follows:

*On the other hand, the acidity of aqueous solutions can be buffered (see Fig. 1; Weber et al. (2016); Song et al. (2019); and Sect. 2.2 for details) by chemical interactions of (i) marine and crustal primary aerosol constituents (e.g., carbonates, phosphates, halogens), (ii) dissolved weak organic acids (e.g. formic acid, acetic acid, etc.), (iii) dissolved weak inorganic acids (e.g., $HNO_3$, HCl, HONO) and bases (e.g., ammonia and amines).*

**R2-C3**:   Figure 1 – Use of the arrow labels "+H^+" and just "H+" does not seem consistent. SOA reactivity should be acid-catalyzed, so without "+", phosphates should react quantitatively, so "+H^+". RCOO- protonation should be "+H^+" etc.

**R2-A3:**   The authors thanks the reviewer for this thoughtful comment. Figure 1 has been revised following the reviewer suggestion and the Figure now looks as follows:

[Figure]

**R2-C4:** l. 81 – "typical" should be written with lower case t.

**R2-A4:** The word is now written with lower case in the revised manuscript.

**R2-C5:** l. 82 - consider using different dashes or the word "to" for the indicated pH ranges

**R2-A5:** We replaced the small dash with an En dash to better indicate the pH ranges.

**R2-C6:** l. 89-91 – sentence has too many "and", please revise for readability

**R2-A6:** The sentence has been revised for readability. It now contains less "and", now reading as follows:

*"We first discuss the uptake of acidic and basic gases as well as buffering phenomena, then describe feedbacks between particle/droplet acidity, aqueous-phase inorganic (SO₂ oxidation and halogen) chemistry, and organic chemistry. Finally, a summary addresses atmospheric implications as well as needs for future investigations, for example, in the context of reduced fossil fuel combustion emissions of key acid precursors in a changing world."*

**R2-C7:** l. 97-99 – partitioning is a two-way process, so it's not clear why evaporation / back transfer is singled out here.

**R2-A7:** The description has been extended to point out the role of this process for the acidity buffering. So, the link to subsection 2.2 is now also clearer. The revised sentence reads now as follows:

*"Condensed-phase acidity also governs back transfer or evaporation of dissociating compounds into the gas phase - an important acidity buffering process (see Sect. 2.2)."*

**R2-C8:**  l. 102 – "aqueous phase"

**R2-A8:**  The dash was removed.

**R2-C9:**  l. 152 – I suggest "on" should be "for"

**R2-A9:**  "on" was replaced by "for".

**R2-C10:**  l. 156 – "acidity-dependent"

**R2-A10:**  A dash was inserted.

**R2-C11:**  l. 165 – please clarify use of the word "efficient", which usually means high outcome for low effort/expense

**R2-A11:**  Thanks for the comment, "efficient" was replaced by "more effective".

**R2-C12:**  l. 166 – "as long as the LWC does not limit the uptake": it is unclear what "limiting" means here. This should probably mean "solely limiting"? From Fig. 2, it seems that the parameter LWC is always influential, unless X is already 1. In turn, it is never uninfluential (which would be marked by vertical isolines), thus always "limiting"? Same with "restrict" in line 172.

**R2-A12:**  The reviewer is right that LWC is always an important factor determining the fraction of a compounds in the gas and aqueous phase. So, the intension of the second part of the sentence was only to remind the reader on this interplay of both acidity and microphysical conditions for the overall phase partitioning of an acid or base. According to the Reviewer comment, the sentence was now split into two sentences which now read as follows:

*"The partitioning into the aqueous phase is more effective for pH values well above the individual $pK_a$ values of each acidic compound. Below the individual $pK_{a,1}$ value, only the Henry's Law constant and the LWC limit the uptake. High LWCs (0.1-1 g $m^{-3}$) typically associated with cloud/fog conditions and, accordingly, less acidic media (pH > 4), favor phase partitioning towards the aqueous phase for most of the weak acids as well as for ammonia."*

The sentence in line 172 was also modified, see the next comment.

**R2-C13:**  l. 172 – Please define "typical aerosol conditions" in analogy to your definition of cloud/fog conditions.

**R2-A13:**  Thanks for this hint. Typical aerosol conditions (pH, ALWC) are now defined in the revised manuscript in analogy to our definition of cloud/fog conditions in the text above. The revised manuscript reads as follows:

*"Under typical aerosol conditions (0 ≤ pH ≤ 4 (see e.g. Pye et al. (2020)); $10^{-6}$ ≤ ALWC ≤ $10^{-4}$ g $m^{-3}$ (see e.g. Herrmann et al. (2015)), the LWC restricts uptake and only very small fractions of the less water-soluble and weak acids can partition in the aqueous particle phase due to their $pK_a$ values (typically above 4)."*

**R2-C14**:  l. 222-223 – The use of different significant digits for pKa and pH here is rather confusing.

**R2-A14**:  To avoid any confusion, the same digits are now used for both pKa and pH in this section.

**R2-C15**:  l. 234-234 – "very high and very low acidity conditions show significantly increased buffering capacities": I am not entirely sure what is referred to here. I think it might be worthwhile explaining the black dashed and solid lines in Fig. 3 here (and why they are dashed or solid, respectively).

**R2-A15**:  Based on the reviewer comment, the text description of Figure 4 has been extended to clarify the buffering capacity behavior. Further, to avoid any confusion, the dashed line is now explained in the legend as pure water system without any other buffers. The revised text reads as follows:

*"Eq. 5 and the plotted examples in Fig. 4 reveal that very high and very low acidity conditions show significantly increased buffering capacities. The first term $(ln10 \cdot \frac{K_W}{[H]^+})$ and second term $(ln10 \cdot [H]^+)$ of Eq. 5 represent the terms for water ($H^+$ and $OH^-$ respectively) and creates the lower buffering capacity limits (dashed lines in Fig. 4a) with a minimum at pH 7 (not shown in in Fig. 4a). The first and the second term of Eq. 5 leads to high $\beta$ values at high and low pH conditions, respectively. The third term adds additional buffering capacity of all other buffers in the aqueous solution. So, added buffers in the solution can introduce local maxima of $\beta$ between very acidic and very alkaline conditions, where the contribution of the first and the second term to the $\beta$ is small."*

The revised Figure is as follows:

[Figure]

*Figure 4.      Buffering capacity $\beta$ of water, ammonia/ammonium and carbonate/bicarbonate/carbonic acid (top: from left to right) as well as formic and acetic acid (bottom: from left to right) as a function of pH. The atmospherically relevant range of cloud and aerosol pH is marked in yellow, and the $pK_a$ values of the corresponding buffers are marked with dotted pink lines.*

**R2-C16**:  Figure 5 – This figure is very busy and difficult to disentangle for the reader. I believe the mention of "blue text" for microphysical processes is missing from the caption. The distinction of

molecules (small grey spheres) and particles (small light red spheres) is somewhat difficult to spot. What happens to the small spheres inside the bottom large cloud droplet, labelled "aqueous phase acid production"? What is indicated by the different colors of arrows?

**R2-A16:** Figure 5 has been revised, please see the revised Figure below. In detail, the gas-phase molecules were replaced by chemical structures, some arrows were removed and the illustration of the in-cloud processing of CCNs were replaced by the text ("CCN aging"). Finally, the caption was extended by a description of the "blue text". The revised Figure is given below and the revised caption reads as follows:

[Figure]

*Figure 5.     Schematic of sources (red text) and conditions of acidity in different aqueous aerosol particles (green text) together with microphysical and chemical processes that are able to influence the acidity of tropospheric aerosols (Fig. created after Raes et al. (2000) and McMurry (2015)). The blue text describes microphysical processes of CCNs and cloud/fog droplets. The dashed gray line represents an aerosol number size distribution based on McMurry (2015).*

**R2-C17:** l. 302 – "anthropogenic primary sources of acidic and alkaline aerosols (see Fig. 5)": where can this be seen in Fig. 5?

**R2-A17:** The "anthropogenic primary sources of acidic and alkaline aerosols" are included in the "Emissions of primary particles (incl. acidic / basic components)".

**R2-C18:** l. 316 – It might be beneficial to briefly explain the concept of "less acid displacement" or at least add "due to" and omit parenthesis.

**R2-A18:** The sentence was rephrased and a link to the companion article of Pye et al. (2020) was added, where the concept of "acid displacement" is emphasized in more detail.

The revised text reads as follows:

*"These environments are characterized by higher contribution of organic acids and chloride due to (i) lower rates of acid displacement (see e.g., Pye et al. (2020) and references therein for further details on this topic) and (ii) lower abundances of sulfate and nitrate mass (see precipitation composition data compiled by Vet et al. (2014))."*

**R2-C19:** Figure 6 – "Total S(IV) rate with and without taking into account ionic strength at the maximum reported limit is shown": Please indicate which line corresponds to which number. The complex figure is overall not much discussed in the text, panels b and c seem not mentioned. Please make clear how and why this figure is different to Fig. 7 and why it is needed in this complexity. What is the dashed line in panel b? Why do lines not add up to the black dashed line in panel c?

**R2-A19:** The authors thanks the reviewer for the detailed look at the Figure and the thoughtful comments. In accordance with the reviewer's comments, Figure 6 has been restructured to better represent the content and messages of the image. One intention of Figure 6 is to illustrate the kinetic aspects and changes resulting from the S(VI) oxidation in aerosol under different initial aerosol conditions. Therefore, initial rates and rates after 10s are depicted and compared. To better illustrate this and enable a better comparison, the two panels are now plotted closer together in the revised Figure as panels a) and b), see below. Then, Figure 6 aims to illustrate the resulting rapid changes in pH under different initial aerosol conditions (see panel c)). So, Figure 6 shows that higher S(IV) to S(VI) oxidation rates under weakly acidic conditions (pH > 5) quickly generate sufficient H$^+$ (after 10 s only) resulting in significant decrease in the pH compared to the initial pH. Thus, in the absence of buffering or a chemical OH$^-$ source compensating for acidification, less acidic or even slightly basic particles are rapidly acidified in the tropospheric multiphase system. Fig. 6 then also shows that processes that are initially important under low acidity conditions quickly become less important as the aerosol acidifies. For example, the importance of the O$_3$ and HNO$_4$ reaction drops significantly after 10 s, while the H$_2$O$_2$ oxidation is still at a similar level. Additionally, Figure 6 also intends to point out the potential influence of the ionic strength of the total S(IV) rate, although these effects have not yet been extensively studied yet. All in all, the purpose and contents of Figure 6 are somewhat different compared to Figure 7.

In order to improve the readability of the Figure and, especially, the different lines in panel a) and b), the legend is now outside of the two panels and significantly larger. To avoid any confusion between the two dashed black lines plotted in the original Figure, the lines in panel c) are now plotted with different colors and dashed-dotted lines. Moreover, we included a separate legend for panel c). We apologies for the error in the former panel c) and thank the reviewer again for finding the error. Due to the reviewer comment, that panel (formerly panel c and now panel b) was revised, so that all lines sum up to the black dashed line now. The new Figure 6 and its revised caption as well as the extended discussion of the Figure is as follows:

[Figure]

*Figure 6. S(IV) oxidation rates for Beijing winter haze conditions (following (Cheng et al., 2016)). Shown are a) initial S(IV) oxidation rates (top left), b) S(IV) oxidation rates after 10 s of reaction (bottom left), and c) aerosol pH after 0 s and 10 s of reaction as a function of the initial aerosol pH (bottom right). In the upper right legend, the S(IV) oxidation rates of the different oxidants ($NO_2$, $HNO_4$, $O_3$, $H_2O_2$, Fe and Mn) shown in a) and b) are listed together with total S(IV) to S(VI) rates both with and without taking into account ionic strength at the maximum reported limit. Rates used were those recommended in this text.*

"Figure 6 illustrates that S(IV) oxidation under urban haze conditions can significantly contribute to the acidification of aerosols on a very short timescale. After a short period of chemical processing, aerosols are expected to reach pH 4.5 or lower. Particularly for haze particles with initial pH conditions above 4, a fast acidification can be modeled as a consequence of the higher initial S(IV) oxidation rates under less acidic conditions. Figure 6 shows that higher S(IV) to S(VI) oxidation rates under weakly acidic conditions (pH > 5) quickly generate sufficient $H^+$ (after only 10 s) resulting in significant decrease in the pH compared to the initial pH. Thus, in the absence of buffering or a chemical $OH^-$ source compensating for acidification, less acidic or even slightly basic particles are rapidly acidified in the troposphere. This is also known for freshly-formed sea salt particles which rapidly become acidified within minutes after their emission characterized by a pH drop by about four pH units (Angle et al., 2021). Furthermore, Fig. 6 illustrates that processes that are initially important under low acidity conditions quickly become less important as the aerosol acidifies. For example, the importance of the $O_3$ and $HNO_4$ reaction drops significantly after 10 s, while the $H_2O_2$ oxidation is still at a similar level. To better understand this issue, in the next subsections, we outline the major S(IV) oxidation pathways, their sensitivity to the pH of the aqueous medium, and their potential to alter pH through the formation of acidic products.

**R2-C20**: l. 409 – Please clarify what k' refers to.

**R2-A20**: Thanks for the comment, k' represents reaction rate constant of proton-catalyzed pathway (R-6a). Thus, k' were replaced by $k_{6a}$ in the revised manuscript.

**R2-C21**: l. 414 – "kinetics"

**R2-A21**: "kinetic" was replaced by "kinetics".

**R2-C22**: l. 468 – It seems there is a word missing before "of".

**R2-A22**: "of" was removed from the sentence. The revised text reads as follows:

*"The high concentrations of organic material in aerosols can quench radical and triplet species (Herrmann et al., 2010; McNeill, 2015; Wang et al., 2020)."*

**R2-C23**: *l. 498-502 – Please revise overly long and complex sentence with multiple sub clauses, missing commas and maybe missing words ("of" in line 500?).*

**R2-A23**: The long sentence was split into 3 sentences to improve the readability. The revised sentences now read as follows:

*"Finally, it should be noted that great care is needed for estimating the rate of R-10 because of (i) lacking knowledge about the present PS\* concentrations in ambient aerosols and cloud droplets as well as (ii) the very rapid quenching and deactivation triplet species by water, dissolved oxygen and organic/inorganic aerosol constituents. The latter might lead to very low PS\* concentrations which can strongly limit or inhibit this pathway similarly to the S(IV) oxidation by free radicals. This oxidation pathway can be effectively inhibited by particle constituents other than S(IV) as described earlier in the present section."*

**R2-C24**: l. 535 – "thus" is superfluous here.

**R2-A24**: ", thus" was removed from the sentence.

**R2-C25**: l. 586 – It is not clear what "artifact HONO" is, please clarify.

**R2-A25**: Based on the reviewer comment, we have extended the description in this paragraph. The paragraph reads now as follows:

*"This mechanism (R-14a – R-15b) was invoked to explain the formation of 'artifact HONO' in a wet denuder when both $NO_2$ and $SO_2$ are present in the ambient gas phase. The study of Spindler et al. (2003) aimed at measurements of gas-phase HONO. However, chemical interactions of dissolved $NO_2$ and $SO_2$ at wetted denuder walls can lead to the formation of the two long-lived intermediates $[NO_2 - SO_3]^{2-}$ and $[NO_2 - HSO_3]^-$ (see R-14a and R-14b) which decays into $NO_2^-$ and $SO_3^{2-}$. In order to quantify this artificial HONO formation and subsequently correct the measured HONO, kinetic data of this reaction system (R-14a – R-15b) were experimentally determined in the study of Spindler et al. (2003) by measuring $NO_2$ in aqueous solution with a laser photolysis-broadband optical absorption experimental set-up. For this review, the kinetic data of Spindler et al. (2003) have been again*

*kinetically analyzed in more detail. The measurements of Spindler et al. (2003) were performed at pH = 4.5 and pH = 10 to investigate either the $HSO_3^-$ or the fully deprotonated form $SO_3^{2-}$. From the T-dependent rate constants (see Table S2) of the forward ($k_{14a}$, $k_{14b}$) and backward reaction ($k_{-14a}$, $k_{-14b}$), the equilibrium constants ($K_{14a}$, $K_{14b}$) were calculated and the Arrhenius expressions were derived as follows."*

**R2-C26**: l. 590-591 – "via this mechanism": I think this has to be explained in slightly more detail. I presume adduct formation effectively reduces the concentration of SO2, hereby limiting potential for S(VI) formation? Please also state how "under environmental conditions" are important here, which is not clear to the reader.

**R2-A26**: The key point here is not that the concentration of $SO_2$ is reduced, rather than that the formation of the two long-lived intermediates $[NO_2 - SO_3]^{2-}$ and $[NO_2 - HSO_3]^-$ limits the overall conversion of S(IV) to S(VI). The term "under environmental conditions" was only added to express, that the originally intended investigations on unwanted side-reaction processes on wetted denuder walls, provided kinetic data which are applicable for different acidity conditions present in the environment.
To better clarify our argumentation, we have extended the paragraph below the reaction mechanism. The revised text reads now as follows:

*"Finally, from the measurements of 'artifact HONO' in the Spindler et al. (2003) publication, the unimolecular rate of decomposition for the adduct was determined as $k_{15a}(T) = (8.4\pm0.1)\ 10^{-3}\ s^{-1}$ (T = 298 K).*
*The most significant difference between the results of Spindler et al. (2003) and earlier studies is that the mechanism identified by Spindler et al. (2003) includes the adduct formation with a slow adduct decomposition (see R-14a – R-15b) which considerably limits the potential for S(VI) formation via this mechanism under environmental conditions. Here, from the viewpoint of aqueous-phase thermochemistry, it should also be noted that such high rate constants for a prompt bimolecular reaction with a concerted single electron transfer from $HSO_3^-$ to $NO_2$ would not be feasible. The one-electron reduction potentials of $NO_{2(aq)}$ and $HSO_{3\ (aq)}^-$ are very similar with $E°(SO_3^-/HSO_3^-) = 0.84$ V vs. NHE at pH = 3.6 (Huie and Neta, 1984; Wardman, 1989) and $E°(NO_2/NO_2^-) = 1.04 \pm 0.02$ V vs. NHE (Armstrong et al., 2013) and, as a consequence, a fast reaction would not be in line with the very limited energetical driving force of the reaction as its Gibbs free enthalpy of reaction. For comparison, the redox potential $E°(SO_3^-/SO_3^{2-})$ is 0.63 V vs. NHE at pH > 7 (Huie and Neta, 1984; Wardman, 1989) implying a faster reaction rate at higher pH.*
*The oxidation of S(IV) by $NO_2$ in aerosol water was previously proposed to be important during wintertime haze episodes in Beijing (Cheng et al., 2016; Wang et al., 2016). The significance of this S(IV) oxidation pathway rests on (a) the hypothesis that aerosols in Beijing have an unusually high pH of about 7 (Wang et al., 2016), which is not supported by thermodynamic models (see Pye et al. (2020) with an average pH value of approximately 4 for China), and (b) the mechanism and relatively fast kinetic parameters of earlier studies by Lee and Schwartz (1983) and Clifton et al. (1988) without considering the more recent findings of Spindler et al. (2003) and the underlying thermochemistry. For completeness, the significantly different S(VI) rates resulting from the different kinetic parameters of Lee and Schwartz (1983), Clifton et al. (1988) and Spindler et al. (2003) considering the $NO_2$ and $SO_2$ conditions for wintertime haze conditions based on Cheng et al. (2016) are shown in Fig. S1 in the Supporting Information."*

**R2-C27**: l. 594 – The reduction potential of HSO3- needs to be referenced to another compound, I assume this is the SO3- radical here. Would this reduction potential not be pH-dependent and lower at higher pH, thus confirming a fast reaction rate at high pH? Thus, please add reduction

potentials for the SO3-*/SO3^(2-) pair for comparison. In which pH range would the SO3-*/SO3^(2-) potential be important? See also next comment.

**R2-A27:** Thanks for this correction hint. We have added "$SO_3^-$/" into the parenthesis and added also the corresponding pH value of the E° value. Furthermore, the reviewer is right that reduction potential is lower at higher pH. This pH behavior has kinetic consequences as the protonation slows down the rates of electron reactions (Wardman (1989)). In turn, this implies a faster reaction rate at higher pH. Based on the $pK_A$ value of 7 (see Table S1) for $HSO_3^-$/$SO_3^{2-}$, it can be calculated that 1% is present as $SO_3^{2-}$ at pH 5, 50% at pH 7, and about 100% from pH 10.

The revised manuscript text reads as follows:

*"The one-electron reduction potentials of $NO_{2(aq)}$ and $HSO_3^-{}_{(aq)}$ are very similar with $E°(SO_3^-/HSO_3^-) = 0.84$ V vs. NHE at pH = 3.6 (Huie and Neta, 1984) and $E°(NO_2/NO_2^-) = 1.04 \pm 0.02$ V vs. NHE (Armstrong et al., 2013) and, as a consequence, a fast reaction would not be in line with the very limited energetical driving force of the reaction as its Gibbs free enthalpy of reaction. For comparison, the redox potential $E°(SO_3^-/SO_3^{2-})$ is 0.63 V vs. NHE at pH > 7 (Huie and Neta, 1984; Wardman, 1989) implying a faster reaction rate at higher pH.)"*

**R2-C28:** l. 598 - "which is not supported by thermodynamic models". Please give a reference or clearly state inputs/outputs, model etc. How much of a difference are we talking here? Can you state the numbers that Cheng et al. assume, the numbers that models return and potentially provide measurement data?

**R2-A28:** The authors fully agree with the reviewer that more information is needed here. Thus, we have (i) added the pH value (≈7) considered by Wang et al. (2016), (ii) included a reference to the comprehensive acidity review of Pye et al. (2020) including the average aerosol pH value given there for China mainland (average of approximately 4). Finally, additional text was added together with a new Figure in the Supporting Information to better address the differences in the kinetic data and their impacts on the S(VI) oxidation rates.
The revised text and the new Figure in the Supporting Information are as follows:

*"The oxidation of S(IV) by $NO_2$ in aerosol water was previously proposed to be important during wintertime haze episodes in Beijing (Cheng et al., 2016; Wang et al., 2016). The significance of this S(IV) oxidation pathway rests on (a) the hypothesis that aerosols in Beijing have an unusually high pH of about 7 (Wang et al., 2016), which is not supported by thermodynamic models (see Pye et al. (2020) with an average pH value of approximately 4 for China), and (b) the mechanism and relatively fast kinetic parameters of earlier studies by Lee and Schwartz (1983) and Clifton et al. (1988) without considering the more recent findings of Spindler et al. (2003) and the underlying thermochemistry. For completeness, the significantly different S(VI) rates resulting from the different kinetic parameters of Lee and Schwartz (1983), Clifton et al. (1988) and Spindler et al. (2003) considering the $NO_2$ and $SO_2$ conditions for wintertime haze conditions based on Cheng et al. (2016) are shown in Fig. S1 in the Supporting Information."*

[Figure]

*Figure S1. Comparison of the calculated S(IV) oxidation rates by the $NO_2$ reaction pathway in mol $L^{-1}$ $s^{-1}$ using the different reaction rate constants of Lee and Schwartz (1983), Clifton et al. (1988) and Spindler et al. (2003) for the urban winter haze scenario of Cheng et al. (2016).*

**R2-C29**: Figure 7 – The colors of CH3C(O)OOH and HNO4 are indistinguishable in my digital version of the manuscript.

**R2-A29**: Due to the reviewer comment, we have changed the colors of $CH_3C(O)OOH$. The revised Figure is as follows:

[Figure]

**R2-C30**: l. 645: "artificially low H2O2 concentrations" – I might have missed this in the paper, but can you provide references for more appropriate H2O2 concentrations? What exactly is "artificial" about the concentration used in that paper?

**R2-A30:** Thanks for the comment, we have added few more sentences to better clarify our statements. The revised text part reads as follows:

*"Only by the combination of applying unusually high aerosol pH values, artificially low $H_2O_2$ and $O_3$ concentrations as well as unrealistically fast kinetic parameters from earlier studies by Clifton et al. (1988) (see subsection 4.5 above), $NO_2$ rates can fall into the range of other key oxidants discussed here*

*(see Cheng et al. (2016)). In detail, the used $H_2O_2$ and $O_3$ concentrations of 0.01 ppb and 1 ppb used by Cheng et al. (2016) for urban haze conditions are far too low. Recent measurements of $H_2O_2$ and $O_3$ concentrations under haze conditions in the North China Plain (Ye et al., 2018a; Fan et al., 2020; Ye et al., 2021) showed substantially higher values of about 0.5 ppb and 10 ppb, respectively."*

**R2-C31**: l. 645: Why can the Clifton et al. (1988) data not be used, what is "unrealistically fast" and why?

**R2-A31**: This is explained above and in the revised manuscript, see **R2-A26**.

**R2-C32**: l. 673-675 – This sentence is not clear. I suppose reactive halogens are a sink for nitrogen oxides? That is hard to understand here, please revise.

**R2-A32**: The reviewer is right that reactive halogens can acts as an effective sink for $NO_x$. According to the reviewer comment, we have revised the sentence which reads now as follows.

*"Tropospheric reactive halogens can impact the oxidation capacity of the atmosphere by (i) acting as an effective sink for ozone ($O_3$), e.g. during bromine explosion events in the Arctic, (ii) acting as an effective sink for nitrogen oxides ($NO_x = NO + NO_2$) and (iii) by influencing the $HO_x$ ($= OH + HO_2$) (Oltmans et al., 1989; Simpson et al., 2015; Schmidt et al., 2016; Sherwen et al., 2016; Hoffmann et al., 2019)."*

**R2-C33**: l. 731 – "constants" (plural)

**R2-A33**: The word was corrected.

**R2-C34**: l. 758 – Please define "DOM".

**R2-A34**: DOM is now defined in the revised manuscript.

**R2-C35**: l. 772-785 – Aldehydes and ketones are not the only carbonyl compounds in the atmosphere, please revise.

**R2-A35**: In accordance to the reviewer comment, we have revised this text. The revised text now reads as follows:

*"5.1    Acidity and hydration of aldehydes or ketones*

*Aldehydes or ketones are omnipresent in the tropospheric gas and aqueous phase, result from primary emissions or are secondary oxidation products. The photolysis of aldehydes or ketones can be important for both their degradation in the troposphere and gas-phase oxidant production. Water-soluble aldehydes or ketones may partition into the aqueous phase of deliquesced aerosols and cloud/fog droplets. Once in the aqueous phase, these compounds can undergo hydration, leading to conversion of carbonyl group into gem-diol moieties. As hydration processes are typically acid- or base-catalyzed, the acidity of an aqueous solution can affect the hydration and consequently all other processes linked to it. With regard to phase partitioning, the hydration equilibria increase the effective partitioning of the carbonyl-containing compound towards the aqueous phase (Sumner et al., 2014). Moreover, compared to the carbonyl group, the diol functionality is photochemically inactive. Thus, partitioning*

*to the aqueous phase and subsequent hydration can, in part, protect aldehydes or ketones from photolysis and shut off possible photochemistry of the carbonyl group (George et al., 2015; Herrmann et al., 2015; McNeill and Canonica, 2016). However, hydrated aldehydes are often characterized by a somewhat lower reactivity with radical oxidants such as OH compared to the unhydrated carbonyl species (Schuchmann and von Sonntag, 1988).*
*This sub-section summarizes the present knowledge on the acidity dependence of carbonyl group hydration constants, and implications for the chemical conversions of aldehydes or ketones in atmospheric aqueous media."*

**R2-C36**: l. 811 – "H3O+ ion": This review might benefit from a systematic treatment of H+ / H3O+. Usage of H3O+ here, while in other places "H+" was used, suggests that the water molecule is important here, is that the case?

**R2-A36**: Generally, Section 5.1.1. refers to aqueous systems, but since the term $H_3O^+$ occurs only four times in the manuscript, we have replaced $H_3O^+$ to $H^+$ in the manuscript as suggested by the reviewer. The changes are in equation 20, in the text at lines 988 and 811 in the former manuscript, and in Table 2.

**R2-C37**: l. 820 – Please indicate to what process "hydrolysis" refers to here.

**R2-A37**: We apologize, this is a typo. It should read "hydration". We have changed the text accordingly.

**R2-C38**: l. 842 – "The impact of acidity and its feedback [...] performed at TROPOS." Please provide references here and in the entire following paragraph unless this is original research conducted for this paper.

**R2-A38**: We thank for the comment, we have added the reference for clarification. The measurements were carried out by T. Schaefer as part of his doctoral thesis. The revised text reads as follows:

*"The impact of acidity and its feedback on the hydration, as well as their impact on the photochemistry of pyruvic acid have been examined by spectroscopic investigations performed at TROPOS by Schaefer (2012)."*

**R2-C39**: l. 867 – Please clarify for me the difference of "acid-driven" and "require acidity". Does the latter mean that participation of a proton occurs in a reaction that is not rate-limiting?

**R2-A39**: Based on the reviewer's comment, we have rewritten the relevant section to avoid inconsistencies with respect to the term "acid-driven" in the manuscript. It reads now:

*"Many organic accretion reactions are acid- or base-driven or, in some cases, even acid-catalyzed. In these acid-driven reactions, the protons ($H^+$) in the reactions are incorporated into the reaction products formed (e.g., ring opening of epoxides, cf. Sect. 5.3) and therefore are not "acid catalyzed"."*

**R2-C40**: l. 885 – I believe there is a redundant instance of "which has been discussed in more detail for acetaldehyde, glyoxal as well as methylglyoxal".

**R2-A40:**  We thank the reviewer for the comment. We have removed the redundancy from the manuscript text. The text reads now:

*"This pathway has been suggested as a source of light-absorbing secondary organic material (i.e., brown carbon) in atmospheric aerosols (Laskin et al., 2015; Nozière et al., 2015), which was discussed in more detail for acetaldehyde, glyoxal as well as methylglyoxal (Noziere and Esteve, 2005; Nozière et al., 2007; Noziere and Esteve, 2007; Noziere and Cordova, 2008; De Haan et al., 2009; Shapiro et al., 2009; Bones et al., 2010; Sareen et al., 2010; Li et al., 2011; Yu et al., 2011; Kampf et al., 2012; Nguyen et al., 2012; Laskin et al., 2015; Lin et al., 2015; Maxut et al., 2015; Nozière et al., 2015; Van Wyngarden et al., 2015; Aiona et al., 2017; Rodriguez et al., 2017)."*

**R2-C41:**  l. 900 – "occurs"

**R2-A41:**  We have corrected the typo in the manuscript text.

**R2-C42:**  l. 920 – It is not clear what is meant with "the nitrogen nucleophile is more important than the acid-catalyzed aldol condensation", because one is functional group/property, the other one is a type of chemical reaction.

**R2-A42:**  We thank the reviewer for the hint. We revised the text accordingly, it reads now:

*"This indicates that the nitrogen, as a nucleophile, is more important than the acid-catalyzed aldol condensation, which is consistent with the observation of Kampf et al. (2012), Kampf et al. (2016) and Yi et al. (2018)."*

**R2-C43:**  l. 922 – I suppose "conduct" should read "conducted"

**R2-A43:**  We have corrected the typo in the manuscript text.

**R2-C44:**  l. 925 – What does "difficulty for ammonium addition" mean here?

**R2-A44:**  According the reviewer comment, we have rephrased the text for clarification. It now reads:

*"A theoretical analysis of glyoxal condensation in the presence of ammonia conducted by Tuguldurova et al. (2019) describes different imidazole formation pathways by the formation of key intermediates, namely, ethanediimine, diaminoethanediol, and aminoethanetriol, required for the imidazole ring cyclization. These authors reported that the imine concentrations are very low due to the high-energy barriers for imine formation. Although a pH decrease due to amino alcohol dehydration leads to higher imine concentrations, but also to higher ammonium cation formation, which hinders due to the electrostatic interactions the ammonium addition to the carbonyl group. Hence, the reaction pathway via the ethandiimine as intermediate is not the main reaction pathway. The second proposed pathway of glyoxal condensation in the presence of ammonia, which includes the formation of the intermediate aminoethanetriol, has a lower energy and apparently is kinetically more favorable due to the higher concentration of this intermediate. Finally, imidazole formation is determined by the glyoxal concentration, the ratio of glyoxal/ (amine or ammonium), the composition of the solvent, and the pH value."*

**R2-C45**: l. 927 – "due to higher concentrations": What species is referred to here? "Higher" compared to what? The entire paragraph is rather condensed and hard to understand. It is not always clear when aldol addition and when aldol condensation is referred to.

**R2-A45**:  Please, see the answer to the comment above (**R2-A44**).

**R2-C46**: l. 929 – "All in all, aldol condensations are today generally regarded as demanding to drastically acidic conditions to be really important in particle and multiphase chemistry." This un-referenced statement seems out of place as most studies in the paragraph seem to show an increased formation rate with increasing pH and l. 900 "aldol condensation only occur at a pH = 4-5", please clarify.

**R2-A46**:  According to the reviewer's comment and because of the still conflicting pH dependencies in the literature, we have removed this sentence.

**R2-C47**: l. 972 – Which "oxidation" is referred to here?

**R2-A47:**  We have rephrased the text for clarity. It now reads:

"The oxidation of organic compounds leads in general to carboxylic acids and proceeds through α- or β-hydroxy acid to esters or oligoesters, similarly to the proposed mechanisms for oligomers in the aerosol phase (Gao et al., 2004; Tolocka et al., 2004; Surratt et al., 2006; Surratt et al., 2007)."

**R2-C48**: l. 1049 – It is not clear what is meant by "Similar to the gas phase radical (…)", please revise.

**R2-A48**:  Here, there was a comma missing between "gas phase" and "radical oxidants". This typo has been corrected and "Similar to the" was replaced by "As in the". The revised sentence reads as follows:

*"As in the gas phase, radical oxidants such as OH and NO$_3$ can react with dissociating organic compounds via H-abstraction."*

**R2-C49**: Sect. 5.4.2.2 – Could you give a reason for the increased reactivity of NO3 with deprotonated acids? I presume a higher electron density changes the reduction potential. Can this be proven using tabulated values?

**R2-A49**:  The main reason for the higher reactivity, as described in the section, lies in the different reaction mechanism for the undissociated acid (H-abstraction) and the dissociated acid (electron transfer reaction), respectively. There is surely a higher electron density for the deprotonated acid (DeRuiter 2005). However, a proof with tabulated values of the reduction potential is hardly possible because missing tabulated data of some one-electron couples necessary. One possibility would be the calculation of the different electron densities by means of theoretical chemical calculations. However, this would be a study by its own.

**R2-C50**: l. 1200 – "This higher reactivity can be explained by the higher electron-withdrawing properties of the carboxylate." – Should this not be the "less electron-withdrawing properties of

the carboxylate" (compared to the carboxylic acid)? I would expect the –I effect to be smaller for the negatively charged group. This is also stated in l. 1208: "From inductive effect theory, it is known that the COOH group is electron-withdrawing and COO- is electron-donating."

*R2-A50:* This was an error in the text. This higher reactivity can be explained by the higher electron-donating properties of the carboxylate. The sentence was revised and reads now as follows:

*"This higher reactivity can be explained by the higher electron-donating properties of the carboxylate."*

*R2-C51*: l. 1207-1208 – "The deprotonation likely leads to a reduction in the electron density at the carbon-carbon double bond enabling an easier O3 addition, i.e. a more rapid oxidation." – Should this not read "increase in the electron density"? See argument in the previous comment.

*R2-A51:* This was an error in the text. The deprotonation leads to an increase in the electron density. The manuscript was revised and reads now as follows:

*"The deprotonation likely leads to an increase in the electron density at the carbon-carbon double bond enabling an easier $O_3$ addition, i.e. a more rapid oxidation."*

*R2-C52*: l. 1368 – The term "aqSOA" is not defined in this manuscript.

**R2-A52:** The term is now explained.

*R2-C53*: l. 1375 – One occurrence of "organic" seems redundant.

**R2-A53:** The second "organic" was removed.

[revised manuscript text omitted]